# Transferable and Adaptable Driving Behavior Prediction

## Abstract

While autonomous vehicles still struggle to solve challenging situations during on-road driving, humans have long mastered the essence of driving with efficient, transferable, and adaptable driving capability. The obvious gap between humans and autonomous vehicles keeps us wondering about the essence of how humans learn to drive. Inspired by humans' cognition model and semantic understanding during driving in a hierarchical learning procedure, we propose HATN, a hierarchical framework to generate high-quality, transferable, and adaptable predictions for driving behaviors (intentions and trajectories)in multi-agent dense-traffic environments. Our hierarchical method consists of a high-level intention identification policy and a low-level trajectory generation policy. We introduce a novel semantic definition for the two policies and generic state representation for each policy, so that the hierarchical framework is transferable across different driving scenarios. Besides, our model is able to capture variations of driving behaviors among individuals and scenarios by an online adaptation module. We demonstrate our algorithms in the task of trajectory prediction for real traffic data at intersections and roundabouts from the INTERACTION dataset, the InD dataset, and the Argoverse dataset. Through extensive numerical studies, it is evident that our method significantly outperformed other methods in terms of prediction accuracy, transferability, and adaptability. Pushing the performance by a considerable margin, we also provide a cognitive view of understanding the driving behavior behind such improvement. We highlight that in the future, more research attention and effort are deserved for the transferability and adaptability of autonomous driving planning and prediction algorithms. It is not only due to the promising performance elevation, but more fundamentally, they are crucial for the scalable and general deployment of autonomous vehicles.

## 1 Introduction

When autonomous vehicles are deployed on roads, they will encounter diverse scenarios varying in traffic density, road geometries, traffic rules, etc. Each scenario comes with different levels of difficulty in understanding and predicting future behaviors of other road participants. Even in a straight street with few road entities, the sensor system of AVs still confronts a daunting amount of information that may or may not be relevant to the behavior prediction task. Let alone in more complex scenarios like crowded, human-vehicle-mixed, complicated-road-geometry intersection or roundabouts, currently deployed AVs tend to timidly take conservative behaviors due to insufficient prediction capability. On the contrary, in many safe driving cases, humans can drive through and across these environments easily, even while talking to friends or shaking to the music.

Moreover, most state-of-art behavior prediction (intention and trajectory prediction)algorithms for AVs, once trained for one scenario, are brittle due to overspecialization and tend to fail when transferred to similar or new scenarios. On the contrary, when a green-hand human driverlearns to understand and predict the behaviors of other drivers at one intersection, such an experience is omni-instructional, also helping to enhance behavior understanding and prediction capability in other intersections and roundabouts.

There is an obvious gap of capability between AVs and human drivers. We naturally wonder what is the secret in humans' brains, which allows us to understand and predict driving behavior so easily and efficiently. Evident from neuroscience, human's efficient shuttling in dense traffic flows and complex environments ben-

efits from two cognition mechanisms: 1) hierarchy (Botvinick et al., 2009; Flet-Berliac, 2019) - cracking the entangling task into simpler sub-tasks; 2) selective attention (Niv, 2019; Radulescu et al., 2019) - identifying efficient and low-dimensional state representations among the huge information pool. Certainly, the two mechanisms are not mutually exclusive but are highly co-related. When dividing a complex task into easier sub-tasks, humans will choose a compact set of low-dimension states relevant to each sub-task respectively. An easy example can be found when a child learns to build a tower with blocks. Usually, the child would divide the overall task into a sub-task of searching for proper blocks and a sub-task of cautiously placing the blocks on the tower (Marcinowski et al., 2019; Spelke & Kinzler, 2007). In the high-level searching task, the child would care about the shape or weight of the blocks, but in the low-level placing task, the child would essentially pay attention to subtly adjusting the position and angle of the block. By choosing state features at different granularity (information hiding) and learning different skills separately (reward hiding) (Dayan & Hinton, 1993; Bacon et al.), children are not only able to build the blocks efficiently and rapidly due to the simplified task and filtered state (efficient learning), but they are also capable of generalizing and reusing the two skills when they confront new scenarios or tasks (generalization).

The benefits of the two mechanisms in efficient learning and generalization are certainly fruitful for hierarchical methods, which end-to-end approaches (Salzmann et al., 2020; Codevilla et al., 2018) cannot enjoy. However, how much we can benefit from these two mechanisms significantly depends on how properly the hierarchies and relevant states are designed. To this point, there are some existing works (Zhao et al., 2020; Gao et al., 2020; Tang & Salakhutdinov, 2019) dividing the driving task into a high-level intention-determination task and a low-level action-execution task. An intention is usually defined as a goal point in the state space (Zhao et al., 2020; Ding et al., 2019; Sun et al., 2018a) or the latent space (Tang & Salakhutdinov, 2019; Rhinehart et al., 2019). Actions are then generated to reach that goal.

However, to gain human-level high-quality and transferable prediction capability, the definition of hierarchy should carry more semantics by referring to how humans think while driving (Shalev-Shwartz et al., 2017). When humans are shuttling through traffic flows, they first exhibit high-level intention to identify which "slot" is most spatially and temporally proper to insert into as shown in Figure 1(a). With the chosen slot to insert into and the map geometry, humans then will generate a desired reference line as a low-level action as shown in Figure 1(b). Then humans will polish their micro-action skills by optimizing how well they can track the reference line.

Such a hierarchical policy with more profound semantics enjoys many advantages. First, the policy is intrinsically scenario-transferable and reusable, because the representation of insertion slot and reference trajectory can be abstracted out and consistently defined across different scenarios. Second, the hierarchical design encourages efficient learning since each sub-task's state space is reduced where only relevant information for the sub-task is left, and each sub-policy's learns individually without information entanglement.

In addition, human behavior is naturally stochastic, heterogeneous, and time-varying. For instance, humans with different driving styles (Wang et al., 2021; Sun et al., 2018b; Schwarting et al., 2019) may result in distinct observed behaviors. Besides, though transferable, human behavior is still task-specific because there exist inevitable distribution shifts across scenarios, making the generalization harder. For instance, speed limits are set differently across different scenarios or cities, calling for driving customization on each scenario. Capturing such behavior variance can not only help to make more accurate customized behavior prediction for individuals, but also encourages better generalization across scenarios. As a result, an advanced prediction algorithm should also harness the power of online adaptation, to embrace the uncertainty in human behavior.

In summary, to generate high-quality, transferable and adaptable driving behavior prediction in multi-agent systems, we should not only design policies by leveraging human's intrinsic hierarchy and selective attention cognition model, but also capture diverse human behaviors with online adaptation methods. However, such a design is not trivial. Harmonious and natural divisions of hierarchies, along with compact and generic state representations, are crucial to achieve what we desire. Also, to seamlessly incorporate adaptation methods into the hierarchy policies, strict mathematical formulations and systematic analysis are required.

In this paper, we propose HATN (*Hierarchical Adaptable and Transferable Network*), a hierarchical framework for high-quality, transferable and adaptable behavior prediction in multi-agent traffic-dense driving environments. The framework consists of three parts: 1) a high-level semantic graph network (SGN) responsible

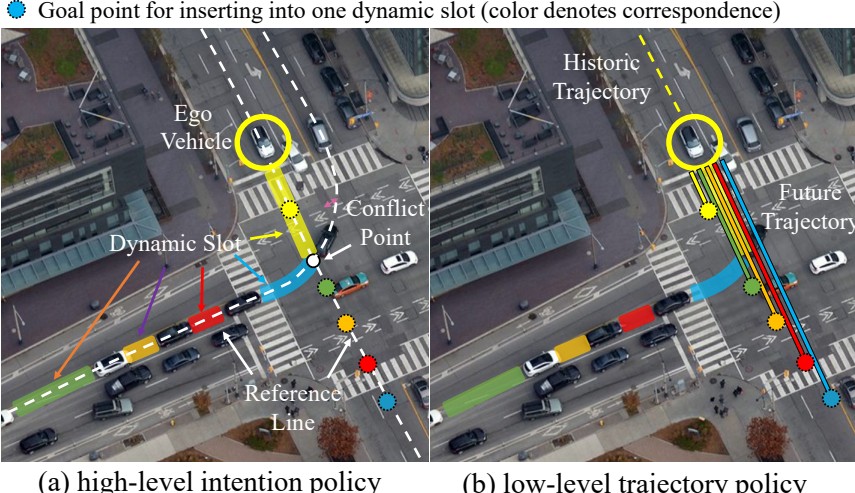

(a) high-level intention policy    (b) low-level trajectory policy

Figure 1: Figure updated.In dense-traffic multi-agent driving scenarios, human drivers have a hierarchical process in understanding and generating driving behavior, which cares about different features in each hierarchy. Humans firstly predict which dynamic slot to insert into in the high-level intention hierarchy as in (a), paying high attention to features related to the dynamic slots on the scene. Then humans are predicted to execute actions to realize the intention in the low-level trajectory hierarchy as in (b), based on finer features related to vehicle dynamics. Such a hierarchical prediction process is 1) simplifying the learning by dividing the whole task into easier sub-tasks, which only takes in state relevant to its sub-goal; 2) intrinsically scenario-transferable as the representation for the "slots" and the "trajectory" can be defined consistently across different scenarios; 3) adaptable to individuals by leveraging historic behaviors to adjust model parameter. One can refer to Fig 5 6 7 for dynamic illustration of the insertion process.

for the slot-insertion task in multi-agent environments; 2) a low-level trajectory encoder decoder network (TEDN)which generates future trajectory according to historic dynamics and intention signals from the high-level policy; 3) an online adaptation module which applied modified extended Kalman filter (MEKF) algorithm to execute online adaptation for better individual customization and scenario transfer. To the best of our knowledge, this is the first method to explicitly and simultaneously take the driver's nature of hierarchy, transferability, and adaptability into account.

In addition to being deployed for high-quality real-time transferable and adaptable trajectory predictions, the proposed method can have diverse practical applications. When the online adaptation module which requires the feedback signal is deactivated, the methods can also be applied across different scenarios as a planning system to: 1) generatemore user-friendly and socially-compatible driving behaviors; 2) providedriving suggestions as an automatic driving assistance system, such as which slot to insert into; 3) reportemergency alert when unsafe driving behavior happens. In the paper, we evaluate our method in the task of trajectory prediction since there are more abundant real data for numerical evaluation.

The key contributions of this paper are as follows:

1. Propose a hierarchical framework that takes novel sub-task definitions with compact and generic representations. Thus our framework canefficiently generates accuratedriving behavior prediction in complex multi-agent intense-interaction environments, which is zero-shot transferable across scenarios.

2. Leverage online adaptation algorithms to capture behavior variance among individuals and scenarios for better customizability and transferability. A new set of metrics is proposed for the systematic evaluation of online adaptation performance.

3. Conduct extensive experiments on real human driving data, which include thorough ablation studies for each module of our method and show how our method outperforms other behavior forecasting methods, in terms of prediction accuracy, transferability and adaptability.

## 2 Related works

Our preliminary results were presented in non-archival workshop. The current version further provides: 1) comprehensive comparison with more state-of-the-art methods in Sec 7.5; 2) detailed description of the methodology in Sec 3 and semantic graph representation in Sec 4; 3) in-depth experiment for evaluation of the online adaptation in Appendix D; 4) discussion on the interacting agent density in Appendix E and algorithm running time Appendix 7.6. In this section, we introduce related works on human behavior prediction categorized by methodology and property. The readers are referred to Rudenko et al. (2020) for a detailed survey.

### 2.1 Traditional prediction methods

The problem of predicting future motion for dynamic agents has been long studied in the literature. Classic physics-based methods include Intelligent Driver Model (Treiber et al., 2000), Kalman Filter (Elnagar, 2001), Rapidly Exploring Random Trees (Aoude et al., 2010), etc. These methods essentially analyze agents and propagate their historic and current state forward in time according to manually designed physical rules. Other classic optimization-based methods model humans as utility-maximization agents, whose future behavior can be predicted by assuming they are optimizing designed or learned reward (Fridovich-Keil et al., 2020; Wang et al., 2021). Other classic pattern-based methods classify driving motion into semantically interpretable maneuver classes, via Hidden Markov Model (Liu & Tomizuka, 2016; Deo et al., 2018), Gaussian Process (Zhang et al., 2021), and Bayesian Network (Schreier et al., 2014). Such classes are then used to facilitate intention-and-maneuver-aware prediction. These methods perform well in scenarios with simple road geometry and weak interaction like highways and straight streets. However, these methods struggle when confronting long-horizon prediction tasks or complex-road-geometry intense-interaction scenarios like crowded roundabouts and intersections. Such performance downgrade usually stems from the limited expressiveness of the model, insufficient interaction and context encoding, and laborious but uncomprehensive task-specific rule design.

### 2.2 Deep-learning-based prediction methods

The success of deep learning ushered in a variety of data-driven methods. Due to the temporal nature of the prediction task, these models often utilizes therecurrent neural network (RNN) variants to process temporal information (Park et al., 2018; Hu et al., 2018a; Zyner et al., 2019; Dequaire et al., 2018; Deo & Trivedi, 2018). When modeling agents' interaction, one intuitive idea is simply using Deep Neuron Network (DNN), by flattening features of all agents and feeding them into deep neuron networks. However, such designs lack flexibility as they usually only consider a fixed number of agents. On the other hand, These methods are also not order-invariant: processing agents in a different order would produce different results, while we would expect the same results for the same scene. Thus such methods have been rarely used in related works. To bypass these problems and further impose agents' spatial relationship in the reasoning process, Convolution Neural Network (CNN) applies convolution operations on data (commonly in grid or pixel form) to model spatial and temporal relationships, such as 3D voxelization, rasterization in 2D bird's-eye view (BEV) and occupancy grid (Radwan et al., 2020; Su et al., 2021; Itkina et al., 2019; Toyungyernsub et al., 2021; Lange et al., 2021; Mohajerin & Rohani, 2019; Thomas et al., 2022; Mahjourian et al., 2022). However, such representations have several drawbacks: 1) there is an uneasy trade-off between the resolution of the spatial grid/image and the field of view. 2) the occupancy grid does not take into account HD map-related information such as lane and lane relations, which are important for accurate future prediction, especially in interactive scenarios. 3) representations obtained from the BEV images or occupancy grids via CNNs are of high abstraction level, which may fail under scenarios that are not well covered by the training data. 4) explicit relationship reasoning among agents is still missing, and it is difficult to capture long-range interactions via convolutions with small receptive fields.To avoid or tackle these drawbacks, thegraph

neural network (GNN) has been recently combined with RNN and CNN to model agent interaction in the prediction tasks (Salzmann et al., 2020; Ding et al., 2019; Li et al., 2019; Ma et al., 2019; Choi et al., 2019; Li et al., 2021; Hu et al., 2020; 2018b; Li et al., 2020). Due to the strong relational inductive bias of GNN (Battaglia et al., 2018), these GNN-based methods successfully achieve flexibility in agent number, ordering invariance, and explicit relationship reasoning. Consequently, in this paper, we also adopt a GNN-based architecture. Readers are also referred to Wang et al. (2022) for detailed suvery on interaction modelling and driver behavior prediction.

### 2.3 Hierarchical prediction

In the driving task, there inherently exists a high-level intention determination sub-task and a low-level motion execution sub-task. Thus the methods which predictsfuture motion in an end-to-end manner (Salzmann et al., 2020; Li et al., 2019; Ma et al., 2019) are usually hard to learn, explain, and verify. To this point, some existing methods exploit the hierarchies in the driving task (Gao et al., 2020; Choi et al., 2019; Li et al., 2021). An intention is usually defined as a goal point in the state space (Ding et al., 2019; Sun et al., 2018a) or the latent space (Tang & Salakhutdinov, 2019; Rhinehart et al., 2019). However, though equipped with hierarchies, most of these methods are still trained in an end-to-end manner. Consequently, the two benefits of hierarchies, task simplification and representation filtering, are hardly exploited since the model is still monolithically trained. There are a few works that not only utilizea hierarchical design, but also trains the model hierarchically in two stages (Zhao et al., 2020).However, the representation of intention in these works is far from generic, and the transferability of the algorithm is still ignored and not verified. In comparison, our method is not only able to monitor and refine the learning in each hierarchy, but also enhance transferability by designing generic and compact representation for each sub-policy so that they can be reused in different scenarios.

### 2.4 Transferable prediction

It is desired to have omnipotent prediction algorithms that can be applied to many different scenarios such as highways, intersections, and roundabouts. However, most existing methods either focus on certain scenarios (Ding et al., 2019; Choi et al., 2019), or train and apply their methods in data from various scenarios without assessing their scenario-wise performance (Salzmann et al., 2020; Li et al., 2019; Ma et al., 2019; Li et al., 2021). As a result, these methods tend to fail when transferred to novel scenarios without learning a new set of model parameters. Many factors contributes to such brittleness, while the representation may be the first to be blame. The input features of these methods are usually in Cartesian coordinate frame or scene images, which leads to two drawbacks: 1) scenario-specific information such as traffic regulations and road geometries are softly and insufficiently incorporated if not completely ignored. 2) these representations are not generic and will change each time a new scenario is encountered.

There have been effective representation definition for efficient and generalizable learning in some tasks, such as the visuomotor control of humanoids where the environment is divided into proprioceptive and exteroceptive state (Merel et al., 2018; Hasenclever et al., 2020), and the visual navigation task where image target is defined (Zhu et al., 2017). However, in the driving context where complex road/traffic information and intense interaction exist, very few works can address the representation issues. There are recent works that vectorized the scene information (lane reference, agent trajectory, etc.) and apply learning modules on such vectorized scene representation for scene understanding relationship reasoning (Zhao et al., 2020; Gao et al., 2020; Liu et al., 2021). Although vectorized representations are applicable to various driving scenarios, all relations between road or agent, which can sometimes be pretty simple and obvious, have to be learned by the network and there is no guarantee that those known relations can be learned correctly. Besides, those works only consider the static multimodality in the environment (such as vehicle's keeping straight, turning). The dynamic and multi-modal agent's interaction is sufficiently considered.One recent work (Hu et al., 2020) achieves transferable prediction by designing generic representations called Semantic Graph, where Dynamic Insertion Areas rather than road entities are regarded as the nodes. In such representations, scenario-specific information such as road geometry and traffic regulations is naturally and comprehensively incorporated in a generic manner. The dynamic relationship/multi-modality among agents are also considered. However, this work only considers the high-level intention prediction sub-task and cannot generate motion trajectory for

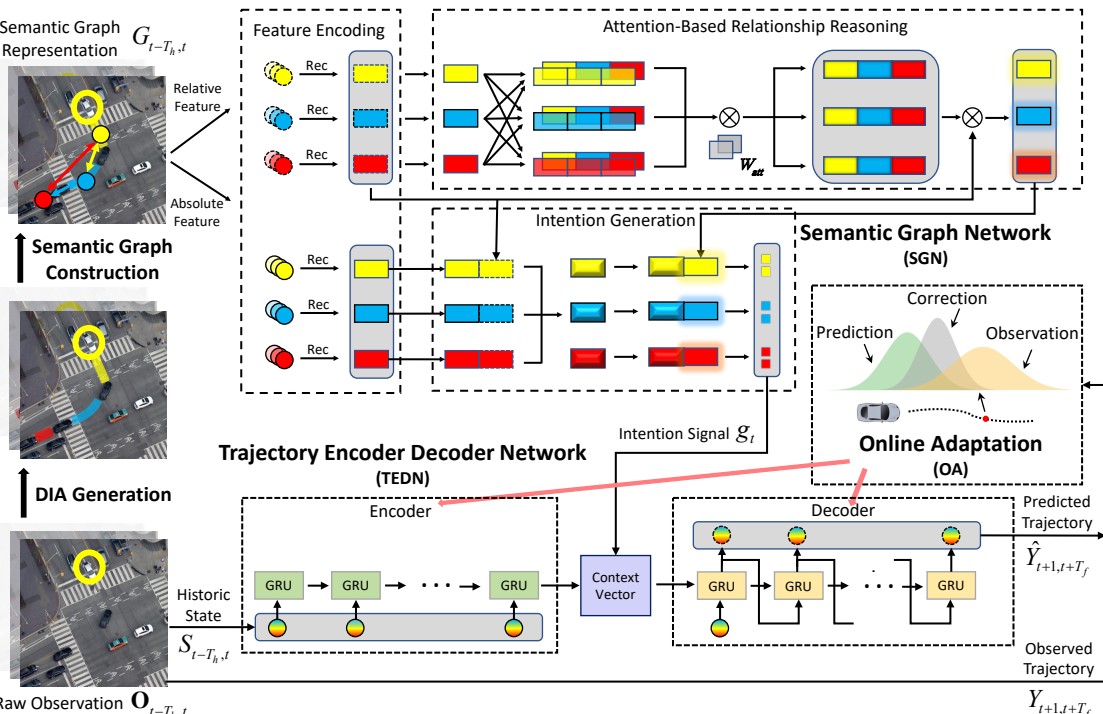

Figure 2: Figure updated.The proposed HATN framework consists of four parts: 1) on the left of the image, we extract ego vehicles' interacting cars and construct a Semantic Graph (SG). In the SG, Dynamic Insertion Areas (DIA) are defined as the nodes of the graph, which the ego vehicle can choose to insert into. 2) Taking the SG as the input, the high-level Semantic Graph Network (SGN) is responsible for reasoning about relationships among vehicles and predicting the intentions of individual vehicles, such as which area to insert into and the corresponding goal state. 3) The low-level trajectory encoder decoder network (TEDN)takes in each vehicle's historic dynamics and intention signal to predict their future trajectories. 4) The Online Adaptation (OA) module can online adapt TEDN's parameter based on historic prediction errors, which captures behavior pattern of different individuals and scenarios. The input and output for each module are clarified formally in Table 1.

practical use. In contrast, our paper adopts such generic representations and train the model hierarchically, which enables us to conduct both intention prediction and motion prediction simultaneously.

## 2.5 Adaptable prediction

Apart from hierarchies and transferability, adaptability is another desired property of the prediction algorithms since there inevitably exists behavior variance in different individuals and scenarios. Consequently, online adaptation has been recently studied in the human prediction research field. Cheng et al. (2020; 2019); Si et al. (2019) adopts recursive least square parameter adaptation algorithm (RLS-PAA) to adapt the parameter of the last layer in the neural network. Abuduweili & Liu (2021); Liu & Liu (2021); Abuduweili & Liu (2020) utilize modified extended Kalman filter (MEKF$_\lambda$) to conduct model linearization, so that multi-step adaptation on random sets of parameters is doable. These method also empirically compared the performance of adapting different sets of parameters with different steps.The recent work most related to ours (Abuduweili & Liu, 2021) adopts an encoder-decoder architecture for multi-task prediction, namely tasks of intention and trajectory prediction. However, compared to our method, the benefit of GNN, hierarchical training, and transferable generic representation is not considered.

# 3 Problem formulation

With the proposed HATN framework, we aim at generating high-fidelity predictions of driving behaviors in multi-agent traffic-dense scenarios[1]. Specifically, with the proposed method shown in Figure 2, we focus on generating behavior predictions for any selected car (which will be called the ego car in the following discussion) and the cars interacting with the ego car in the next $T_f$ seconds $\hat{\mathbf{Y}}_{t+1,t+T_f}$, based on observations of the last $T_h$ seconds $\mathbf{O}_{t-T_h,t}$:

$$\hat{\mathbf{Y}}_{t+1,t+T_f} = f_{HATN}(\mathbf{O}_{t-T_h,t}). \tag{1}$$

High-fidelity trajectory prediction is challenging especially when the horizon extends, where the intention and inter-vehicle interaction have increasingly larger impacts on driving decisions. Evident from cognition science (as discussed in Sec 1), when humans drive in dense traffic flows, their decision-making policy naturally consist of hierarchies. Specifically, in the high-level hierarchy, human drivers intuitively search for the proper slot to insert into. Thus we first adopt a generic representation about the environment called the semantic graph (SG). In SG, dynamic insertion areas (DIA) are defined as the node of the graph, among which the vehicles can decide to insert into or not. Such a representation is compact, efficient, and generic, which captures sufficient information for intention determination and can be generically used across different driving scenarios. Illustrated in the left part of Figure 2, the process of extracting semantic graph representation $\mathcal{G}_{t-T_h,t}$ from raw observations $\mathbf{O}_{t-T_h,t}$ can be formally described:

$$\mathcal{G}_{t-T_h,t} = f_{SG}(\mathbf{O}_{t-T_h,t}), \tag{2}$$

where $\mathcal{G}_{t-T_h,t} \in \mathbb{R}^{M \times T_h \times N_1}$ denotes the extracted semantic graph, consisting of $M$ DIAs from the past $T_h$ step, each with $N_1$ features. $\mathbf{O}_{t-T_h,t} \in \mathbb{R}^{M \times T_h \times N_2 + K}$ is the environment observation including $K$ reference lines, and $M-1$ vehicles interacting with the ego carin the past $T_h$ steps, each with $N_2$ features. Namely there is $M-1$ DIAs corresponding to the $M-1$ interacting vehicles respectively, and there is another DIA corresponding to the ego vehicle.$f_{SG}$ denotes the SG extraction function that selects cars interacting with ego car, extracts DIAs, and constructs SG.

With the semantic graph, we then propose a semantic graph network (SGN), which takes the semantic graph as input, inferences relationships and interactions among vehicles, and outputs two intention features: 1)the probability for ego car to insert into each of the $M$ DIAs $w_t \in \mathbb{R}^M$; 2) the distribution of the goal state in the future horizon$p(g_t) \in \mathbb{R}^{M \times J}$ for ego car and the interacting cars:

$$w_t, p(g_t) = f_{SGN}(\mathcal{G}_{t-T_h,t}), \tag{3}$$

where for each of the $M$ vehicles, $J$ parameters are used to describe the distribution of the goal state.

In the low-level hierarchy, our insight is that when humans drive, conditional on the goal state as the intention signal, they conceptually track a reference trajectory via micro muscle actions. Thus we design a low-level trajectory-generation policy to imitate such behavior. Besides the intention signal, the trajectory generation procedure should also be subject to the instantaneous dynamics of the vehicles, which requires the encoding of the historic state $\mathbf{S}_{t-T_h,t}$. Furthermore, the policy should also be able to express various motion patterns, i.e. constant velocity and varying acceleration. To ensure dynamics continuity and motion diversity, we use the trajectory encoder decoder network (TEDN)as the behavior-generation model, where the historic dynamics are processed by the encoder and then used by the decoder to generate diverse maneuvers. With model parameter $\theta$, the task of TEDNis illustrated as in the lower part of Figure 2 and can be summarized as:

$$\hat{\mathbf{Y}}_{t+1,t+T_f} = f_{TEDN}(\mathbf{S}_{t-T_h,t}, g_t, \theta), \tag{4}$$

where $\mathbf{S}_{t-T_h,t} \in \mathbb{R}^{M \times T_h \times N_3}$ denotes the state of $M$ vehicles in the past $T_h$ time step, each with $N_3$ features. $\hat{\mathbf{Y}}_{t+1,t+T_f} \in \mathbb{R}^{M \times T_f \times N_4}$ denotes the prediction for the $M$ vehicles in the future $T_f$ time steps, each with $N_4$ features.

---

[1]This paper considers intersection and roundabout scenarios, which are relatively interaction-intense, while our method can be easily applied to other scenarios like highway and parking lot.

Up to this point, our model is capable of generating high-level intentions and low-level trajectories efficiently and transferably. However, the trained model can only capture the motion pattern in an average sense, while the nuances among individuals are hardly reflected. Besides, the behavior patterns also vary with different scenarios. To capture these behavior nuances across different individuals and scenarios, we set up an online adaptation module (OA), where a modified Extended Kalman Filter ($MEFK_\lambda$) algorithm is used to moderately adjust the model parameters for each agent based on its historic behaviors. Specifically, we regard TEDN as a dynamic system and estimate its parameter $\theta$ by minimizing the error between ground-truth trajectory $\mathbf{Y}_{t-\tau,t}$ and predicted trajectory $\hat{\mathbf{Y}}_{t-\tau,t}$ in the past $\tau$ steps:

$$\theta_t = f_{MEKF_\lambda}(\theta_{t-1}, \mathbf{Y}_{t-\tau,t}, \hat{\mathbf{Y}}_{t-\tau,t}), \tag{5}$$

where $\hat{\mathbf{Y}}_{t-\tau,t} \in \mathbb{R}^{M \times \tau \times N_4}$ denotes the prediction for $M$ vehicles from $\tau$ time step earlier to now, and $\hat{\mathbf{Y}}_{t-\tau,t} \in \mathbb{R}^{M \times \tau \times N_4}$ denotes the ground-truth observation of the $M$ vehicles in the past $\tau$ time steps, each with $N_4$ features.

The rest of this paper is organized as follows. In Sec 4, we introduce the high-level intention-identification policy in detail, including the definition of semantic graph and the architecture of semantic graph network. In Sec. 5, we describe the low-level behavior-generation policy, including the design of trajectory encoder decoder network (TEDN) and the method of integrating the intention signal into the TEDN. In Sec. 6, we introduce the formulation and the utilized algorithm ($MEKF_\lambda$) of online adaptation module. Note that the possible design choices of each module are systematically discussed in each corresponding section. In Sec. 7, we conduct extensive empirical studies of our methods on real data, including a case study illustrating how our method works in Sec. 7.2, an evaluation on the SGN in Sec 7.3, a brief summary of ablation studies on TEDN and OA as a quick takeaway for the reader in Sec 7.4, and a comparison with other methods in Sec 7.5. More detailed ablation studies on TEDN and OA to empirically find the optimal design choice can be found in Appendix B C D.

Table 1: Input and output for each module in the proposed framework.

| Module | Input | Output |
|---|---|---|
| SGN | Semantic graph (Dynamic insertion area) | Probability distribution of future inserting area and goal state |
| TEDN | Most likely goal state, Historic dynamics | Most likely future trajectory |
| OA | Historic observation, Historic prediction, Prior TEDN parameter | Updated TEDN parameter, Adapted future trajectory |

## 4 High-level intention-identification policy

Human behaviors are usually hierarchically divided for better efficiency and generalizability. In this section, we introduce the high-level intention-identification policy in detail, including design insight, the definition of semantic graph, and the architecture of semantic graph network (SGN).

In the high-level policy, humans usually take in low-dimension state feature to make decisions at a low resolution (Dayan & Hinton, 1993; Botvinick et al., 2009; Niv, 2019). Specifically in the driving task, human drivers first make a high-level decision on which area on the road is the most temporally and spatially suitable to insert into. Such areas are usually formed by the slots between cars, traffic signs, and road geometries. To imitate humans' intention of inserting into slots in dense traffic, we first adopt the dynamic insertion area (DIA) introduced in Hu et al. (2020) to define the slot formally. The extracted DIAs are then regarded as nodes to form a semantic graph to construct a generic and compact representation of the scenario. We then introduce the semantic graph network which generates agents' intention by reasoning about their internal relationships. The advantages of adopting dynamic insertion area are threefold: (1) It explicitly describes humans' insertion behavior considering the map, traffic regulations, and interaction information. (2) It filters scene information and only extracts a compact set of vehicles and states crucial for the intention prediction task. (3) DIA is a generic representation, which can be used across different scenarios.

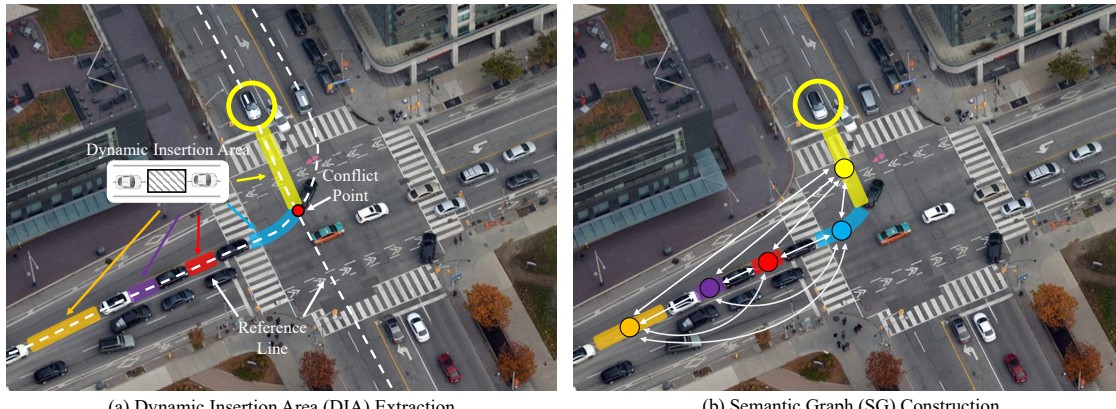

(a) Dynamic Insertion Area (DIA) Extraction  (b) Semantic Graph (SG) Construction

Figure 3: Dynamic insertion area (DIA) extraction and semantic graph (SG) construction procedure: when other car's lane reference line crosses ego car's lane reference line, based on the conflict point, the DIAs are extracted and regarded as nodes to construct the SG.

## 4.1 Semantic graph

The semantic graph utilizes dynamic insertion areas as basic nodes for a generic spatial-temporal representation of the environment. As shown in Figure 3, when extracting DIAs from the scene, we first identify each agent's reference line by Dynamic Time Warping algorithm (Berndt & Clifford, 1994). Next we identify interacting cars whose lane reference crosses with the ego car's lane reference. Interaction essentially happens among these cars and the ego car as they are driving into a common area (the conflict point) (Markkula et al., 2020). Then DIAs are extracted by the definition: *a dynamic area that can be inserted or entered by ego agent on the road.* Each DIA consists of a front boundary formed by a front agent, a rear boundary formed by a rear agent, and two side boundaries formed by reference lines. Note that DIAs can be formed by vehicles, traffic signs and road geometries. Specifically, the rear boundary of one DIA is always a vehicle, but the front boundary of one DIA can be a vehicle, a traffic sign (e.g. stop sign and yield sign), the end point of one lane reference line, or a conflict point of two lane reference lines.To capture each DIA's crucial information for humans' decision, we extract four high-level features under the Frenet coordinate for each DIA: $\mathbf{X} = (d_{f/r}^{lon}, v_{f/r}, \phi_{f/r}, l)$, where $d_{f/r}^{lon}$ denotes longitudinal distance to the conflict point of front or rear boundary; $v_{f/r}$ denotes the velocity of front or rearboundary; $\phi_{f/r}$ denotes the angle of front or rearboundary; $l = d_r^{lon} - d_f^{lon}$ measures the length of the DIA. These features $\mathbf{X}$ are defined as the absolute features of one DIA. To facilitate relationship inference among DIAs, we also define the relative feature $\mathbf{X}'$for each DIA by aligning it with the reference DIA. Specifically, we choose the front DIA as the reference DIA $\mathbf{X}_{ref}$, because the ego vehicle is implicitly represented by the rear boundary of the front DIA. Each DIA's relative feature $\mathbf{X}'$ is then derived by subtracting the DIA's absolute feature $\mathbf{X}$ by the reference DIA's absolute feature $\mathbf{X}_{ref}$.

With the extracted DIAs as nodes, the 3D spatial-temporal semantic graph $\mathcal{G}^{t-T_h \to t} = (\mathcal{N}^{t-T_h \to t}, \mathcal{E}^{t-T_h \to t})$ can be constructed, where $t - T_h \to t$ denotes the time span from a previous time step $t - T_h$ to current time step $t$ with $T_h$ denoting the horizon. Such a representation differs from other methods in the criteria of choosing interacting cars and the definition of node in the graph, which is discussed in detail in Appendix E. The readers are referred to Hu et al. (2020) for more detailed description on the DIA properties and DIA extraction algorithm.

## 4.2 Semantic graph network

As the high-level intention identification policy, the architecture of the SGN is shown in Figure 2. SGN takes the spatial-temporal 3D semantic graph from historic time step $t - T_h$ to the current time step $t$ as the input, rather than only the spatial 2D semantic graph of the current time step $t$ in previous work (Hu et al., 2020). Such a change aims at capturing more temporal dynamics and interactions among vehicles. SGN

then decides which area to insert into and generate the associated goal state distribution. For simplicity, the mean of the goal state distribution is then delivered to a low-level policy for generating more human-like behaviors. The impact of sampling in the goal state distribution is left for future work to discuss.

### 4.2.1 Feature encoding layer

In this layer, we essentially encode the absolute and relative features for each node from historic time step $t - T_h$ to the current time step $t$:

$$h_i^t = f_{rec}^1(\mathbf{X}_i^t), \tag{6}$$

$$h'^t_i = f_{rec}^2(\mathbf{X}'^t_i), \tag{7}$$

where $\mathbf{X}_i^t = [x_i^{t-T_h}, ..., x_i^t]$ and $\mathbf{X}'^t_i = [x'^{t-T_h}_i, ..., x'^t_i]$ respectively denote the absolute features and relative features of node $i$ from time step $t - T_f$ to current time $t$; $h_i^t$ and $h'^t_i$ denote the hidden states encoded from absolute and relative features respectively, namely the outputs of the recurrent function $f_{rec}^1$ and $f_{rec}^2$. $h_i^t$ and $h'^t_i$ are further embedded for later use.

$$\hat{h}_i^t = f_{enc}^1(h_i^t), \tag{8}$$

$$\hat{h}'^t_i = f_{enc}^2(h'^t_i). \tag{9}$$

### 4.2.2 Attention-based relationship reasoning layer

To infer relationships between any two nodes, inspired by Graph Attention Network (Veličković et al., 2017), we design an attention-based relationship reasoning layer. In this layer, we exploit the soft-attention mechanism (Luong et al., 2015; Bahdanau et al., 2014) to compute the node $n_i$'s attention coefficients on node $n_j$:

$$a_{ji}^t = f_{att}(concat(\hat{h}'^t_j, \hat{h}'^t_i); \mathbf{W}_{att}), \tag{10}$$

where function $f_{att}$ maps each concatenated two features into a scalar with the parameter $\mathbf{W}_{att}$. The attention coefficient is then normalized across all nodes $\mathcal{N}^t$ at time step $t$:

$$\alpha_{ji}^t = \frac{exp(a_{ji}^t)}{\sum_{n \in \mathcal{N}^t} exp(a_{ni}^t)}. \tag{11}$$

Eventually, node $i$'s relationships with all nodes in the graph (including node $i$ itself) are derived by the attention-weighted summation of all encoded relative features:

$$\bar{h}_i^t = \sum_{n \in \mathcal{N}^t} \alpha_{ni}^t \odot \hat{h}'^t_n, \tag{12}$$

where $\odot$ denote element-wise multiplication.

### 4.2.3 Intention generation layer

In the high-level policy, there are two intention features to be predicted for each vehicle. The first is which DIA the vehicle will insert into. This is a classification problem and our model will output the probability of inserting into each DIA $w_i$. Another intention signal is the distribution of the goal point in a certain future $g$. When predicting these intention features, in addition to DIAs' relationships beween each other, each DIA's own features are also required. Thus we first concatenate and encode each node's embedded absolute and relative feature:

$$\widetilde{h}_i^t = f_{enc}^3(concat(\hat{h}_i^t, \hat{h}'^t_i)). \tag{13}$$

Each DIA's future evolution in the latent space is then derived by combining its relationships with other nodes and its own features:

$$z_i^t = f_{enc}^4(concat(\bar{h}_i^t, \widetilde{h}_i^t)). \tag{14}$$

Thus the latent vector representing each DIA's evolution is then used to predict the first intention feature, the probability of one DIA being inserted by the ego vehicle:

$$w_i^t = \frac{1}{1 + exp(f_{out}^1(z_i^t))}, \tag{15}$$

which is then normalized across all DIAs in current time step such that $\sum_{i \in \mathcal{N}^t} w_i^t = 1$. The second intention feature, the goal state of each vehicle, is also need to guide the trajectory prediction in the low-level policy. Thus we further use a Gaussian Mixture Model (GMM) to generate a probabilistic distribution over each vehicle's future goal state $g$ in a certain horizon[2]:

$$f(g_i^t|z_i^t) = f(g_i^t|f_{out}^2(z_i^t)), \tag{16}$$

where the function $f_{out}^2$ maps the latent state $z_i^t$ to the parameters of GMM (i.e. mixing coefficient $\alpha$, mean $\mu$, and covariance $\sigma$). The goal state $g$ then can be retrieved by sampling in the GMM distribution.

### 4.2.4 Loss function

We not only expect the largest probability to be associated with the actual inserted area ($\mathcal{L}_{class}$), but also the ground-truth goal state to achieve the highest probability in the output distribution ($\mathcal{L}_{regress}$). Thus we define the loss function as:

$$\begin{aligned}
\mathcal{L} &= \mathcal{L}_{regress} + \beta \mathcal{L}_{class} \\
&= -\sum_{\mathcal{G}_s} \left( \sum_{i \in \mathcal{N}^s} log\Big(p\big(\check{g}_i|f_{out}^2(z_i)\big)\Big) + \beta \sum_{i \in \mathcal{N}^s} \check{w}_i log(w_i) \right),
\end{aligned} \tag{17}$$

where $\mathcal{G}_s$ denotes all the training graph samples; $\mathcal{N}^s$ denotes all the nodes in one training graph sample; $\check{g}_i$ and $\check{w}_i$ denote the ground-truth label for goal state and insertion probability of node $n_i$. Though our goal is to predict the ego vehicle's future motion, we output the goal state for all interacting vehicles rather than only the ego vehicle (as done in Hu et al. (2020)) to encourage sufficient reasoning of interactions, and also realize data augmentation.

Note that though only the goal state $g_t$ is delivered and used in the downstream low-level behavior prediction, the learning for insertion probability $w_t$ serves as an auxiliary task to stabilize the goal state learning (Mirowski et al., 2016; Hasenclever et al., 2020). Defining goal state in the state space instead of the latent space also offers us accessible labels to monitor the high-level policy learning and provides more interpretability of the model. The detailed description of the layers can be found in Table 5 of the appendix, and detailed empirical evaluations on the high-level policy can be found in Sec 7.3.

## 5 Low-level behavior-generation policy

Once the high-level policy determines where to go, the low-level policy is then responsible to achieve that goal by processing information at a finer granularity. Thus in this section, we describe the low-level behavior-generation policy, including the architecture of trajectory encoder decoder network (TEDN), the methods of integrating the intention signal into the TEDN, and possible design choices which requires empirical studies at the end of the section.

In the driving task, humans will generate a sequence of micro actions like steering and acceleration based on vehicle dynamics to reach their goals. To generate future behaviors of arbitrary length and ensure sufficient expressiveness, we use the trajectory encoder decoder network (TEDN)(Cho et al., 2014; Neubig, 2017) as the low-level behavior-generation policy, given the historic information and intention signal from the high-level policy. The low-level policy enjoys two benefits from the hierarchical design: 1) the learning is simplified as the low-level policy only needs to care about vehicle's own dynamics, while the consideration for interactions, collision avoidance, road geometries are left to the high-level policy to take care (information hiding); 2)

---

[2]Note that each DIA's rear bound is formed by one vehicle, so that the movement of the DIA and the rear vehicle is correlated. Besides, in this paper, we use the relative traveled distance in future 3 seconds as the goal state representation.

the policy is only optimized for reaching the goal (reward hiding), so that the effect of different design and training tricks can be better verified explicitly; 3) the learned policy is tranferable and reusable in different scenarios.

## 5.1 Trajectory encoder decoder network (TEDN)

The TEDNconsists of two GRU (graph recurrent unit) networks, namely the encoder and the decoder. At any time step $t$, the encoder takes in the sequence of historic and current vehicle states $\mathbf{S}_{t-T_h,t} = [\mathbf{s}_{t-T_h}, ...\mathbf{s}_t]$, and compresses all information into a context vector $c_t$. The context vector is then fed into the decoder as the initial hidden state to recursively generate future behaviors $\hat{\mathbf{Y}}_{t+1,t+T_f} = [\hat{\mathbf{y}}_{t+1}, ..., \hat{\mathbf{y}}_{t+T_f}]$. Specifically, the decoder takes the vehicle's current state as the initial input to generate the first-step behavior. In every following step, the decoder takes the output value of the last step as the input to generate a new output. Mathematically, the relationship among encoder, decoder, and context vector can be summarized:

$$c_t = f_{enc}(\mathbf{S}_{t-T_h,t}; \theta^E), \tag{18}$$

$$\hat{\mathbf{Y}}_{t+1,t+T_f} = f_{dec}(c_t, s_t; \theta^D), \tag{19}$$

where the context vector $c_t$ is the last hidden state of the encoder and is also used as the decoder's initial hidden state; the current state $\mathbf{s}_t$ is fed as the decoder's first-step input. In this paper, we use a single-layer GRU and stack three dense layers on the decoder for stronger decoding capability.

The goal of TEDNis to minimize the error between the ground-truth trajectory and the generated trajectory. Taking a deterministic approach, the loss function is simply designed to be:

$$\mathcal{L} = \sum_{i=0}^{N} ||\hat{\mathbf{Y}}_{t+1,t+T_f}^{i} - \mathbf{Y}_{t+1,t+T_f}^{i}||_p, \tag{20}$$

where $N$ denotes the number of training trajectory samples. The objective can be measured in any $l_p$ norm, while in this paper we consider $l_2$ norm.

## 5.2 Integrating the intention signal

The TEDNcan be regarded as a motion generator given the historic dynamics, while the encoding for interaction and map information is left to the high-level intention policy to handle. Such a hierarchical policy simplifies the learning burden for each sub-policy and offers better interpretability. However, it remains unknown *what* intention signals should be considered and *how* to integrate them into the low-level policy.

In our case, we aim at generating high-fidelity human-like predictions in a certain future horizon. So we naturally expect the intention signals to include the goal state $g_t$ in the future horizon to guide the TEDN's generation process. Besides, considering the fact that the same GRU cell is recursively utilized at each step, the GRU cell is unaware of whether the current decoding lies in the earlier horizon or the later horizon. Consequently, we introduce the current decoding step $t'$as another intention signal to help the decoder to better track the goal state $g_t$. Introducing these intention signals would then modify the decoder definition from Eq (19):

$$\hat{\mathbf{Y}}_{t+1,t+T_f} = f_{dec}(c_t, s_t, g_t, t'; \theta^D). \tag{21}$$

There are various ways to incorporate the intention signal into the time series model. In general, when the additional feature is a temporal series, it is intuitive to append it to the end of the original input feature vector or output vector of the GRU (before the dense layers) as in Cheng et al. (2020; 2019). However, when we have a non-temporal-series additional feature, directly appending it to the original feature vector may create harder learning by polluting the temporal structure. A more delicate approach is to embed the

additional feature with a dense layer and add it to the hidden state of RNN at the first-step decoding, so that the non-temporal signal is passed in the GRU cell state along the decoding sequence as in Karpathy & Fei-Fei (2015); Vinyals et al. (2015). Besides, in our case, the goal state intention signal is defined as the goal position in the physical world, so another approach is to directly transform the original input state to the state relative to the goal state, such that the model is implicitly told to reach origin at the last step of decoding.

Besides how to incorporate the intention signal, it also remains unclear what coordinate, features, and representation should be employed for the best performance? Thus we conducted systematic experiments to evaluate the effect of these factors. A brief summary of these experiments is provided in Sec 7.4, while detailed evaluations can be found in Appendix B C. We empirically found that in the test data distribution, the best performance goes: 1) in Frenet coordinate, 2) including input features like velocity and yaw, 3) applying representation trick like incremental prediction and position alignment, 4) appending intention signal like goal state and decoding step into the input feature.

# 6 Online adaptation

In this section, we introduce the motivation, formulation and the utilized algorithm ($\text{MEKF}_\lambda$) of online adaptation module. Possible design choices are also discussed at the end of the section.

Though humans are usually assumed to be rational, the standards of "optimal plan" may still vary across different agents or circumstances (Baker et al., 2006; 2007). Consequently, human behaviors are naturally heterogeneous, stochastic, and time-varying. Different driving scenarios also inevitably create additional behavior shifts. We thus utilize online adaptation to inject customized individual and scenario patterns into the model. The key insight for online adaptation is that, though drivers cannot communicate directly, their historic behaviors can be a vital clue for their driving patterns, based on which we can adapt parameters of our model to better fit the individual or scenario.

## 6.1 Multi-step feedback adaptation formulation

The goal of online adaptation is to improve the quality of behavior prediction with feedback from the historic ground-truth information. In our case, the policy is hierarchically divided, with two sub-policies to be adapted. Due to the large delay in obtaining the long-term ground-truth intention label, we only consider the online adaptation for the low-level behavior-prediction policy, while keeping the high-level intention-identification policy intact. The intuition behind the online adaptation is thus that, though given the same goal state, drivers still have diverse ways to achieve it. Capturing such customized patterns can improve the human-likeness of generated behavior.

Formally, at time step $t$, online adaptation aims at exploiting local over-fitting in the historic observation to improve individual behavior prediction quality:

$$\min_\theta ||\hat{\mathbf{Y}}_{t-\tau,t} - \mathbf{Y}_{t-\tau,t}||_p, \tag{22}$$

where $\mathbf{Y}_{t-\tau,t}$ is the ground-truth observed trajectory in the historic $\tau$ steps; $\hat{\mathbf{Y}}_{t-\tau,t}$ is the predicted trajectory by the TEDN with the model parameter $\theta$ in the historic $\tau$ steps. Assume that the model parameter changes slowly, namely $\dot{\theta} \approx 0$. Then the model parameter that generates the best predictions in the future can be approximated by the model parameter that best fits the historic ground-truth observation. Also note that online adaptation can be iteratively executed for one agent once a new observation is received.

Practically, since online adaptation can be conducted as soon as at least one-step new observation is available, the length of ground-truth observation $\tau$ dose not necessarily have to match the behavior generation horizon $T_f$. Thus the online adaptation is indeed a multistep feedback strategy (Abuduweili & Liu, 2021). By definition, at time step $t$, we have the recent $\tau$ step ground-truth observation $\mathbf{Y}_{t-\tau,t} = [y_{t-\tau+1}, y_{t-\tau+2}, ..., y_t]$. From the memory buffer we also have the generated behavior at $t - \tau$ steps earlier $\hat{\mathbf{Y}}_{t-\tau,t} = [\hat{y}_{t-\tau+1}, \hat{y}_{t-\tau+2}, ..., \hat{y}_t]$. Then the model parameter can be adapted based on the recent $\tau$-step error:

$$\hat{\theta}_t = f_{adapt}(\hat{\theta}_{t-1}, \hat{\mathbf{Y}}_{t-\tau,t}, \mathbf{Y}_{t-\tau,t}), \tag{23}$$

---

**Algorithm 1** $\tau$ step online adaptation with MEKF$_\lambda$

---

**Input:** Offline trained TEDNnetwork with parameter $\theta$, initial variance $\mathbf{Q}_t$ and $\mathbf{R}_t$ for measurement noise and process noise respectively, forgetting factor $\lambda$

**Output:** A sequence of predicted future behavior $\{\hat{\mathbf{Y}}_{t+1,t+T_f}\}_{t=1}^T$

1: **for** $t = 1, 2, ..., T$ **do**
2:      **if** $t \geq \tau$ **then**
3:          stack recent $\tau$-step observations:
4:              $\mathbf{Y}_{t-\tau,t} = [y_{t-\tau+1}, ..., y_t]$
5:          stack recent $\tau$-step generated trajectory:
6:              $\hat{\mathbf{Y}}_{t-\tau,t} = [\hat{y}_{t-\tau+1}, ..., \hat{y}_t]$
7:          adapt model parameter via MEKF$_\lambda$:
8:              $\hat{\theta}_t = f_{MEKF_\lambda}(\theta_{t-1}, \hat{\mathbf{Y}}_{t-\tau,t}, \mathbf{Y}_{t-\tau,t})$
9:      **else**
10:          initialization: $\hat{\theta}_t = \theta$
11:      **end if**
12:      collect goal state $g_t$ from SGN and input features $\mathbf{S}_{t-T_h}$.
13:      generate future behavior:
14:          $\hat{\mathbf{Y}}_{t+1,t+T_f} = [\hat{y}_{t+1}, ... \hat{y}_{t+T_f}] = f_{TEDN}(\mathbf{S}_{t-T_h}, g_t, \hat{\theta}_t)$.
15: **end for**
16: **return** a sequence of predicted future behavior $\{\hat{\mathbf{Y}}_{t+1,t+T_f}\}_{t=1}^T$

---

where $f_{adapt}$ denotes the adaptation algorithm to be discussed in detail in Sec 6.2. The adapted model is then used to generate behaviors in the future $T_f$ steps from the current time $t$. It is worth noting that here only $\tau$-steps errors are utilized and we expect better performance in $T_f$ steps. Nevertheless, behavior prediction error usually grows exponentially as the horizon extends. When $\tau$ is too small, we may not obtain enough information for the online adaptation to benefit behavior prediction in the whole future $T_f$ step horizon. Intuitively, the problem can be mitigated by using errors of more steps. However, there exists a $\tau$-step time lag in $\tau$-step adaptation strategy. Too many steps may also create a big gap between historic behavior and current behavior, so that the model adapted at an earlier time may be outdated and incapable of tracking the current behavior pattern. Thus there is indeed a trade-off between obtaining more information and maintaining behavior continuity when we increase observation steps $\tau$. It also remains theoretically unknown which layer is the best to adapt. Consequently, we propose a new set of metrics shown in Figure 4 to investigate such a trade-off in detail:

1. ADE 1: This metric evaluates the prediction error of the adapted steps on the historic trajectory $Y_{t-\tau,t}$. Because these steps are the observation source used to conduct online adaptation, the metric can verify whether the algorithm is working or not.

2. ADE 2: This metric evaluates the prediction error of the adaptation steps on the current trajectory, which aims at verifying how the time lag is influencing the adaptation. Also, this method can be used to verify whether adaptation could improve short-term behavior prediction.

3. ADE 3: This metric evaluates the prediction error of the whole historic trajectory, which shows if we have gotten enough information on the behavior pattern.

4. ADE 4: This metric evaluates the prediction performance of the whole current trajectory, which shows whether or not the adaptation based on historic information can help current long-term behavior prediction.

The effect of utilizing different steps' observation is briefly discussed in 7.4 and analysed in detail in Appendix D, where we empirically found that in the tested data distribution, the best performance goes when 2 or 3 steps of observation are used.

## 6.2   Robust nonlinear adaptation algorithms

There are many online adaptation approaches, such as stochastic gradient descent (SGD) (Bhasin et al., 2012), recursive least square parameter adaptation algorithm (RLS-PAA) (Ljung & Priouret, 1991). In this paper, we choose the modified extended Kalman filter with forgetting factors (MEKF$_\lambda$) (Abuduweili & Liu, 2021) as the adaptation algorithm due to its robustness to data noises and efficient use of second-order information. Compared to the previous work (Abuduweili & Liu, 2021) which considers predicting the human motion (wrist trajectory), we use the method for driving behavior prediction in dense and interactive traffic scenarios, a more complex problem.

The MEKF$_\lambda$ regards the adaptation of a neural network as a parameter estimation process of a nonlinear system with noise:

$$\mathbf{Y}_{t+1,t+T_f} = f_{TEDN}(\hat{\theta}_t, \mathbf{S}_{t-T_h,t}) + \mathbf{u}_t \qquad (24)$$

$$\hat{\theta}_t = \hat{\theta}_{t-1} + \omega_t \qquad (25)$$

where $\mathbf{Y}_{t+1,t+T_f}$ is the observation of the ground-truth trajectory; $f_{TEDN}(\hat{\theta}_t, \mathbf{S}_t) = \hat{\mathbf{Y}}_{t+1,t+T_f}$ is the generated behavior by the TEDN policy $f_{TEDN}$ with the input $\mathbf{S}_t$ at time step $t$; $\hat{\theta}_t$ is the estimate of the model parameter of the TEDN; the measurement noise $u_t \sim \mathcal{N}(0, \mathbf{R}_t)$ and the process noise $\omega_t \sim \mathcal{N}(0, \mathbf{Q}_t)$ are assumed to be Gaussian with zero mean and white noise. Since the correlation among noises are unknown, it is reasonable to as-

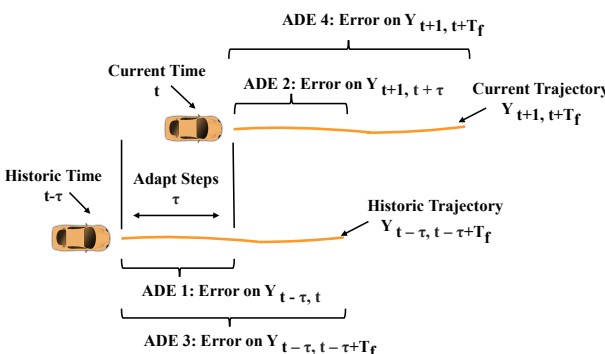

Figure 4: Figure updated.An illustration of online adaptation and 4 metrics for performance analysis. At time step $t$, the model parameters can be adapted by minimizing prediction error of the trajectory in past $\tau$ steps $\mathbf{Y}_{t-\tau,t}$. The adapted parameter is then used to generate prediction $\mathbf{Y}_t$ on current time $t$. A new set of 4 metrics is proposed to analyze the adaptation performance. ADE 1 can verify how the adaptation works on the source trajectory $\mathbf{Y}_{t-\tau,t}$. ADE 2 can verify whether adaptation can benefit short-term prediction in the presence of behavior gap between earlier time and current time. ADE 3 can verify whether we have obtained enough information from the source trajectory $\mathbf{Y}_{t-\tau,t}$. ADE 4 verifies whether adaptation can benefit the long-term prediction in the current time.

sume noises are identical and independent of each other. For simplicity, we assume $\mathbf{Q}_t = \sigma_q \mathbf{I}$ and $\mathbf{R}_t = \sigma_r \mathbf{I}$ where $\sigma_q > 0$ and $\sigma_r > 0$. Applying MEKF$_\lambda$ on the above dynamic equations, we obtain the following equations to update the estimate of the model parameter based on the prediction error in the historic $\tau$ time steps:

$$\hat{\theta}_t = \hat{\theta}_{t-1} + \mathbf{K}_t \cdot (\mathbf{Y}_{t-\tau,t} - \hat{\mathbf{Y}}_{t-\tau,t}) \qquad (26)$$

$$\mathbf{K}_t = \mathbf{P}_{t-1} \cdot \mathbf{H}_t^T \cdot (\mathbf{H}_t \cdot \mathbf{P}_{t-1} \cdot \mathbf{H}_t^T + \mathbf{R}_t)^{-1} \qquad (27)$$

$$\mathbf{P}_t = \lambda^{-1}(\mathbf{P}_t - \mathbf{K}_t \cdot \mathbf{H}_t \cdot \mathbf{P}_{t-1} + \mathbf{Q}_t) \qquad (28)$$

where $\mathbf{K}_t$ is the Kalman gain. $\mathbf{P}_t$ is a matrix representing the uncertainty in the estimates of the parameter $\theta$ of the model; $\lambda$ is the forgetting factor to discount old measurements; $\mathbf{H}_t$ is the gradient matrix by linearizing the network:

$$\mathbf{H}_t = \frac{\partial f_{TEDN}(\hat{\theta}_{t-1}, \mathbf{S}_{t-1,t-1-T_h})}{\partial \hat{\theta}_{t-1}} = \frac{\partial \hat{\mathbf{Y}}_{t-1,,t-1+T_f}}{\partial \hat{\theta}_{t-1}} \qquad (29)$$

In implementation, we need to specify initial conditions $\theta_0$ and $\mathbf{P}_0$. $\theta_0$ is initialized as the offline trained model parameter. For $\mathbf{P}_0$, due to absence of prior knowledge on the initial model parameter uncertainty, we simply set it as an identity matrix $\mathbf{P}_0 = p_i \mathbf{I}$ with $p_i > 0$. The whole process of the online adaptation is summarized in Algorithm 1, which enables us to adapt the parameter of any layer in the model. The performance of adapting different layers is briefly discussed in Sec 7.4 and analysed in detail in Appendix D, where we empirically found that in the tested data distribution, the best performance goes when the last layer of the network is adapted.

# 7 Experiment

By detailed experiments on real data, we aim at answering the following key questions:

1. In the high-level intention identification policy (SGN), what features in the input and output should be considered? What graph network architecture works the best?

2. In the low-level behavior prediction policy (TEDN), what coordinates and features in the input and output work the best? Whether commonly used representation tricks and mechanisms in the encoder decoder architecture would improve performance?

3. How to integrate the intention signal into the TEDN? How much can the intention signal improve the prediction accuracy?

4. How to systematically evaluate the performance of online adaptation (OA)? How many steps of observation are the best to adapt? What is the best layer in the network to adapt?

5. How does the whole proposed method perform compared to other methods in terms of prediction accuracy, transferability and adaptability?

In this section, we first provide a case study illustrating how our method works. To answer the first four questions above, we then provide an evaluation on the SGN, a brief summary on the evaluations TEDNand OA, and a comparison with other methods for the 5-th question. For detailed evaluations and analysis on TEDNand OA, we refer interested readers to the Appendix B C D respectively.

## 7.1 Experiment setting

**Dataset** We verified our proposed method with real human driving data from the INTERACTION dataset (Zhan et al., 2019), the InD dataset (Bock et al., 2020), and the Argoverse 1 dataset(Chang et al., 2019). Two scenarios from the INTERACTION dataset were utilized: a 5-way unsignalized intersection (Figure 6) and an 8-way roundabout (Figure 7). The intersection scenario was used to train our policy and evaluate the performance of behavior prediction. The roundabout scenario was used to evaluate the transferability of our method. In the intersection scenario, we had 19084 data points, which were split to 80% for training data and 20% for validation data. In the roundabout scenario, there were 9711 data points. Another 4-way unsignalized intersection scenario from the InD Dataset is also included to verify transferability, which consists of 12837 data points. Adaptability is evaluated on all above scenarios. Besides, Argoverse dataset is also used to compare our method with prior methods, and we considered 9213 data points, where 80% come from the training set and 20% come for the validation set. Road reference paths and traffic regulations were extracted from the provided high-definition map. The online adaptation was conducted on the INTERACTION and InD dataset with access to multiple trajectory segments of one agent, and online adaptation was not conducted on the Argoverse dataset as only on trajectory segment is provided for one vehicle.

**Implementation Details** In our experiments, we choose the historic time steps $T_h$ as 10 and future time step $T_f$ as 30, which means we utilized historic information in the past 1 second to generate future behavior in the next 3 seconds. In addition to the long-term behavior prediction evaluation in the whole future 30 steps, we also evaluated short-term behavior prediction in the future 3 steps, as short-term behavior is safety-critical especially in close-distance interactions. The method was implemented in Pytorch on a desktop computer with an Intel Core i7 9th Gen CPU and a NVIDIA RTX 2060 GPU. For each model, we performed optimizations with Adam and sweep over more than 20 combinations of hyperparameters to select the best one including batch size, hidden dimension, learning rate, dropout rate, etc. We trained each method with three different seeds and displayed the best performance. The distributional metrics $\pm$ is used to denote the standard deviations of the prediction errors over the data.

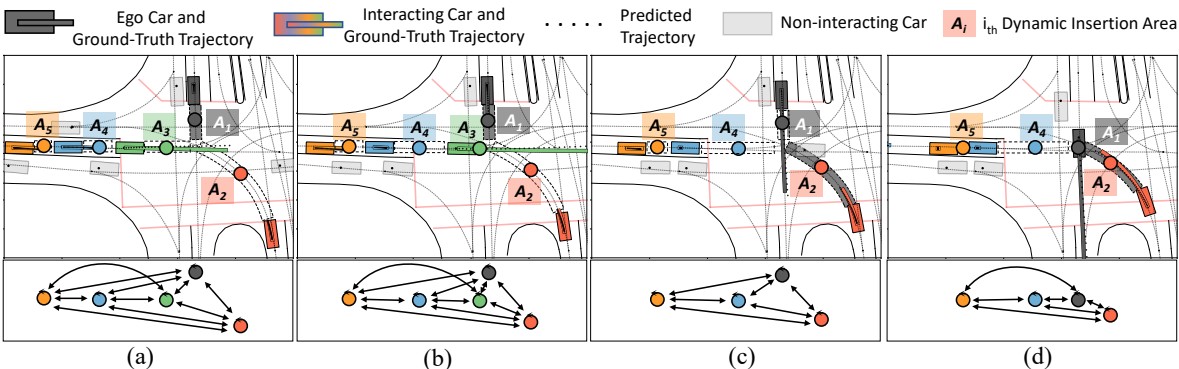

Figure 5: Figure updated.Case 1 - an illustration of how our method works during one interaction. Our methods can predict which DIA ego car will insert into, and the future trajectories of ego car and its interacting cars. Black car denotes the ego car; bright-color cars denote vehicles interacting with ego car, based on which the DIAs are extracted; transparent gray cars denote non-interacting vehicles.Each DIA is marked with dashed-line box and one node. The same color is used for one DIA's node, notation, and the DIA's rear-bound vehicle. The graphical relationships at one scene is displayed underneath each scene. The darker the DIA is, the more likely the ego car is going to insert into that area. The ground-truth and predicted most likely future trajectory of the ego vehicle and its interacting vehicles are displayed. In this case, DIA $A_1$ denotes the slot between ego car and the conflict point; DIA $A_2$ denotes the slot between orange car and the conflict point; DIA $A_3$ denotes the slot between green car and the conflict point; DIA $A_4$ denotes the slot between the blue car and green car; DIA $A_5$ denotes the slot between yellow car and blue car. During this interaction, the ego car first yielded the green car in (a)(b), and then passed before other cars (c)(d).

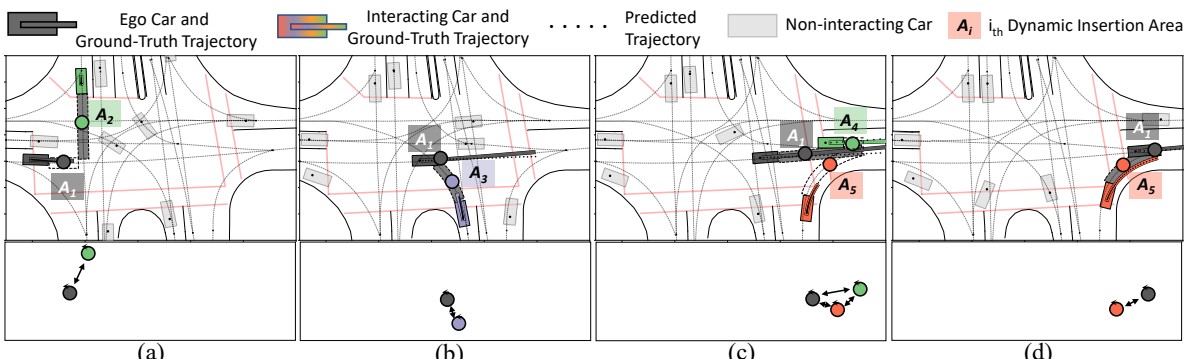

Figure 6: Figure updated.Case 2 - an illustration of how our method works during a sequence of interactions. When ego car crosses the whole scene, our method can constantly identify interactions and extract interacting cars. The ego car first interacted with the green car in (a), then with the purple car in (b), then the green and orange car in (c)(d).

## 7.2 Case study

We first illustrate how our method works with three examples in Figure 5-7: 1) how ego vehicle interacted with other vehicles to pass a common conflict point (one interaction); 2) how the ego vehicle interacted with other vehicles to pass a sequence of conflict points (a sequence of interactions); 3) how ego vehicle interacted with other vehicles when it is zero-shot transferred to the roundabout scenario without retraining (scenario-transferable interactions).

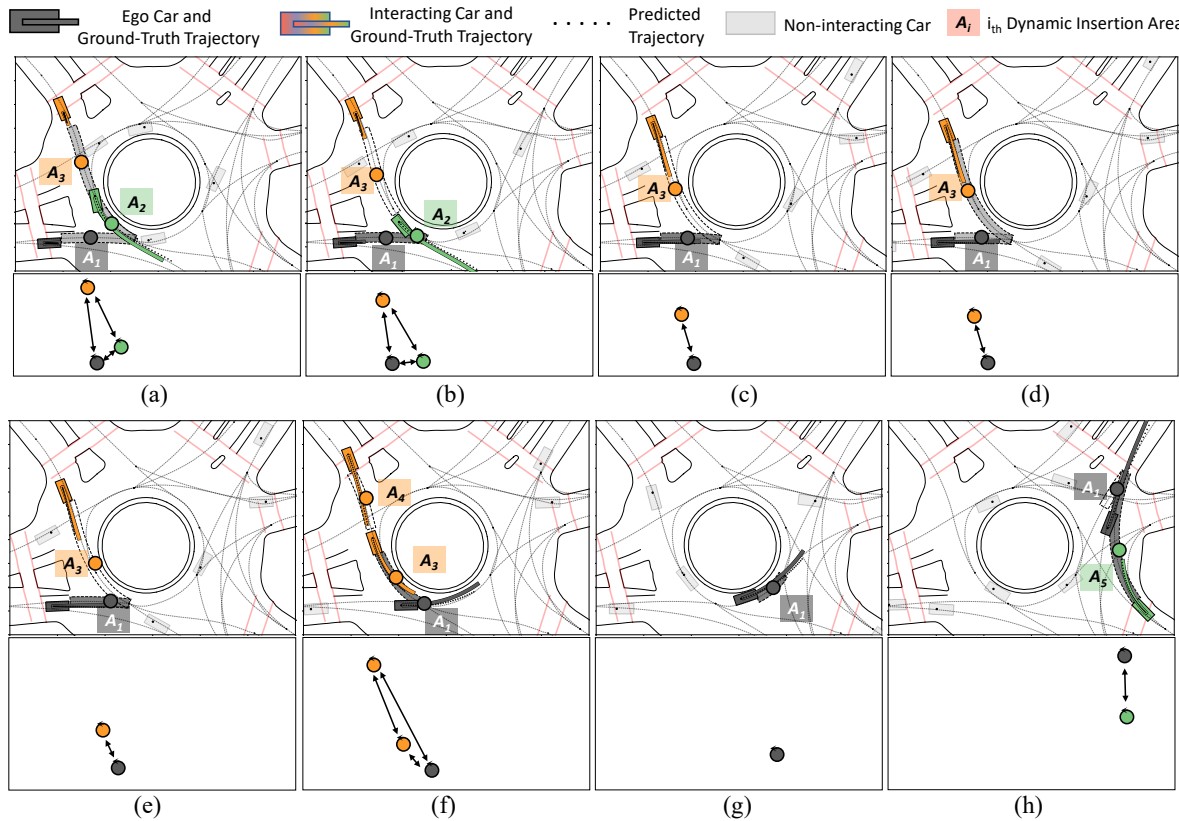

Figure 7: Figure updated.Case 3 - an illustration of how our method is directly transferred to the roundabout scenario without learning a new set of parameters, after it is trained in the intersection scenario. The ego car first yielded the green car in (a)(b), then passed before the yellow cars (c)(d)(e)(f). Note that humans' hesitating and intention switch in this period was captured by our method, where the ego vehicle first decided to yield the yellow car in (c), then changed it mind in (d), and finally passed before the yellow car in (e)(f). The ego car then continued to run in (g), and left the roundabout before the green car in (h).

### 7.2.1 Case 1: one interaction

As in Figure 5, we first show how our method works in one interaction. Once we chose the ego vehicle, we extracted cars whose reference lines conflict with the ego car's reference line. These cars were regarded as the interacting vehicles and corresponding DIAs were extracted. Our method would then predict the intention and future trajectory of the ego vehicle and its interacting vehicles. In Figure 5, we drew the ego vehicle with black color, the interacting vehicles with bright colors, and the non-interacting vehicles with transparent gray color. The darker (less transparent) a DIA is, the more likely the ego vehicle would insert into that DIA. The ground-truth and predicted future trajectories of the ego vehicle and the interacting vehicles are also displayed.

As in Figure 5(a)(b), the black ego car initially had 5 areas to choose to insert into: $A_1, A_2, A_3, A_4, A_5$. The ego vehicle braked so our method predicted that it would insert into its front DIA $A_1$, which means yielding to other vehicles. In Figure 5(c), the green vehicle behind DIA $A_3$ droveaway and the ego vehicle accelerated, so our method predicted ego vehicle would insert into DIA $A_2$, which means passing before other vehicles. In Figure 5 (d) the ego vehicle crossed the conflict point and finished this interaction.

### 7.2.2 Case 2: A sequence of interactions

We illustrate how one vehicle crossed the intersection with a sequence of interactions in Figure 6. Specifically, the ego vehicle initially interacted with the upper green vehicle as in Figure 6(a). As the black ego vehicle

was running at a high speed, our method predicted it would insert into DIA $A_2$ and pass before the green vehicle. After finishing the first interaction, the ego vehicle then interacted with the purple vehicle below as in Figure 6(b). Our method predicted the ego vehicle would continue to pass and insert into the DIA $A_3$. Later in Figure 6(c), the ego vehicle's reference path conflicted with that of the green and orange car. The ego vehicle first decelerated and our method predicted it would insert into its front DIA $A_1$ and yield other cars. In Figure 6(d), after the green car droveaway, the ego vehicle then accelerated and inserted into DIA $A_5$ to pass ahead of the orange car.

### 7.2.3 Case 3: Scenario-transferable interactions

In Figure 7, we show that after our policy was trained in the intersection scenario, it can be zero-shot transferred to the roundabout scenario. In Figure 7(a), the ego vehicle just entered the roundabout with some speed, so it was predicted to be equally likely to insert into the three DIAs $A_1, A_2, A_3$. In Figure 7(b), the ego vehicle decelerated, so it was predicted to yield other vehicles and insert into its front DIA $A_1$. In Figure 7(c), the green car moved away but the ego vehicle still remained at a low speed, leading to the prediction that the ego car would continue to yield other vehicles. But later we witnessed a change of plan. In Figure 7(d), the ego vehicle accelerated so it became equally likely to insert into either DIA $A_1$ or DIA $A_3$. In Figure 7(e) the ego vehicle gained a high speed, and it was predicted to pass before the yellow car by inserting into DIA $A_3$. After finishing the first interaction in Figure 7(f), the ego vehicle continued to run while there were no other interacting vehicles as in Figure 7(g). In Figure 7(h), one green car entered the roundabout and the ego vehicle decided to pass before it by inserting into DIA $A_5$

### 7.3 Ablation studies on the high-level policy (SGN)

In the high-level intention-identification task, we evaluated how well out method is able to predict future intentions by comparing the performance of our semantic graph network (SGN) with that of the other six approaches/variants. Three of them are set to explore the effect of different features and representations in the input and output. And the rest of them are the variants of the proposed network, which explored the effect of frequently used network architectures and tricks.

1. No-Temporal: This method does not take historic information into account, namely only considering the information of the current time step $t$.

2. GAT: This method uses the absolute feature to calculate relationships among nodes instead of using the relative feature. This method corresponds to the original graph attention network (Veličković et al., 2017).

3. Single-Agent: This method only considers the loss of ego vehicle's intention prediction, and does not consider the intention prediction for other interactive vehicles.

4. No-Class-Loss: This method only considers the regress loss for goal state prediction ($\mathcal{L}_{regress}$), and does not consider the class loss for insertion area prediction ($\mathcal{L}_{class}$). Refer to Eq. 17 for detailed definition of the two loss terms. This method is designed to evaluate the effectiveness of incorporating insertion area prediction as a auxiliary task for goal point prediction.

5. Goal-Pt-Sample: This method samples in the predicted goal point distribution to get the goal point, in contrast to our method where we choose the mean of the distribution as the goal point.

6. Two-Layer-Graph: This method has a two-layer graph to conduct information embedding, namely exploits the graph aggregations twice (Sanchez-Gonzalez et al., 2018).

7. Multi-Head: This method employs the multi-head attention mechanism to stabilize learning (Veličković et al., 2017), namely operating the relationship reasoning multiple times in parallel independently, and concatenates all features as the final aggregated feature. In our case, we set the head number as 3.

Table 2: The statistical evaluation of the high-level intention prediction policy on two scenarios from the INTERACTION dataset. We compared our method with the other six approaches/variants to explore the effect of different representations, features, and architectures. Note that all the models are trained in the intersection scenario and then directly transferred and tested on the roundabout scenario without further training. Our method outperformed all other methods in the goal state prediction (ADE).

| Scenario | Measure | Representation/Loss Ablation Study | | | | | Architecture Ablation Study | | | Ours |
|---|---|---|---|---|---|---|---|---|---|---|
| | | No-Temporal | GAT | Single-Agent | No-Class-Loss | Goal Pt Sample | Two-Layer-Graph | Multi-Head | Seq-Graph | |
| Intersection | Acc (%) | 87.44 ± 33.13 | **91.93 ± 27.05** | 88.8 ± 31.53 | 75.64 ± 42.92 | 90.50 ± 28.70 | 90.15 ± 29.79 | 90.00 ± 30.00 | 89.8 ± 30.24 | 90.50 ± 28.70 |
| | ADE (m) | 1.59 ± 1.67 | 1.18 ± 1.51 | 1.04 ± 0.90 | 1.00 ± 0.76 | 1.31 ± 1.462 | 0.98 ± 0.93 | 0.97 ± 0.75 | 1.33 ± 1.91 | **0.94 ± 0.73** |
| Roundabout (Transfer) | Acc (%) | **93.92 ± 23.88** | 91.21 ± 28.12 | 92.20 ± 26.75 | 83.53 ± 37.08 | 90.70 ± 29.08 | 90.54 ± 29.26 | 92.10 ± 26.96 | 91.60 ± 27.62 | 90.70 ± 29.08 |
| | ADE (m) | 3.62 ± 6.72 | 2.70 ± 5.16 | 1.88 ± 2.48 | 1.79 ± 1.62 | 1.97 ± 2.70 | 1.87 ± 2.51 | 2.79 ± 20.78 | 3.10 ± 3.10 | **1.70 ± 1.99** |

8. Seq-Graph: This method first conducts relationship reasoning for the graph at each time step and then feeds the sequence of aggregated graphs into RNN for temporal processing. As a comparison, our method first embeds each node's sequence of historic features with RNN and then conduct relationship reasoning using each node's hidden state from RNN at the current time step.

The models were trained and tested on the intersection scenario from the INTERACTION dataset. The trained models were also directly tested on the roundabout scenario from the INTERACTION dataset to evaluate the zero-shot transferability. The inserted area prediction accuracy was evaluated by the multi-class classification accuracy. The performance of goal state prediction was evaluated by the absolute distance error (ADE) between the generated goal state and ground-truth state.

### 7.3.1 Inserted area prediction accuracy

On the inserted area prediction accuracy shown in Table 2, we first provide some overview analysis. Except for No-Class-Loss which does not optimize the inserted area prediction,all other models achieved close accuracy of around 90%, generally benefiting from the representation of the semantic graph. Another overall observation is that most models' transferability performance in the roundabout scenario surprisingly surpassed the performance on the intersection scenario on which the models were originally trained. This is because the intersection is a harder scenario than the roundabout, as the vehicles need to interact with many vehicles from different directions simultaneously when they are entering the intersection, while vehicles in the roundabout only need to interact with the cars from nearby branches.

We then show some detailed analysis. The No-Class-Loss method had the lowest accuracy and largest variance on both the intersection and the roundabout scenario, since it does not optimize the classification loss for area insertion prediction.The No-Temporal method performed badly in the intersection scenario (87.44±22.13%). This is because it lacks temporal information, which could otherwise efficiently help to identify which DIA to insert into by considering historic speed and acceleration. What is interesting is that the No-Temporal method contrarily achieved the highest insertion accuracy (93.92±23.88) in the roundabout scenario. One possible explanation is that, and the absence of temporal information constrained the model's capability and thus avoided over-fit, so the No-Temporal method has the best transferability performance. Such hypothesis also helps to explain: 1) why the Two-Layer-Graph method had the lowest insertion accuracy in the roundabout scenario (90.54±29.26%), as twice aggregations make the model brittle to over-specification; 2) why the GAT had the highest performance while our method followed as the second - since the inserted area prediction is a relatively simple classification task, while absolute features may be enough for the model to learn, additional relative features may provide redundant information and make the learning more difficult.

### 7.3.2 Goal state prediction error

The goal state prediction is main task as it is directly delivered to low-level policy as the intention signal to guide the behavior generation process. On the other hand, it is much harder than the insertion area prediction task as it requires more delicate information extraction and inference. Consequently, we can see that the performance of different models varied a lot as shown in Table 2. Also, the models' performance significantly

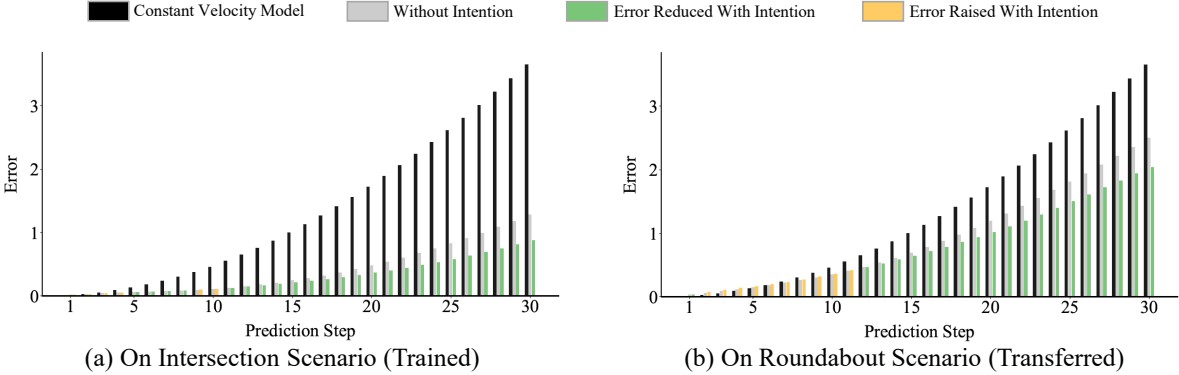

Figure 8: Figure updatedPrediction error (m) of each step on two scenarios of the INTERACTION dataset. The prediction error grew exponentially as horizon extends. The intention signal effectively suppressed the error growth, especially in the long term. The model is trained and validated on the intersection scenario, and is then directly transferred to the roundabout scenario.

down-graded when they were directly transferred to the roundabout scenario, as the two scenarios have different geometries and different driving patterns such as speed and steering.

Specifically, we have several observations: 1) our method achieved the lowest error in both intersection and roundabout scenarios; 2) the No-Temporal method was the worst in both intersection and roundabout scenarios, due to the lack of temporal information; 3) the GAT method generated much higher errors than our method especially in the roundabout scenario (58%), which shows the necessity of using the relative features in relationship reasoning; 4) our method outperformed the Single-Agent method, which implies the advantages of data augmentation and encouraging interaction inference by taking all vehicles' generated goal state into the loss function. 5) our method outperformed the No-Class-Loss method, indicating that incorporating the area prediction task as an auxiliary task can indeed help with the goal point prediction;6) our method outperformed the Goal-Pt-Sample method, indicating the advantage of choosing the mean of the predicted distribution as the goal point for trajectory prediction.7) The Two-Layer-Graph method was the closest one to our method, though it came with serious over-fitting, for which we conducted more hyperparameter tuning to find a proper dropout value; 8) The Multi-Head method achieved the second-best accuracy in the intersection scenario but much worse performance in the roundabout scenario, which could be possibly improved by more delicate searching for a proper head number; 9) The Seq-Graph method was the second-worst in both the intersection and the roundabout scenario, which may imply that the complex encoding for past interactions could hardly help prediction but indeed makes the learning harder.

With the results above, a summary of conclusions on the intention prediction policy is that it is necessary to consider temporal information, use the relative feature for relationship reasoning, predict the intention of all agents in the scene, and incorporate the area prediction as an auxiliary task. On the contrary, the architectures like Two-Layer-Graph, Multi-Head mechanism, and Seq-Graph do not help here. .

### 7.4 A summary of ablation studies on the low-level policy (END) and the online adaptation (OA)

AS shown in Table 3, we briefly summarize all ablation studies conducted to evaluate the low-level policy (TEDN) and the online adaptation as a quick takeaway for the reader. A detailed version of these ablation studies can be found in Appendix B C D.The model is trained and validated on the intersection scenario from the INTERACTION dataset, and is then directly transfered to a roundabout scenario (INTERACTION dataset) and one intersection scenario (InD dataset).

**Evaluation on TEDN** Starting from a baseline TEDNwhich takes in the historic position and predicts future trajectory, we first explored the effect of coordinate, features, and representations in TEDN. By transforming the coordinate from Cartesian to Frenet, the ADE in 3 seconds is reduced by 42%, 76%, and

Table 3: A brief summary of ablation studies conducted to develop our method. We incrementally investigated: the effect of the coordinate, feature, and representation in TEDN; the benefit of intention signal to prediction; the performance of online adaptation. We evaluated the performance by calculating ADE (m) and FDE (m) between predicted trajectory and ground-truth trajectory under different horizons. The model is trained and validated on the intersection scenario from the INTERACTION dataset,, and is then directly transfered to a roundabout scenario (INTERACTION dataset) and one intersection scenario (InD dataset).

| Scenario Dataset | Metric | Horizon | Constant Velocity Model | TEDNExploration | | | | Intention Integration | | Online Adaptation |
|---|---|---|---|---|---|---|---|---|---|---|
| | | | | Baseline | Coordinate | Feature | Representation | Goal State Signal | Time Signal | Ours |
| Intersection INTERACTION | ADE | 3s | $1.279 \pm 1.123$ | $1.525 \pm 1.189$ | $0.884 \pm 0.594$ | $0.629 \pm 0.397$ | $0.407 \pm 0.328$ | $0.302 \pm 0.251$ | $0.305 \pm 0.253$ | $\mathbf{0.301 \pm 0.250}$ |
| | | 0.3s | $0.016 \pm 0.022$ | $0.346 \pm 0.268$ | $0.409 \pm 0.245$ | $0.319 \pm 0.224$ | $0.027 \pm 0.020$ | $\mathbf{0.021 \pm 0.017}$ | $0.029 \pm 0.020$ | $0.023 \pm 0.014$ |
| | FDE | 3s | $3.591 \pm 3.157$ | $2.960 \pm 2.708$ | $1.850 \pm 1.684$ | $1.416 \pm 1.175$ | $1.279 \pm 1.130$ | $0.890 \pm 0.831$ | $\mathbf{0.876 \pm 0.835}$ | $0.877 \pm 0.830$ |
| | | 0.3s | $0.032 \pm 0.039$ | $0.419 \pm 0.338$ | $0.423 \pm 0.245$ | $0.324 \pm 0.229$ | $0.040 \pm 0.034$ | $0.036 \pm 0.025$ | $0.043 \pm 0.032$ | $\mathbf{0.032 \pm 0.024}$ |
| Roundabout INTERACTION (Transfer) | ADE | 3s | $1.508 \pm 1.157$ | $12.315 \pm 5.961$ | $2.924 \pm 4.695$ | $2.201 \pm 3.999$ | $0.941 \pm 0.778$ | $0.845 \pm 0.564$ | $0.815 \pm 0.526$ | $\mathbf{0.815 \pm 0.526}$ |
| | | 0.3s | $0.022 \pm 0.018$ | $6.164 \pm 5.258$ | $1.572 \pm 4.029$ | $1.543 \pm 3.957$ | $0.062 \pm 0.071$ | $0.060 \pm 0.060$ | $0.073 \pm 0.065$ | $\mathbf{0.052 \pm 0.068}$ |
| | FDE | 3s | $4.230 \pm 3.203$ | $19.262 \pm 7.761$ | $5.446 \pm 7.157$ | $4.123 \pm 5.227$ | $2.494 \pm 2.070$ | $2.081 \pm 1.440$ | $2.038 \pm 1.409$ | $\mathbf{2.041 \pm 1.409}$ |
| | | 0.3s | $0.044 \pm 0.035$ | $6.374 \pm 5.316$ | $1.546 \pm 3.894$ | $1.544 \pm 3.944$ | $0.088 \pm 0.104$ | $0.091 \pm 0.101$ | $0.108 \pm 0.107$ | $\mathbf{0.079 \pm 0.114}$ |
| Intersection InD (Transfer) | ADE | 3s | $1.959 \pm 1.248$ | $10.482 \pm 3.906$ | $2.758 \pm 3.119$ | $2.589 \pm 1.838$ | $1.264 \pm 1.264$ | $1.038 \pm 0.543$ | $1.029 \pm 0.670$ | $\mathbf{0.914 \pm 0.537}$ |
| | | 0.3s | $0.030 \pm 0.019$ | $2.554 \pm 2.612$ | $1.239 \pm 1.239$ | $1.081 \pm 0.682$ | $0.078 \pm 0.073$ | $0.119 \pm 0.119$ | $0.074 \pm 0.071$ | $\mathbf{0.057 \pm 0.052}$ |
| | FDE | 3s | $5.477 \pm 3.552$ | $19.598 \pm 8.165$ | $5.225 \pm 3.665$ | $4.986 \pm 3.700$ | $3.359 \pm 2.316$ | $2.503 \pm 1.670$ | $2.480 \pm 1.477$ | $\mathbf{2.361 \pm 1.473}$ |
| | | 0.3s | $0.058 \pm 0.038$ | $2.851 \pm 2.437$ | $1.200 \pm 0.625$ | $1.220 \pm 1.005$ | $0.107 \pm 0.107$ | $0.180 \pm 0.141$ | $0.113 \pm 0.112$ | $\mathbf{0.080 \pm 0.086}$ |

75% in the three scenarios respectively.By adding features such as speed and yaw, we reduced the ADE in 3 seconds by 28%, 24%, and 6% in the three scenarios. In the representation aspect, we conducted incremental prediction and position alignment. The position alignment trick aligns the positions of each step to the vehicle's current position (Park et al., 2018). Specifically, when considering prediction of a trajectory, the coordinate of all waypoints on the trajectory will be subtracted by the coordinate of the waypoint in the current step. The incremental prediction trick predicts the relative position compared to the position of the last step (displacement of each step), rather than directly predicting the absolute position (Li et al., 2019).Such design not only effectively reduced the ADE in 3 seconds by 35%, 57%, and 51% in the three scenarios, but also significantly reduced the ADE in 0.3 seconds by 91%, 95%, and 92% with better vehicle dynamic continuity. We can consequently conclude that the information of speed and yaw, along with the incremental prediction and position alignment, are vital for the prediction performance in encoder decoder architecture.

Next, we integrated intention signals into the TEDN. According to the Table 3, integrating the goal state signal significantly reduced the ADE in 3 seconds by 25%, 10% and 17% in the three scenarios. The time signal barely benefitted the prediction in the intersection scenario, but it reduced the ADE in 3 seconds of the roundabout scenario by 3.5%. In Figure 8, we also displayed the error by step in the future 30 prediction steps on two scenarios. We can clearly see that as the prediction horizon extended, the prediction error grew exponentially. But the intention signal effectively suppressed the error growth, especially in the long horizon. Such results effectively demonstrated the necessity of intention integration.

**Evaluation on OA** Finally, the online adaptation was implemented. As shown in Table 3, the online adaptation barely improved the long-term prediction in the next 3 seconds on the INTERACTION dataset, but improved the long-term prediction in the next 3 seconds by 11% on the InD dataset. One explanation could be that the performance of online adaptation would depend on the data distribution. When the data lies in different distribution, there will be more space for the adaptation.Besides, the short-term prediction in 0.3 seconds was improved by 20%, 28%, and 22% in the three scenarios respectively. Such improvement is valuable especially in close-distance prediction, where a small prediction shift can make a big difference in terms of safety. Besides, the improvement in long-term prediction could also provide better vehicle dynamic continuity in the predicted trajectory.

Besides, as discussed in Sec 6.1, there are also multiple design choices for the online adaptation. For example, the adaptation steps $\tau$ pose a problem of trade-off between obtaining more information and maintaining temporal behavior continuity when we increase observation steps $\tau$. It also remains unknown which layer of parameter is the best to adapt. To empirically investigate these problems, as in Figure 9, we show the percentage of short-term prediction error (ADE 2) reduced or raised after adaptation, under different adaptation step $\tau$, different adapted parameters, and two different scenarios from the INTERACTION

dataset. We have several observations: 1) as the adaptation step $\tau$ increased, the percentage of error reduction increased and reached the peak at 2 or 3 steps. But after that the help of adaptation decayed. After 7 steps, the online adaptation does not help because too many steps leads to a big gap between historic behavior and current behavior, so that the model adapted at an earlier time may be outdated and incapable of tracking the current behavior pattern; 2) Intuitively, the adaptation worked better in the roundabout scenario, compared to the intersection scenario, as the model was trained on the intersection scenario and directly transferred to the roundabout scenario. 3) The best adaptation performance is usually achieved by adapting the layer $W_3^F$, which is the last layer of the FC network in the decoder.

### 7.5 Comparison with other methods

In this section, as in Table 4, we compared our method with other methods in terms of behavior prediction accuracy, transferability, and adaptability in different horizons and scenarios. All models are trained and validated in the intersection scenario from INTERACTION dataset, and then zero-shot transferred to the unseen roundabout scenario from INTERACTION dataset and the intersection scenario from InD datset to evaluate the transferability. Besides, we also trained the models and evaluated them on the Argoverse dataset.

**Rule-based methods** We first considered three rule-based methods. The IDM (Treiber et al., 2000) method basically follows its front car on the same reference path. The FSM-based (Zhang et al., 2017) method additionally considers cars on the other reference paths that have conflicts with the ego car, and follows the closest front car using the IDM model. When deciding the closest front car, the FSM-D method calculates each car's distance to the conflict point and chooses the closest one. The FSM-T method first calculates each vehicle's time needed to reach the conflict point by assuming they are running at a constant speed, and then chooses the closest one. We set the parameter of the IDM model as the values identified in urban driving situations (Liebner et al., 2012). As shown in Table 4, the rule-based method did not work well in the prediction task. Several reasons may be possible. First is that intersections, roundabouts and urban scenarios are really complicated with intense multi-agent interactions. Simple rules can hardly capture such complex behaviors while systematic rules are hard to manually design. A second reason is that, the parameter in the driving model is hard to specify as it is also scenario-and-individual-specific.

**Learning-based methods** We later evaluated several learning-based methods. Vallina LSTM (V-LSTM) adopts encoder decoder architecture with LSTM cells, which processes historic trajectories and generate future trajectory for each agent. Note that the representation tricks of position alignment and incremental prediction is also applied. To consider

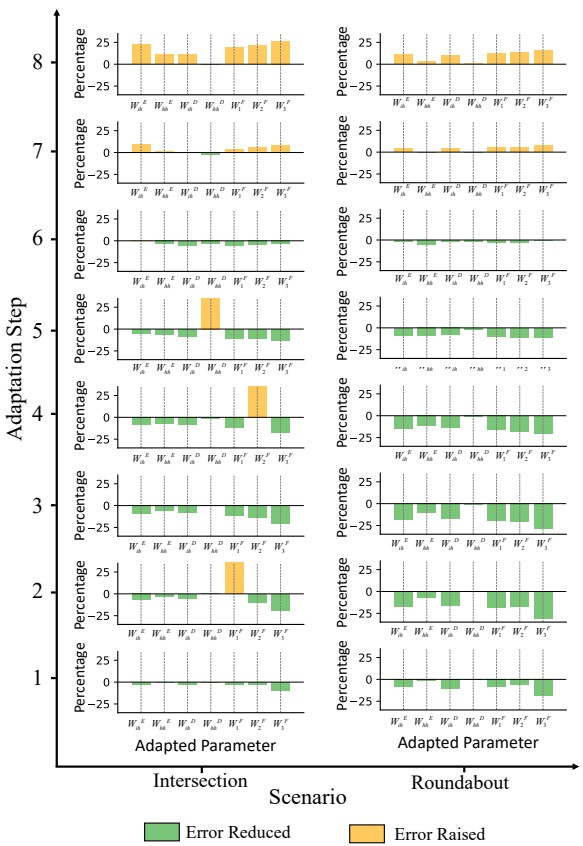

Figure 9: Percentage of short-term prediction error (ADE 2) reduced or raised after online adaptation, under 1) different adaptation step $\tau$; 2) different layer of parameters adapted; 2) different scenarios from the INTERACTION dataset. Three conclusions from the results: 1) online adaptation works the best around the adaptation step of 2 or 3; 2) prediction accuracy was improved by a higher percentage in the roundabout scenario (transferred) than in the intersection scenario (trained); 3) the best adaptation performance is obtained usually by adapting $W_3^F$, the last layer of the FC network in the decoder.

Table 4: Performance comparison with other methods. We evaluated the performance by calculating ADE (m) and FDE (m) between predicted trajectory and ground-truth trajectory in different horizons and scenarios/datasets. All models are trained and validated in the intersection scenario from INTERACTION dataset, and then zero-shot transferred to the unseen roundabout scenario from INTERACTION dataset and the intersection scenario from InD datset to evaluate the transferability. Besides, we also trained the models and evaluated them on the Argoverse dataset, where adaptation was not conducted as only one trajectory segment is provided for one agent.

| Scenario Dataset | Metric | Horizon | Rule-Based Method | | | Learning-Based Method | | | | | | |
|---|---|---|---|---|---|---|---|---|---|---|---|---|
| | | | IDM | FSM-D | FSM-T | V-LSTM | S-LSTM | S-GAN | Grip++ | Trajectron++ | HATN w/o adaptation | HATN (Ours) |
| Intersection INTERACTION | ADE | 3s | 2.847 ± 1.963 | 3.181 ± 2.403 | 3.372 ± 2.495 | 1.315 ± 1.177 | 1.277 ± 1.295 | 1.372 ± 1.221 | 0.949 ± 0.670 | 0.510 ± 0.440 | 0.305 ± 0.253 | **0.301 ± 0.250** |
| | | 0.3s | 0.042 ± 0.014 | 0.051 ± 0.034 | 0.054 ± 0.034 | 0.073 ± 0.006 | 0.047 ± 0.003 | 0.073 ± 0.004 | 0.024 ± 0.002 | **0.009 ± 0.008** | 0.029 ± 0.020 | 0.023 ± 0.014 |
| | FDE | 3s | 7.655 ± 5.536 | 8.709 ± 7.081 | 9.011 ± 7.192 | 3.386 ± 2.229 | 3.160 ± 2.387 | 3.469 ± 2.265 | 2.646 ± 5.773 | 1.617 ± 1.517 | 0.876 ± 0.835 | **0.877 ± 0.830** |
| | | 0.3s | 0.091 ± 0.033 | 0.103 ± 0.059 | 0.111 ± 0.059 | 0.140 ± 0.003 | 0.112 ± 0.001 | 0.153 ± 0.002 | 0.037 ± 0.007 | **0.014 ± 0.013** | 0.043 ± 0.032 | 0.032 ± 0.024 |
| Roundabout INTERACTION (Transfer) | ADE | 3s | 5.271 ± 1.950 | 4.637 ± 1.448 | 4.824 ± 1.509 | 2.202 ± 4.295 | 2.459 ± 4.675 | 2.273 ± 4.448 | 1.543 ± 1.021 | 1.250 ± 0.849 | 0.815 ± 0.526 | **0.815 ± 0.526** |
| | | 0.3s | 0.126 ± 0.062 | 0.093 ± 0.070 | 0.101 ± 0.004 | 0.061 ± 0.001 | 0.099 ± 0.002 | 0.090 ± 0.003 | 0.034 ± 0.004 | **0.015 ± 0.011** | 0.073 ± 0.065 | 0.052 ± 0.068 |
| | FDE | 3s | 13.891 ± 5.845 | 13.133 ± 4.208 | 13.505 ± 4.407 | 6.136 ± 8.445 | 6.668 ± 9.081 | 6.354 ± 9.499 | 4.352 ± 8.420 | 4.063 ± 2.706 | 2.038 ± 1.409 | **2.041 ± 1.409** |
| | | 0.3s | 0.206 ± 0.096 | 0.157 ± 0.105 | 0.162 ± 0.102 | 0.189 ± 0.008 | 0.162 ± 0.003 | 0.211 ± 0.006 | 0.055 ± 0.001 | **0.025 ± 0.020** | 0.108 ± 0.107 | 0.079 ± 0.114 |
| Intersection InD (Transfer) | ADE | 3s | 5.774 ± 1.950 | 4.924 ± 1.678 | 4.988 ± 1.651 | 2.922 ± 4.872 | 3.317 ± 5.388 | 3.333 ± 6.893 | 4.978 ± 39.385 | 1.655 ± 0.877 | 1.029 ± 0.670 | **0.914 ± 0.537** |
| | | 0.3s | 0.102 ± 0.060 | 0.081 ± 0.063 | 0.081 ± 0.063 | 0.151 ± 0.007 | 0.099 ± 0.002 | 0.189 ± 0.006 | 0.427 ± 0.506 | **0.032 ± 0.024** | 0.074 ± 0.071 | 0.057 ± 0.052 |
| | FDE | 3s | 15.212 ± 4.747 | 14.392 ± 4.527 | 14.634 ± 4.355 | 8.183 ± 9.133 | 6.668 ± 9.081 | 9.024 ± 11.733 | 11.080 ± 161.228 | 4.689 ± 2.404 | 2.480 ± 1.477 | **2.361 ± 1.473** |
| | | 0.3s | 0.184 ± 0.095 | 0.139 ± 0.097 | 0.139 ± 0.097 | 0.285 ± 0.013 | 0.162 ± 0.003 | 0.374 ± 0.015 | 0.055 ± 0.001 | **0.056 ± 0.045** | 0.113 ± 0.112 | 0.080 ± 0.086 |
| Urban Argoverse | ADE | 3s | 3.934 ± 3.498 | 5.651 ± 3.478 | 5.765 ± 3.587 | 2.827 ± 2.542 | 2.490 ± 2.512 | 3.087 ± 2.735 | 2.145 ± 2.255 | 1.551 ± 1.094 | **0.862 ± 0.629** | Same as the Left Column |
| | | 0.3s | 0.447 ± 0.648 | 0.457 ± 0.462 | 0.501 ± 0.651 | 0.756 ± 0.063 | 0.512 ± 0.055 | 0.786 ± 0.071 | 0.334 ± 0.008 | **0.071 ± 0.069** | 0.244 ± 0.216 | |
| | FDE | 3s | 8.917 ± 7.203 | 14.550 ± 6.971 | 14.823 ± 7.105 | 7.415 ± 4.725 | 6.288 ± 4.726 | 7.944 ± 4.983 | 3.745 ± 11.344 | 3.039 ± 2.831 | **1.797 ± 1.706** | |
| | | 0.3s | 0.602 ± 0.864 | 0.624 ± 0.873 | 0.587 ± 0.756 | 0.8866 ± 0.096 | 0.8437 ± 0.062 | 0.9438 ± 0.004 | 0.392 ± 0.547 | **0.089 ± 0.085** | 0.286 ± 0.267 | |

interaction between agents, Social LSTM (S-LSTM) (Alahi et al., 2016) additionally pools nearby agents' hidden states at every step using social pooling operation. Social-GAN (Gupta et al., 2018) modelled each agent as a LSTM-GAN, where a generator generates trajectory, which is then evaluated against the ground-truth trajectory by a discriminator. As shown in Table 4, both the three methods achieved much higher accuracy than traditional rule-based methods, distinguishing the power of deep learning. Though equipped with social pooling, S-LSTM only performed closely to V-LSTM, similar to experiment in Gupta et al. (2018); Salzmann et al. (2020). S-GAN suffered from unstable convergence during training and achieved slightly worse performance. In practice, the long running time caused by pooling operations render these methods intractable in real-time deployment. In comparison, benefit by GNN operation, our method achieved real-time computation and scaled pretty well with the agent number as shown in Appendix 7.6.

**Graph-based learning methods** We then evaluated other graph-based learning methods. Grip++ (Li et al., 2019) represents the interactions of close agents with a graph, applies graph convolution operations to extract spatial and temporal features, and subsequently uses an encoder-decoder LSTM model to make future predictions. We also implemented Trajectron++ (Salzmann et al., 2020), a GNN-based method. Trajectron++ takes vehicles as nodes of a graph and utilizes a graph neural network to conduct relationship reasoning. The map information is integrated by embedding the image of the map. A generative model is then used to predict future actions. The future trajectory is then generated by propagating the vehicle dynamics with the predicted actions. According to Table 4, we found these methods effectively surpassed previous three learning-based methods, benefit from the representation and relational inductive bias of graph. Among these methods, Trajectron++ and our method performed much better than Grip++ especially in transferability, which could be showing the better reasoning capability of graph operations compared to convolution operations. Among the two best methods, our method significantly outperformed Trajectron++, with ADE in 3 seconds lower by 41%, 34%, 42%, and 44% in the four scenarios respectively. Such results demonstrated that our method's great capability in the long-term prediction, benefiting from our design of semantic hierarchy and transferable representation. Nevertheless, Trajectron++ performed better than ours in the short-term prediction. One important reason is that the predicted trajectory of Trajectron++ is strictly dynamics-feasible, as it essentially predicts future actions and propagates them through the dynamics to retrieve future trajectories. Such results motivate us to include dynamic constraintin our future work. Besides, our method also differs from the two methods in the criteria to selecting interacting vehicles, which is analyzed in detail in Appendix E.

### 7.6  Running time

A key consideration in robotics is the runtime complexity. In particular, we care about how the number of agents would affect the model's running time. Consequently, we evaluated the time it took our method to perform forward inference in different agent number. For points with insufficient number of agents, we imputed values by copying existing agents. As shown in Figure 12, both the SGN and TEDNscaled well to the number of agents. For SGN, the running time was always near 0.001 seconds as the agent number increased to 100. The primary reason of such rapid computation is that in the SGN, the calculation for agents is conducted simultaneously in the form of matrix operation. In the TEDN, the computation time was near 0.018 seconds, which is slower than the SGN due to the iterative decoding process in the decoder. The running time scaled well as all the agents can be batched and calculated simultaneously in one forward inference. As for the online adaptation, the running time correlates with the adaptation steps. For instance, the adaptation took an average of 0.03 seconds per sample when set the adaptation step as $\tau$ 1, but the average time for adaptation step of 3 was 0.1 seconds for per sample. According to these results, our method successfully meets the real-time computation requirement. Note that such a real-time computation speed is achieved via code in python, which, in practical deployment, can be rewritten in C++ and paralleled for further acceleration.

## 8  Conclusion

**Summary** In this paper, inspired by humans' cognition model and semantic understanding during driving, we proposed a hierarchical framework to generate high-quality driving behavior prediction in a multi-agent dense-traffic environment. The proposed method consists of: 1) constructing semantic graph as a generic representation for the environment, which is transferable across different scenarios; 2) a semantic graph network for high-level intention prediction; 3) an trajectory encoder decoder networkfor low-level trajectory prediction; 4) an online adaptation module to adapt to different individuals and scenarios. The proposed method hierarchically divided the driving task into two sub-tasks with profound semantics, which simplifies the learning and provides more interpretability. Due to the generic representation for each hierarchy/sub-task, our method can be directly transferred to new scenarios after it is trained in one scenario. The online adaptation module can also adapt our method to different individual and scenarios for high-fidelity predictions.

In the experiments on real human data, we empirically investigated 1) what features and graph network architecture should be utilized in the high-level intention prediction; 2) whether commonly used features, representation tricks, and mechanisms would benefit low-level trajectory prediction; 3) how to integrate intention signal into the low-level trajectory prediction policy; 4) systematically evaluation of the online adaptation with a new set of metrics, including how much the prediction accuracy can be improved, what is the best step of observation and layer to adapt; 5) how our method outperforms other methods.

**Limitation and Future Work** In the future, we emphasize that the transferability and adaptability of prediction and planning algorithms are critical and worth more research efforts for the general and wide deployment of autonomous vehicles on the roads. As for our paper, there are several limitations and important next steps include: 1) including vehicle dynamics into the model and training to provide dynamic-feasible trajectory predictions and more accurate short-term predictions; 2) simultaneously adapt the parameter in both the high-level policy and low-level policy; 3) evaluate transferablity and generalizability in more scenarios and benchmarks; 4) comprehensively consider the multimodality in the prediction task, such as sampling in the high-level intention (insertion areas and the end point) and low-level trajectory, and formulating probabilistic matching with the lane reference; 5) explore the characteristics of hierarchical training and end-to-end training (McAllister et al., 2022; 2019), such as evaluating each module's effect on the overall system influence, adding an additional end-to-end training stage to explore the interaction between high-level policy and low-level policy, where they can accommodate to each other and improve overall performance.

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

## A    Network architecture detail

The detailed design of the network architecture can be found in Tabel 5.

Table 5: Architecture detail for SGN and EDN.

| SGN Architecture Detail | |
| --- | --- |
| $f_{rec}^1$ | GRU cells |
| $f_{rec}^2$ | GRU cells |
| $f_{enc}^1$ | Dense layer with tanh activation function |
| $f_{enc}^2$ | Dense layer with tanh activation function |
| $f_{att}$ | Dense layer with leaky relu activation function |
| $f_{enc}^3$ | Dense layer without activation function |
| $f_{enc}^4$ | Dense layer with tanh activation function |

| TEDN Architecture Detail | |
| --- | --- |
| Encoder | GRU cells |
| Decoder | GRU cell stacked with 3 dense layers, each with tanh activation function and dropout |

## B    Trajectory encoder decoder network evaluation

There are many existing works exploiting the encoder decoder architecture for the driving behavior generation (Park et al., 2018; Tang & Salakhutdinov, 2019; Zyner et al., 2019), but several questions still remain unclear: what coordinate should be employed? what features should be considered? what representation performs better? and whether commonly-used mechanisms in encoder decoder architecture can improve the performance in the driving task? To answer these questions, we conducted extensive ablation studies on the trajectory encoder decoder network (TEDN), where the intention signal is removed to avoid additional variance from the high-level policy. As shown in Table 6, starting from a naive encoder decoder that simply takes in position features and predicts positions, we incrementally added more tricks to test their effectiveness. Two metrics are set, absolute distance error (ADE) and final distance error (FDE). Note that the values in Table 6 may not exactly match with the values in Table 3. This is because in online adaptation process, early trajectory segments of one vehicle is purely used for adapting model parameters and not evaluated in prediction accuracy. For fair comparison, in Table 3, all methods also discarded early trajectory segments of one vehicle and evaluated the rest segments, so they are slightly different from Table 6, where all segment of one vehicle is used.

### B.1    Coordinate study

We first investigated which coordinate we should employ between Frenet and Cartesian coordinate. As shown in Table 6(a), the TEDN with Frenet coordinate performed 40% (ADE) and 35% (FDE) better than the TEDN with Cartesian coordinate in the intersection scenario. In the zero-transferred roundabout scenario,

Table 6: The statistical evaluation of the low-level behavior generation policy. We conducted experiments incrementally to explore the performance under different coordinates, features, representations, and mechanisms. The best performance is achieved when taking the Frenet coordinate, the speed and yaw feature in the input, and representation of incremental prediction and position alignment.

| (a) Coordinate | Intersection | | Roundabout (Transfer) | |
|---|---|---|---|---|
| | ADE | FDE | ADE | FDE |
| Cartesian | $1.53 \pm 1.22$ | $2.90 \pm 2.77$ | $12.57 \pm 5.68$ | $19.77 \pm 7.43$ |
| Frenet | $\mathbf{0.91 \pm 0.59}$ | $\mathbf{1.87 \pm 1.48}$ | $\mathbf{2.96 \pm 4.65}$ | $\mathbf{5.52 \pm 7.20}$ |

| (b) Speed Ablation | | Intersection | | Roundabout (Transfer) | |
|---|---|---|---|---|---|
| In | Out | ADE | FDE | ADE | FDE |
| × | × | $0.91 \pm 0.59$ | $1.87 \pm 1.48$ | $2.96 \pm 4.65$ | $5.52 \pm 7.20$ |
| ✓ | × | $0.71 \pm 0.54$ | $1.53 \pm 1.27$ | $\mathbf{2.33 \pm 3.84}$ | $\mathbf{4.30 \pm 5.44}$ |
| × | ✓ | $0.78 \pm 0.53$ | $1.74 \pm 1.41$ | $2.51 \pm 4.16$ | $4.97 \pm 6.35$ |
| ✓ | ✓ | $\mathbf{0.70 \pm 0.49}$ | $\mathbf{1.46 \pm 1.26}$ | $2.40 \pm 4.10$ | $4.67 \pm 6.03$ |

| (c) Yaw Ablation | | Intersection | | Roundabout (Transfer) | |
|---|---|---|---|---|---|
| In | Out | ADE | FDE | ADE | FDE |
| × | × | $0.71 \pm 0.54$ | $1.53 \pm 1.27$ | $2.33 \pm 3.84$ | $4.30 \pm 5.44$ |
| ✓ | × | $\mathbf{0.67 \pm 0.46}$ | $\mathbf{1.45 \pm 1.19}$ | $\mathbf{2.23 \pm 3.95}$ | $\mathbf{4.14 \pm 5.19}$ |
| × | ✓ | $0.73 \pm 0.50$ | $1.59 \pm 1.36$ | $2.34 \pm 3.96$ | $4.49 \pm 6.46$ |
| ✓ | ✓ | $0.67 \pm 0.48$ | $1.46 \pm 1.24$ | $2.51 \pm 4.60$ | $4.77 \pm 6.43$ |

| (d) Repre Ablation | | Intersection | | Roundabout (Transfer) | |
|---|---|---|---|---|---|
| Inc | Ali | ADE | FDE | ADE | FDE |
| × | × | $0.67 \pm 0.46$ | $1.45 \pm 1.19$ | $2.23 \pm 3.95$ | $4.14 \pm 5.19$ |
| × | ✓ | $0.48 \pm 0.44$ | $1.32 \pm 1.32$ | $1.26 \pm 0.95$ | $3.04 \pm 2.44$ |
| ✓ | × | $0.43 \pm 0.35$ | $1.36 \pm 1.19$ | $1.07 \pm 1.10$ | $2.66 \pm 2.31$ |
| ✓ | ✓ | $\mathbf{0.41 \pm 0.33}$ | $\mathbf{1.29 \pm 1.14}$ | $\mathbf{0.96 \pm 0.80}$ | $\mathbf{2.53 \pm 2.13}$ |

| (e) Mech Ablation | | Intersection | | Roundabout (Transfer) | |
|---|---|---|---|---|---|
| TF | Att | ADE | FDE | ADE | FDE |
| × | × | $\mathbf{0.41 \pm 0.33}$ | $\mathbf{1.29 \pm 1.14}$ | $\mathbf{0.96 \pm 0.80}$ | $\mathbf{2.53 \pm 2.13}$ |
| ✓ | × | $0.41 \pm 0.34$ | $1.32 \pm 1.16$ | $0.97 \pm 0.83$ | $2.54 \pm 2.14$ |
| × | ✓ | $0.43 \pm 0.34$ | $1.32 \pm 1.13$ | $1.10 \pm 1.32$ | $2.57 \pm 2.50$ |

though the performance of both two methods downgraded, the performance of the method on Cartesian coordinate decayed more significantly, with ADE higher by 324% and FDE higher by 258% compared to the method on Frenet coordinate. This is because the Frenet coordinate implicitly incorporates the map information into the model. Compared to running in any direction in the Cartesian coordinate, in the Frenet coordinate the vehicles only need to follow the direction of references paths, which constrains its behavior in a more predictable pattern. Note that in the Frenet coordinate, we use Dynamic Time Warping algorithm (Berndt & Clifford, 1994) to determine the most likely reference line that each agent lies on.

## B.2 Feature study

Second, we explored the effect of features, specifically, the speed feature and yaw feature. For each feature, we consider two circumstances. The first is to incorporate the feature into the input of the encoder to provide more information. The second is to set the feature as additional desired outputs of the decoder, which could possibly help to stabilize the learning for position prediction. Thus for each feature, we explored 4 settings in terms of whether or not to add the feature into the input or output.

As in Table 6(b), incorporating speed feature in either the input or output could both effectively improve performance in the two scenarios. When incorporating it into the input and output simultaneously, compared to only considering it in the input, the performance was slightly improved in the intersection scenario and

slightly degraded in the roundabout scenario. Concluding from the average performance in the two scenarios, we chose to only take the speed into the input of the encoder.

For the yaw feature, as shown in Table 6(c), taking it into input could slightly benefit the performance, while incorporating it into the output made the performance worse. One possible reason for such performance decay is that the yaw information has been already implicitly covered in the longitudinal and lateral speed information. Adding the yaw information into the output of the decoder could provide little additional information but indeed made the learning harder. We thus decided to incorporate the yaw feature only into the input of the encoder.

### B.3 Representation study

There are two commonly-used representation techniques to shape the data distribution. The first technique is called incremental prediction (Li et al., 2019). On each prediction step, this trickpredicts the relative position compared to the position of the last step (displacement of the time step), rather than directly predicting the absolute position. The second technique is called position alignment, which aligns the positions of each step to the vehicle's current position (Park et al., 2018). Specifically, when considering the prediction of a trajectory, the coordinate of all waypoints on the trajectory will be subtracted by the coordinate of the waypoint in the current step.According to Table 6(d), both two techniques could significantly improve the prediction accuracy, and applying both of them worked the best, improving the ADE by 38% in the intersection and by 56% in the roundabout.

### B.4 Mechanism study

There are two frequently used mechanisms in the encoder decoder architecture: teacher forcing (Williams & Zipser, 1989) and attention mechanism (Bahdanau et al., 2014). Teacher forcing aims at facilitating the learning of complex tasks while the attention is designed to attend differently to different historic input. From the results in Table 6(c), we can see neither of the two mechanisms could benefit the performance. Considering that the encoder decoder is used as a dynamics approximator, which is a relatively simple task, the teacher forcing has pretty limited performance improvement since the TEDNitself can already learn well enough. The attention mechanism indeed made the performance worse as the vehicle dynamics are most related to the recent state so previous states may not be necessarily informative.

As a summary of the ablation studies above, the best performance goes when taking the Frenet coordinate, the speed and yaw feature in the input, and representation of incremental prediction and position alignment. The teacher forcing and attention could not benefit the performance. Here we would also like to provide more precise discussion on the attention mechanism. There are actually multiple methods to use attention. In high-level policy, we did apply the attention mechanism to help with the relationship/interaction reasoning among multiple agents, which common in many existing works(Salzmann et al., 2020; Hu et al., 2020; Tang & Salakhutdinov, 2019; Li et al., 2020). But in the low-level policy, the attention is on the temporal information, namely attending to historic steps differently when predicting future trajectory. Such temporal attention has been quite popular in language processing field(Luong et al., 2015; Yang et al., 2016). However, our experiments in Table 6check the reference in the modified final versionindicate that such attention on one vehicle's own historic trajectory did not help much.

## C Intention signal integration evaluation

As mentioned in Sec 5.2, we have two intention signals, namely the goal state and the decoding step. The goal state signal refers to the goal position in the future horizon. The decoding step refers to the which step the decoding cell lies in the whole decodinghorizon. The two signals can be integrated into the low-level policy TEDNin several ways, such as appending it into the input or output of the decoder (note as Input or Output), embedding it into the hidden state at the first step (note as Hidden). For the goal state intention signal, we can additionally choose to introduce it by transforming the origin state of the vehicle into the state relative to the goal state (note as Transform). In this section, we evaluate the performance when the different intention signals are incorporated in different ways, as shown in Tabel 7.

Table 7: The statistical evaluation of different methods to introduce intention signals (the goal state and the decoding step) into the low-level trajectory prediction policy. The best performance is achieved when both the two intention signals are appended into the input feature.

| Scenario | Metric | No-Intention | With-Intention | | | |
|---|---|---|---|---|---|---|
| (a) Ground-truth Goal State Introduction | | | | | | |
| | | | Transform | Input | Output | Hidden |
| Intersection | ADE (m) | $0.41 \pm 0.11$ | $\mathbf{0.10 \pm 0.10}$ | $0.15 \pm 0.13$ | $0.17 \pm 0.16$ | $0.13 \pm 0.14$ |
| | FDE (m) | $1.29 \pm 1.30$ | $\mathbf{0.15 \pm 0.25}$ | $0.29 \pm 0.40$ | $0.38 \pm 0.41$ | $0.26 \pm 0.39$ |
| Roundabout (Transfer) | ADE (m) | $0.96 \pm 0.64$ | $\mathbf{0.42 \pm 0.37}$ | $0.51 \pm 0.46$ | $0.60 \pm 0.54$ | $0.48 \pm 0.41$ |
| | FDE (m) | $2.53 \pm 4.54$ | $\mathbf{0.72 \pm 0.66}$ | $0.94 \pm 0.73$ | $1.03 \pm 0.74$ | $0.84 \pm 0.67$ |

| Scenario | Metric | No-Intention | With-Intention | | | |
|---|---|---|---|---|---|---|
| (a) Predicted Goal State Introduction | | | | | | |
| | | | Transform | Input | Output | Hidden |
| Intersection | ADE (m) | $0.41 \pm 0.11$ | $0.31 \pm 0.25$ | $\mathbf{0.30 \pm 0.25}$ | $0.32 \pm 0.25$ | $0.31 \pm 0.25$ |
| | FDE (m) | $1.29 \pm 1.30$ | $0.92 \pm 0.85$ | $0.89 \pm 0.83$ | $0.89 \pm 0.83$ | $\mathbf{0.89 \pm 0.82}$ |
| Roundabout (Transfer) | ADE (m) | $0.96 \pm 0.64$ | $0.89 \pm 0.56$ | $\mathbf{0.86 \pm 0.61}$ | $0.92 \pm 0.75$ | $0.87 \pm 0.54$ |
| | FDE (m) | $2.53 \pm 4.54$ | $2.14 \pm 1.44$ | $\mathbf{2.12 \pm 1.51}$ | $2.19 \pm 1.69$ | $2.22 \pm 1.46$ |

| Scenario | Metric | No-Intention | With-Intention | | | |
|---|---|---|---|---|---|---|
| (a) Decoding step Introduction | | | | | | |
| | | | Transform | Input | Output | Hidden |
| Intersection | ADE (m) | $0.41 \pm 0.11$ | $0.30 \pm 0.25$ | $\mathbf{0.30 \pm 0.25}$ | $0.30 \pm 0.25$ | $0.30 \pm 0.24$ |
| | FDE (m) | $1.29 \pm 1.30$ | $0.89 \pm 0.83$ | $0.88 \pm 0.83$ | $0.89 \pm 0.83$ | $\mathbf{0.88 \pm 0.81}$ |
| Roundabout (Transfer) | ADE (m) | $0.96 \pm 0.64$ | $0.86 \pm 0.61$ | $\mathbf{0.82 \pm 0.53}$ | $0.87 \pm 0.59$ | $0.84 \pm 0.53$ |
| | FDE (m) | $2.53 \pm 4.54$ | $2.12 \pm 1.51$ | $\mathbf{2.06 \pm 1.43}$ | $2.15 \pm 1.51$ | $2.11 \pm 0.23$ |

## C.1 Integrating ground-truth goal state

First, we introduced the ground-truth goal state into the TEDN to measure the most performance improvement we can get from the ground-truth intention. According to Table 7(a), the Transform method had the best performance and reduced the ADE by 75% and 56% in the two scenarios, which represents the most benefit we can get with ground-truth intention but is also impossible as there exist inevitable errors in the predicted goal state.

## C.2 Integrating predicted goal state

When integrating the predicted goal state into TEDN, the error in the goal state prediction would perturb the performance. As in Table 7(b), while the performance of these goal state integration methods was close, appending the goal state into the input feature list performed the best, reducing the ADE by 26% and 10% in the two scenarios.

## C.3 Integrating decoding step

After introducing the goal state into the input feature, we further investigate the performance when decoding step signal is introduce as in Table 7(c). Similarly, appending it into the input performed the best, reducing the error in the roundabout scenario by 5%.

## C.4 Visualizing the effect of intention signal

In Figure 8, we illustrated the effect of introducing the intention signal (the predicted goal state and the decoding step), by calculating the prediction error of each step in the future 30 steps. Obviously, as the prediction horizon extended, the prediction became more difficult and the error grew exponentially. After introducing the intention signal, the error growth was effectively suppressed, especially in the long horizon.

# D    Online adaptation evaluation

The online adaptation aims at capturing nuances in different individuals and scenarios by exploiting historic observations to subtly adjust model parameters. In this section, we address two questions by empirical evaluation 1) How many steps of observation are the best to adapt? 2) What is the best layer in the network to adapt? The new set of metrics mentioned in Sec 6 and shown in Figure 4 are used for the evaluation.

## D.1    Trade-off in adaptation step

The adaptation step $\tau$ is an important parameter in the multi-step online adaptation algorithm. On the one hand, we can obtain more information by increasing $\tau$. On the other hand, the behavior gap between the current time and historic time also increases. As a result, there is a performance trade-off when we increase the adaptation step $\tau$. To empirically answer the question that how many steps are the best, We run the online adaptation on both the intersection and roundabout scenarios, and collected statistical results of the absolute distance error (ADE) between the ground-truth and the predicted trajectory. Note that here when evaluating one adaptation step $\tau$, we adapted parameters in different layers and chose the best performance as the performance of that adaptation step $\tau$.

Figure 10(a) shows the online adaptation results in the intersection scenario. According to ADE 1 and ADE 2 in the first two images, as the adaptation step $\tau$ increased, the prediction error of the first $\tau$ step on both the historic trajectory (ADE 1) and the current trajectory (ADE 2) increased. But the adaptation can depress such error growth. For ADE 1, the error was reduced by higher percentage when more steps $\tau$ of observation are used for adaptation, because more information was gained. For the ADE 2, as the adaptation steps $\tau$ increased, the error reduction percentage first increased and reached a peak of 20.5% at 3 adaptation steps. After that, the error reduction percentage decreased as the behavior gap had come into effect due to a longer time lag $\tau$. The last two images show the error in the whole historic trajectory (ADE 3) and current trajectory (ADE 4). For the ADE 3, longer $\tau$ led to a higher percentage of error reduction due to more information gained. However, for the ADE 4 evaluating the performance of long-term prediction, due to the insufficient information and behavior gap, the improvement was limited. Similar results can be found in the roundabout scenario in Figure 10(b). But in the ADE 2, more improvement (28%) was achieved in the short-term prediction, due to the fact that the model was not trained on the roundabout scenario and there was more space for adaptation.

With these analyses, a conclusion is that though the adaptation does not help with the long-term behavior prediction in the next 3 seconds, the short-term behavior prediction in the next 0.3 seconds is effectively improved by 20.5% and 28.7% in the two scenarios. Such improvement in short-term prediction is valuable as it can effectively enhance safety in close-distance interactions.

## D.2    Adaptation layer choice

The neural network consists of many layers of parameters and it remains a question which layer shall we adapt in order to get the best adaptation performance. Thus in the section, we empirically analyze the performance of adapting different layers.

We first claim and denote all the layers. In the trajectory encoder decoder network, both the encoder and decoder consist of single-layer gated recurrent units (GRU), and the decoder is additionally stacked with three layers of fully connected (FC) networks. We denote the encoder GRU's input-hidden weights as $W_{ih}^E$, the encoder GRU's hidden-hidden weights as $W_{hh}^E$, the decoder GRU's input-hidden weights as $W_{ih}^D$, the decoder GRU's hidden-hidden weights as $W_{hh}^D$, and the weights of the three-layer FC as $W_1^F$, $W_2^F$, $W_3^F$ respectively.

As in Figure 9, we show the percentage of the change of ADE 2 after adaptation, under different adaptation step $\tau$, different parameters, and different scenarios. We have several observations: 1) as the adaptation step $\tau$ increased, the percentage of error reduction increased and reached the peak at 2 or 3 steps. But after that the help of adaptation decayed, and after 7 steps, the predictions became even worse due to a big behavior gap; 2) Intuitively, the adaptation worked better in the roundabout scenario, compared to the intersection

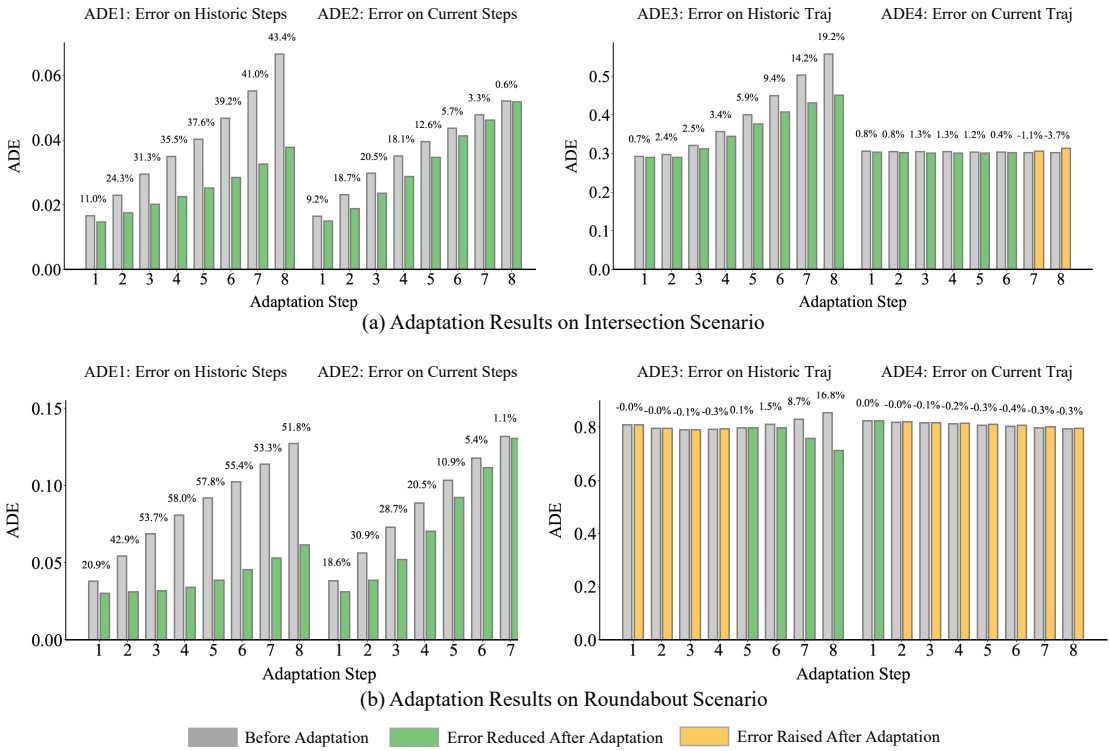

Figure 10: Online adaptation performance analysis. According to ADE 1 and ADE 3 in (a)(b), as adaptation step $\tau$ increased, the online adaptation could get more information and improve prediction accuracy by a higher percentage. In ADE 2, as the adaptation step $\tau$ increases, the improved percentage first grew higher due to more information obtained, but then declined due to the behavior gap between earlier time and current time. When $\tau$ was 3, the short-term prediction was improved by 20% and 28% on the two scenarios respectively. In ADE 4, we can see online adaptation can barely benefit long-term prediction.

scenario, as the model was trained on the intersection scenario and directly transferred to the roundabout scenario. 3) The best adaptation performance is usually achieved by adapting the layer $W_3^F$, which is the last layer of the FC network in the decoder.

## E  Interacting car density

In multi-agent systems, it is important to answer the question which vehicles should be considered as the interacting vehicles. In some works, all the vehicles in the scene are considered as the interacting vehicles; while in some works, the interacting vehicles are defined as the vehicles within a certain range of the ego vehicle (Salzmann et al. (2020); Li et al. (2019)). However, we argue that distance may not necessarily determine interaction. For instance, cars that are close but driving in opposite direction may not be interacting at all. Essentially, interactions will happen among cars which are driving into a common area. We consider the interacting vehicles as the vehicles whose reference lines conflict with ego vehicles' reference line, and regard them as the node in our graph.

In Figure 11, we show the interacting density distribution as we choose interacting vehicles by different criteria. These results are collected from the real human data in the intersection scenario of the Interaction Dataset. In Figure 11(a), when we considered all the vehicles in the scene as the interacting vehicles, each ego car would be assumed to interact with 10 to 17 cars for most of the time. In Figure 11(b), when we took the vehicles within the range of 30 meters as the interacting car, there would be 5 to 10 interacting vehicles. Such results are counter-intuitive as it would be tough for humans to attend to so many cars at the same time. But when we considered cars within the range of 10 meters, as shown in Figure 11(c), the

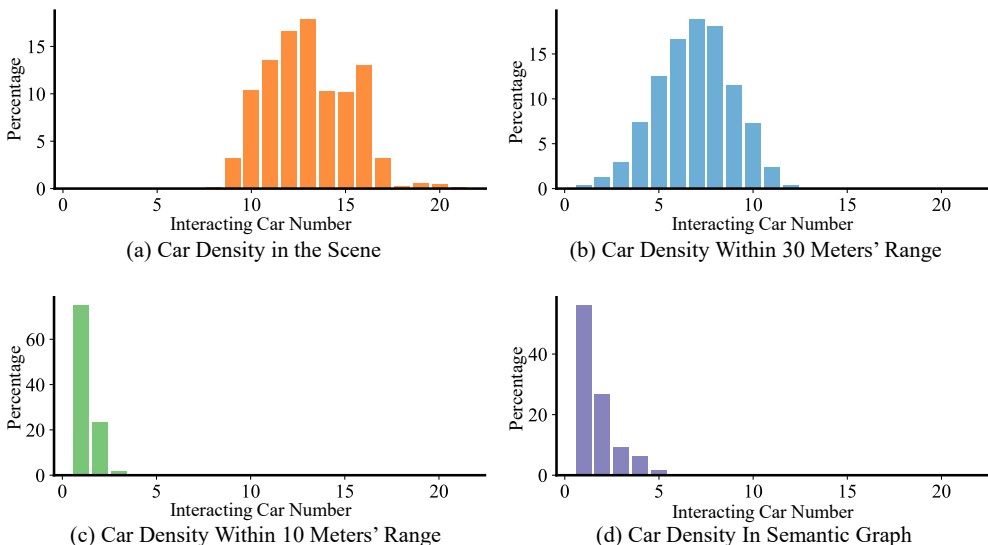

Figure 11: Percentage of interacting car number, when different criteria is used to choose interacting vehicles.

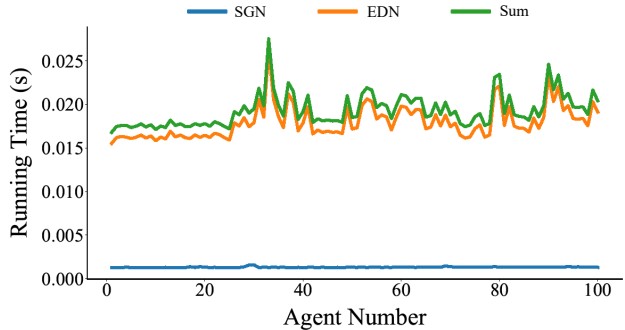

Figure 12: Running time of our method under different agent numbers.

number of interacting cars was further reduced to be no more than 2. Nevertheless, such assumption also has its drawback, as people may still care about vehicles far from them as long as they are intervening each other. In our method, we care about vehicles whose reference lines conflict with ego car's reference line. As in Figure 11(d), the percentage of interacting vehicles number gradually decayed untill 5, which may be closer to real driving situations.

## F  Running Time

As shown in Figure 12, our method meets the real-time computation requirement and performs well when the agent of number increases.

