# OpenReview forum: "Transferable and Adaptable Driving Behavior Prediction"
_TMLR — Rejected by TMLR_

### Review · Reviewer_exFQ · 2022-09-02

**Summary Of Contributions:**

The paper presents an approach for trajectory prediction of multiple agents in the context of automated driving. The proposed semantic graph generates a representation for each agent that is later provided to the semantic graph network (SGN). The SGN outputs the probability of each agent occupying a dynamic insertion area and the goal state distribution for each agent and the interacting agents. In addition, it is proposed an encoder-decoder approach for the future trajectory prediction of each agent. Finally, there is an online adaptation scheme for improving the model's accuracy with a Kalman filter. The approach is evaluated on a single benchmark where it shows promising results for an intersection scenario.


**Broader Impact Concerns:**

No concerns noticed.

**Requested Changes:**

- In the introduction, the paper motivates the idea of behaviour prediction, but it does not define the task. Based on the explanation, it reduces to trajectory prediction.

- Related work to be discussed:
	- Lee, Namhoon, et al. "Desire: Distant future prediction in dynamic scenes with interacting agents." Proceedings of the IEEE conference on computer vision and pattern recognition. 2017.
	- Deo, Nachiket, and Mohan M. Trivedi. "Multi-modal trajectory prediction of surrounding vehicles with manoeuvre based lstms." 2018 IEEE Intelligent Vehicles Symposium (IV). IEEE, 2018.
	- Zhao, Tianyang, et al. "Multi-agent tensor fusion for contextual trajectory prediction." Proceedings of the IEEE/CVF Conference on Computer Vision and Pattern Recognition. 2019.
	- Tang, Charlie, and Russ R. Salakhutdinov. "Multiple futures prediction." Advances in Neural Information Processing Systems 32 (2019).
	- Liu, Yicheng, et al. "Multimodal motion prediction with stacked transformers." Proceedings of the IEEE/CVF Conference on Computer Vision and Pattern Recognition. 2021.


- Overall, the paper should be rewritten with a focus on trajectory prediction.

- The definition of the intention signal and goal state after Eq. 3 is vague.

- An additional evaluation would be welcome.

- The ablation studies from the Appendix are important. Some of them could be moved to the main paper, e.g. model component evaluation.

**Strengths And Weaknesses:**

Strengths:

+ The approach is well-described and it is easy to understand.

+ The idea of modelling all agents and treating them as ego vehicles is interesting. The paper deals with a challenging problem.

+ The presented results are promising on the INTERACTION dataset.

+ The online adaptation approach is lightweight and generalises well.

Weaknesses:


- The introduction refers to the task of behaviour prediction of multi-agent systems. However, the paper is about trajectory prediction. Thus, the introduction is a bit misleading since behaviour is more than trajectories, e.g. intention, activity forecasting and recognition among others. It would be helpful to be more clear about the paper's content in the introduction and avoid abstractions or claims that are not completely supported by the proposed approach.

- The paper claims human-like driving behaviour prediction, but it is not clear in which sense. It reads like presenting an interface that allows humans to interact with the vehicle's behaviour.

- The high-level intention generation is not well-defined. 1. It is not clear what contributes to the results and 2. what means to be high-level. In addition, the intention is currently described as occupancy map filling, e.g. Sec. 4.2.3.

- The approach is pretty complex. It combined graph neural networks with an encoder-decoder network and finally it has also to include an extended Kalman filter. At last, there is also a Gaussian mixture model at the intention layer. It is not clear whether the approach can generalise to different types of scenes. Also, the single benchmark evaluation does not support the generalization to a wide range of settings.

- Evaluation on more than a single benchmark is necessary to demonstrate generalisation, e.g. nuScenes, pedestrian datasets such as ETH and UCY datasets.

---

> ### Author Response · Authors · 2022-10-11
> **Response to reviewer exFQ (Part 1/5)**
>
> We thank the reviewer for taking the time to review the paper and giving detailed and helpful suggestions. We are glad to know that you like our idea, approach, presentation, and experiment. We appreciate your recognition! The following are responses to the comments and suggestions you made. The index of some tables and figures may be different from the initially submitted paper due to the modifications, and we use the updated index in the response. We hope that they could solve your concerns. Please let us know if you want to know anything further or discuss any questions and concerns further.
>
> > Comment 1: The introduction refers to the task of behaviour prediction of multi-agent systems. However, the paper is about trajectory prediction. Thus, the introduction is a bit misleading since behaviour is more than trajectories, e.g. intention, activity forecasting and recognition among others. It would be helpful to be more clear about the paper's content in the introduction and avoid abstractions or claims that are not completely supported by the proposed approach.
>
> **Response:** Thank you for pointing this out. We agree that behavior prediction may mean differently to people with different backgrounds such as robotics and computer vision. In our paper, we aim at predicting the intentions and trajectories of human drivers. Specifically, our high-level module (SGN) can predict vehicles' intention (insertion area) and our low-level module (TEDN) can predict future trajectory. So we have added the clarifications that we aim at predicting driving behaviors (intentions and trajectory) in the abstract and introduction in the updated manuscript, which is also attached as below. We thank the reviewer to help us make our paper clearer.
>
> In the abstract: *“we propose HATN, a hierarchical framework to generate high-quality, transferable, and adaptable predictions for driving behaviors **(intentions and trajectories)** in multi-agent dense-traffic environments.”*
>
> In the introduction: *"Moreover, most state-of-art behavior prediction **(intention and trajectory prediction) algorithms for AVs, once trained for one scenario, are brittle due to overspecialization and tend to fail when transferred to similar or new scenarios."*
>
> > Comment 2: The paper claims human-like driving behaviour prediction, but it is not clear in which sense. It reads like presenting an interface that allows humans to interact with the vehicle's behaviour.
>
> **Response** We thank the reviewer the question. The goal of the prediction task is to generate a vehicle's future trajectory that is close to real human driving data, the closer (more accurate) the better. As a result, human-likeness is used to describe such accuracy. To avoid confusion here, we simply replace it with "accurate" in the updated manuscript. We thank the reviewer to make our paper clear.
>
> > Comment 3: The high-level intention generation is not well-defined. 1. It is not clear what contributes to the results and 2. what means to be high-level. In addition, the intention is currently described as occupancy map filling, e.g. Sec. 4.2.3.
>
> **Response:** We thank the reviewer for the valuable question that can make our paper clear. In the high-level policy, there are two intention signals to be predicted. The first is which DIA (slot) the vehicle will insert into. This is a classification problem and our model will output the probability of inserting into each DIA $w_i$. Another intention signal is the goal point of one vehicle in a certain future horizon $g$, which can be used to guide trajectory prediction in the low-level policy. The term high-level refers to the here we predict the intention of the vehicle, such as the goal point. In comparison, the low-level policy is responsible to leverage the goal point to predict a detailed future trajectory. Actually, in our submitted paper, we did state the definition of intention, such as Sec 3 and Sec 4.2.3. But we agree that these statements may not have been obvious and clear. And the new contents above have been added to the updated manuscript. We really appreciate this comment to make our paper clear.

---

> ### Author Response · Authors · 2022-10-11
> **Response to reviewer exFQ (Part 2/5)**
>
> > Comment 4: The approach is pretty complex. It combined graph neural networks with an encoder-decoder network and finally it has also to include an extended Kalman filter. At last, there is also a Gaussian mixture model at the intention layer. It is not clear whether the approach can generalise to different types of scenes. Also, the single benchmark evaluation does not support the generalization to a wide range of settings.
>
> **Response:** We thank the reviewer for raising the question. Yes, our method takes a hierarchical framework, consisting of a high-level policy for intention prediction, a low-level policy for trajectory prediction, and an adaptation module to adjust model parameters online.
> As for the model complexity, including intention prediction and trajectory prediction have actually been quite prevailing in related works [1-4], since the two sub-task inherently exist in the driving task. The major difference is that, in those works, the model is usually trained in an end-to-end manner. Graph neural network is also popular for driving prediction in recent years[5-7], and online adaptation of the prediction models is also raising more attention[8-10]. In fact, one major contribution of our paper is that, this is the very first method to explicitly and simultaneously consider drivers’ nature of hierarchy, transferability, and adaptability, to the best of our knowledge. The gaussian mixture model is also a quite common trick in probabilistic prediction methods [5][11].
>
> As for the generalizability, as indicated in the introduction of our paper, such a hierarchical design springs from humans' cognition mechanism: the hierarchy [12][13] and selective attention [14][15]. By dividing the whole framework into different hierarchies, we can get several advantages: 1）the hierarchical design can simplify the monolithic difficult learning task into several easier sub-tasks. So the state space of each sub-task is reduced to only include compact and key features relevant to the sub-task, and each sub-policy is only responsible to learn the sub-task alone without information entanglement. Thus the learning can be more efficient; 2) each learned sub-policy is intrinsically scenario-transferable and reusable because the representation of insertion slot and reference trajectory can be abstracted out and consistently defined across different scenarios. These advantages of the hierarchical framework have been exploited in many works [16-18]. In our paper, we evaluate the transferability of our method by zero-shot transferring our model to unseen scenarios. As shown in Table 4, our method successfully outperformed other methods in transferability. Thus we think our claim is proper. We also added two additional benchmarks to evaluate the transferability, see the next response for details.
>
> [1] C. Tang and R. R. Salakhutdinov, “Multiple futures prediction,” Advances in Neural Information Processing Systems, vol. 32, 2019.
>
> [2] N. Rhinehart, R. McAllister, K. Kitani, and S. Levine, “Precog: Prediction conditioned on goals in visual multi-agent settings,” in Proceedings of the IEEE/CVF International Conference on Computer Vision, pp. 2821–2830, 2019
>
> [3] Hang Zhao, Jiyang Gao, Tian Lan, Chen Sun, Benjamin Sapp, Balakrishnan Varadarajan, Yue Shen, Yi Shen, Yuning Chai, Cordelia Schmid, et al. Tnt: Target-driven trajectory prediction. arXiv preprint arXiv:2008.08294, 2020.
>
> [4] C. Choi, S. Malla, A. Patil, and J. H. Choi, “Drogon: A trajectory prediction model based on intention-conditioned behavior reasoning,” arXiv preprint arXiv:1908.00024, 2019
>
> [5] Tim Salzmann, Boris Ivanovic, Punarjay Chakravarty, and Marco Pavone. Trajectron++: Dynamicallyfeasible trajectory forecasting with heterogeneous data. In Computer Vision–ECCV 2020: 16th European Conference, Glasgow, UK, August 23–28, 2020, Proceedings, Part XVIII 16, pp. 683–700. Springer, 2020.
>
> [6] Jiachen Li, Fan Yang, Masayoshi Tomizuka, and Chiho Choi. Evolvegraph: Multi-agent trajectory prediction with dynamic relational reasoning. Advances in neural information processing systems, 33:19783–19794, 2020.
>
> [7] Xin Li, Xiaowen Ying, and Mooi Choo Chuah. Grip++: Enhanced graph-based interaction-aware trajectory prediction for autonomous driving. arXiv preprint arXiv:1907.07792, 2019.
>
> [8] Abulikemu Abuduweili and Changliu Liu. Robust nonlinear adaptation algorithms for multitask prediction networks. International Journal of Adaptive Control and Signal Processing, 35(3):314–341, 2021.
>
> [9] Letian Wang, Liting Sun, Masayoshi Tomizuka, and Wei Zhan. Socially-compatible behavior design of autonomous vehicles with verification on real human data. IEEE Robotics and Automation Letters, 6(2): 3421–3428, 2021.
>
> [10] Ivanovic, Boris, James Harrison, and Marco Pavone. "Expanding the deployment envelope of behavior prediction via adaptive meta-learning." arXiv preprint arXiv:2209.11820 (2022).

---

> ### Author Response · Authors · 2022-10-11
> **Response to reviewer exFQ (Part 3/5)**
>
>
> [11] Yeping Hu, Wei Zhan, and Masayoshi Tomizuka. Probabilistic prediction of vehicle semantic intention and motion. In 2018 IEEE Intelligent Vehicles Symposium (IV), pp. 307–313. IEEE, 2018b.
>
> [12] Botvinick, Matthew M., Yael Niv, and Andew G. Barto. "Hierarchically organized behavior and its neural foundations: A reinforcement learning perspective." Cognition 113.3 (2009): 262-280.
>
> [13] Flet-Berliac, Yannis. "The promise of hierarchical reinforcement learning." The Gradient 9 (2019).
>
> [14] Radulescu, Angela, Yael Niv, and Ian Ballard. "Holistic reinforcement learning: the role of structure and attention." Trends in cognitive sciences 23.4 (2019): 278-292.
>
> [15] Niv, Y., 2019. Learning task-state representations. Nature neuroscience, 22(10), pp.1544-1553.
>
> [16] Dayan, Peter, and Geoffrey E. Hinton. "Feudal reinforcement learning." Advances in neural information processing systems 5 (1992).
>
> [17] Bacon, Pierre-Luc, Jean Harb, and Doina Precup. "The option-critic architecture." Proceedings of the AAAI Conference on Artificial Intelligence. Vol. 31. No. 1. 2017.
>
> [18] Liu, Siqi, et al. "From motor control to team play in simulated humanoid football." Science Robotics 7.69 (2022): eabo0235.
>
> > Comment 5: Evaluation on more than a single benchmark is necessary to demonstrate generalisation, e.g. nuScenes, pedestrian datasets such as ETH and UCY datasets.}
>
> **Response:** We thank the reviewer for the valuable comment. We agree with the reviewer that evaluation on more benchmarks is needed. Since we concentrate on driving behavior prediction (intention and trajectory), we consider real vehicle datasets instead of pedestrian datasets. Besides, our method need centerlines of the scene to check conflict among agents and extract DIAs from the scene. Thus we choose to evaluate our method on additional two datasets: the Ind dataset and the Argoverse 1 dataset, which provide rich map information. As shown in Table 4, our method also outperforms other methods by a considerable margin on the two added benchmarks (beat Trajectron++ by 44 \% and 40\% in ADE 3s on the two datasets respectively).
>
> > Comment 6: In the introduction, the paper motivates the idea of behaviour prediction, but it does not define the task. Based on the explanation, it reduces to trajectory prediction.
>
> **Response:** Thank you for pointing this out. This comment is responded earlier in comment 1 but we also attach the response here for easy checking. We agree that behavior prediction may mean differently to people with different backgrounds such as robotics and computer vision. In our paper, we aim at predicting the intentions and trajectories of human drivers. Specifically, our high-level module (SGN) can predict vehicles' intention (insertion area) and our low-level module (TEDN) can predict future trajectory. So we have added the clarifications that we aim at predicting driving behaviors (intentions and trajectory) in the abstract and introduction in the updated manuscript, which is also attached as below. We thank the reviewer to help us make our paper clearer.
>
> In the abstract: *“we propose HATN, a hierarchical framework to generate high-quality, transferable, and adaptable predictions for driving behaviors **(intentions and trajectories)** in multi-agent dense-traffic environments.”*
>
> In the introduction: *"Moreover, most state-of-art behavior prediction **(intention and trajectory prediction)** algorithms for AVs, once trained for one scenario, are brittle due to overspecialization and tend to fail when transferred to similar or new scenarios."*

---

> ### Author Response · Authors · 2022-10-11
> **Response to reviewer exFQ (Part 4/5)**
>
> > Comment 7: Related work to be discussed:
> Lee, Namhoon, et al. "Desire: Distant future prediction in dynamic scenes with interacting agents." Proceedings of the IEEE conference on computer vision and pattern recognition. 2017.
>
> Deo, Nachiket, and Mohan M. Trivedi. "Multi-modal trajectory prediction of surrounding vehicles with manoeuvre based lstms." 2018 IEEE Intelligent Vehicles Symposium (IV). IEEE, 2018.
>
> Zhao, Tianyang, et al. "Multi-agent tensor fusion for contextual trajectory prediction." Proceedings of the IEEE/CVF Conference on Computer Vision and Pattern Recognition. 2019.
>
> Tang, Charlie, and Russ R. Salakhutdinov. "Multiple futures prediction." Advances in Neural Information Processing Systems 32 (2019).
>
> Liu, Yicheng, et al. "Multimodal motion prediction with stacked transformers." Proceedings of the IEEE/CVF Conference on Computer Vision and Pattern Recognition. 2021.
>
> **Response:** We thank the reviewer for pointing these references out. Below are some brief discussions of the papers and we have added them to the related work in the updated manuscript.
>
> The first paper applies a CAVE-based RNN encoder-decoder architecture, where semantic scene context/element is embedded by CNN. Several scene elements are considered, such as road, sidewalk, and vegetation. However, important information such as lane relationships is not considered. Besides, there are several drawbacks when embedding BEV images or occupancy grids via CNNs: 1) there is an uneasy trade-off between the resolution of the spatial grid/image and the field of view. 2) representations obtained from the BEV images or occupancy grids via CNNs are of high abstraction level, which may fail under scenarios that are not well covered by the training data. 3) explicit relationship reasoning among agents is still missing, and it is difficult to capture long-range interactions via convolutions with small receptive fields.
>
> The second paper follows a typical LSTM-based trajectory prediction approach. The author predefined high-level intentions (left/right lane change, keep straight), classify trajectories into different intention classes, and then predict future trajectories. However, the paper only trains and tests on the highway dataset (NGSIM), which is not challenging enough compared to the urban scenarios we considered in our paper. In general, the method used in this paper cannot be easily adapted to other complicated urban scenarios with varying types of high-level intentions and interactions.
>
> The third paper applies an LSTM+GAN approach. Agents are encoded into spatial tensors in a similar form of occupancy grid methods, and extract scene context via CNN-bade models, which has similar drawbacks as the first paper.
>
> The fourth paper takes a typical RNN-based encoder-decoder prediction network. Rasterized image is used to represent contextual scene information and pass through CNN, which has similar drawbacks as the first and third papers. The output layer is similar to our approach, which outputs the mean and variance of the Gaussian distribution. Driving and interaction modes are automatically learned in latent space instead of predefined.
>
> The fifth paper used vectorized representation for scene context, similar to Vectornet. Although vectorized representations are applicable to various driving scenarios, all relations between roads or agents, which can sometimes be pretty simple and obvious,  have to be learned by the network and there is no guarantee that those known relations can be learned correctly. Besides, this is an interesting paper with a unique way to select multi-modality in driving. However, compared to our method, this work only considers the multimodality in the static environment and missed the dynamic information. In contrast, our method considers multi-modal goals/intentions (insertion area) by taking into account the behaviors of other surrounding dynamic agents.
>
> > Comment 8: The definition of the intention signal and goal state after Eq. 3 is vague.
>
> **Response:** We thank the reviewer for the valuable question that can make our paper clear. In the high-level policy, there are two intention signals to be predicted. The first is which DIA (slot) the vehicle will insert into. This is a classification problem and our model will output the probability of inserting into each DIA $w_i$. Another intention signal is the goal point of one vehicle in a certain future horizon $g$, which can be used to guide trajectory prediction in the low-level policy. The term high-level refers to the here we predict the intention of the vehicle, such as the goal point. In comparison, the low-level policy is responsible to leverage the goal point to predict a detailed future trajectory. In the updated manuscript, these contents have been added right above Eq.3 and in Sec 4.2.3. We really appreciate this comment to make our paper clear.

---

> ### Author Response · Authors · 2022-10-11
> **Response to reviewer exFQ (Part 5/5)**
>
> > Comment 9: An additional evaluation would be welcome.
>
> **Response:** We thank the reviewer for the valuable comment. We agree with the reviewer that evaluation on more benchmarks is needed. Since we concentrate on driving behavior prediction (intention and trajectory), we consider real vehicle datasets instead of pedestrian datasets. Besides, our method needs centerlines of the scene to check conflict among agents and extract DIAs from the scene. Thus we choose to evaluate our method on additional two datasets: the Ind dataset and Argoverse 1 dataset, which provide rich map information. As shown in Table 4, our method also outperforms other methods by a considerable margin on the two added benchmarks (beat Trajectron++ by 44 \% and 40\% in ADE 3s on two added datasets respectively).
>
> > Comment 10: The ablation studies from the Appendix are important. Some of them could be moved to the main paper, e.g. model component evaluation.
>
> **Response:** We thank the reviewer for the valuable comments, and we definitely agree. Since the ablation studies on the low-level policy and adaptation have been summarized in the main paper, we have further moved the model component evaluation on the high-level policy into the main paper.

---

### Review · Reviewer_Qe3M · 2022-09-02

**Summary Of Contributions:**

The authors propose the Hierarchical Adaptable and Transferable Network (HATN) framework. HATN consists of a high-level intention module (a semantic graph network), a low-level trajectory prediction stage (an encoder-decoder architecture), and an online adaptation component (a modified EKF). The paper demonstrates capabilities for one-shot transfer to new scenarios (from an intersection to a roundabout), extensively ablates the components of the approach, and portrays handling of interactions in real-world human driving data from the INTERACTION dataset.

**Broader Impact Concerns:**

The authors provide thoughtful consideration to the various components of the proposed HATN framework. They conduct extensive ablation studies to determine the influence of the different moving parts and representations. It would be helpful to include an explicit discussion of the limitations of the proposed approach. In addition to my suggestions above, it would be good to talk about concerns like out-of-distribution examples that may be encountered in the real-world during system deployment.

**Requested Changes:**

My requested changes are based on the weaknesses described above along with some additional requested clarifications. For more details, please see my response in the previous question.

1. Baseline against a version of HATN that directly predicts the goal in the high-level stage without considering the insertion into slots mechanism. (Critical)

2. Improve Fig. 1 intuition (see above) and incorporate notation from Sec. 3 into Fig. 2 for improved clarity. (Highly encouraged)

3. Improved clarity surrounding whether DIAs are formed only with respect to other vehicles or also traffic signs/road geometries and whether $M$ vehicles correspond to the selected interacting vehicles or whether there are just $M$ vehicles in the scene. (Highly encouraged)

4. Clarification for why some observations have two time coordinates, while others have one in Fig. 4 (e.g., $Y_{t-\tau,t}$ versus $Y_{t-\tau}$). (Highly encouraged)

5. Clarification in the first paragraph of Sec. 6.2 for what was the problem considered by Abuduweili & Liu (2021) as compared to driving behavior prediction. (Highly encouraged)

6. There was no validation set used in the experiments. Do the authors believe that this could have resulted in over-fitting to the test set in the results? (Nice to have)

7. The tables have distributional metrics indicated. Are these standard errors over the data? Were the experiments run over multiple seeds? (Highly encouraged)

8. In Table 3, were HATN and all other considered learning-based methods trained only on the intersection scenario and then one-shot transferred to the roundabout scenario? How were methods like Trajectron++ able to generalize given that the HD map serves as input into that architecture? (Highly encouraged)

9. Real-time performance discussion should be moved to the main paper. Why were the adaptation run times not included in Fig. 12? (Highly encouraged)

10. Some sweeping statements in the paper need to be made more precise (see above). (Critical)

11. A discussion should be included for how the hierarchical and interpretability goals of the approach are achieved while not constraining the low-level policy to end in the goal position. (Critical)

12. Some context around occupancy-based BEV prediction approaches would be helpful (see above [1-6]). (Nice to have)

13. Clarification surrounding the contribution of the paper as compared to prior work. (Critical)

14. More detailed definitions for relative and absolute features as well as position alignment and incremental prediction would be useful. (Highly encouraged)

15. Including ground-truth future trajectories in the illustrations. Potential update to the color scheme (see above for suggestions). (Highly encouraged)

16. Fix the color referencing in the caption of Fig. 6. (Critical)

17. Significantly improved paper polish. (Critical)

18. More detailed discussion of limitations. (Highly encouraged)

19. The references should be proofread (e.g., to ensure the year is not entered twice in a citation, conference proceedings are formatted consistently, the conference is listed instead of ArXiv whenever possible). (Nice to have)

20. The inclusion of context in Sec. B.1 for why GAT might do better than the proposed method. (Nice to have)

21. Improved precision on the conclusion in Sec. 4 regarding attention not being helpful to the behavior prediction task (see above). (Highly encouraged)

**Strengths And Weaknesses:**

### Strengths:
* Well motivated by a practical issue in the need for generalizable, transferrable, and real-time trajectory prediction approaches.
* The authors do a good job of motivating the hierarchical approach to behavior prediction and framing many existing ideas in a compelling manner.
* The authors highlight and investigate the importance of the coordinate frame representation for generalizability to different scenarios.
* Fig. 3 does a good job at illustrating the DIA/SG construction.
* The authors do a great job at emphasizing the benefits of the proposed approach throughout the paper (e.g., second paragraph in Sec. 5).
* Fig. 4 effectively visualizes the proposed online adaptation metrics.
* The discussion regarding the trade-offs for the number of adaptation steps was helpful and insightful.
* The experiments were thoughtfully constructed and fairly extensive, supporting most of the claims made in the paper.
* The ablation studies were helpful in understanding the influence of the different components (e.g., "the intention signal ... suppressed error growth").
* Fig. 9 provided a lot of intuition for the online adaptation algorithm and the chosen number of adapted steps. Specifically, it was great to see that the unseen during training scenario saw the biggest improvement.
* The discussion regarding potential reasons for certain observed trends was insightful (e.g., rule-based methods failing in highly interactive scenes, graph-based approaches having success in interactive scenarios).
* The discussion regarding the No-Temporal method being able to better generalize was interesting in Sec. B.1.

### Weaknesses:
* There are quite a few sweeping statements and/or assumptions in the paper that need to be softened to be made more precise, in my opinion. For example, "humans can drive through and across these environments easily, even while talking to friends or shaking to the music". This is not necessarily true, as accidents often happen on the road due to distracted driving, which motivates the need for AVs. Another example is, "humans usually take in [low-dimensional] state [features] to make decisions at a low resolution". This is a large assumption about human cognition that should either be softened or supported with citations.
* In Fig. 1, it is not clear why the ego vehicle is considering inserting into oncoming traffic. This becomes clear later in the paper (the insertion slots are dynamic), however requires more context at this stage of the paper.
* Since the trajectories from the low-level policy are not constrained by the goal point from the high-level policy, they do not have the same end-point (see Fig. 1(b)). This seems to defeat the purpose of a hierarchical approach as the low-level policy is doing the majority of the work to predict the trajectory with optional guidance from an input goal position. This also hurts the discussed added interpretability of the approach.
* The paper argues that BEV-based prediction approaches can handle only a fixed number of agents and are not order-invariant. It would be helpful to also discuss occupancy grid prediction approaches [1-6] here as they circumvent these issues.
* From my understanding, the proposed approach combines three mostly existing approaches together into a hierarchical framework. Although this is a valid contribution, it was not immediately clear that this is the case. For example, I am not fully clear on what parts of the semantic graph network (SGN) have been published previously.
* Multimodality of the goal intention is not considered in the proposed approach for simplicity. Multimodality at the high level is often important for behavior reasoning; however, in HATN it can be circumvented with multimodality at the low-level as it is not constrained by the end goal position.
* I could not find descriptions of what exactly are relative and absolute features.
* The decoding step is not included in Eq. (21).
* Figs. 5-7 are difficult to evaluate without ground-truth future trajectories illustrated. The darker DIA corresponding to higher likelihood was difficult to see in the figures. A potential solution would be to make non-interacting cars gray and transparent, and have transparency correspond to likelihood for the DIA?
* Fig. 6 caption is referring to a yellow car in Fig. 6(a) and an orange car in Figs. (c) and (d), but the car colors seem to be the same.
* It is not clear that the slot-insertion mechanism is crucial for the performance of the proposed framework as only the goal is passed to the low-level module.
* The real-time capability of the method should be moved into the main paper as this is an important point from the contributions paragraph.
* The paper lacks polish and needs thorough proof-reading as the typos and grammatical errors detract from the presented message. I include a few examples below.
    * "essence of how human learns to drive" -> "essence of how humans learn to drive"
    * "fresh human driver"
    * "to: 1) generating ... 2) providing ... 3) reporting" -> "to 1) generate ... 2) provide ... 3) report"
    * Contribution 1. is a run-on sentence and should be broken up into a few sentences.
    * Many missing (e.g., "Convolution Neural Network (CNN) applies" -> "the convolutional neural network (CNN) applies) and extraneous (e.g., "investigate the such a trade-off" -> "investigate such a trade-off") articles.
    * "the interaction among vehicles have been insufficiently reasoned" - grammatically awkward
    * "On the other hand, These methods"
    * "Thus the methods which predicting" -> "Thus the methods which predict"
    * "There are a few works that not only utilizes ... but also trains" -> "There are a few works that not only utilize ... but also train"
    * Missing period in first paragraph on page 5.
    * "To be able to conduct ... [they] utilize ... to conduct model linearization and empirically compare the results of adapting different sets of parameters with different steps." - the phrasing needs to be improved.
    * Repeated redefinition of acronyms (e.g., SG, SGN, DIA).
    * "read boundary" -> "road boundary" (2x)
    * Missing period at the end of footnote 2.
    * Frenet coordinate is sometimes not capitalized.
    * Fig. 6 caption is missing a period.
    * "a sequence of interaction" -> "a sequence of interactions"
    * "the orange vehicle behind DIA $A_3$ ran away" -> "the orange vehicle behind DIA $A_3$ drove away (this error is repeated)
    * A section title starts at the end of a page (pg. 18).
    * Missing period - top of pg. 19.
    * "include dynamic constrain in our future work" -> "include dynamic constraints in our future work"
    * The references need proofreading to ensure proper capitalization, consistent conference titles, and no duplications (e.g., year twice in one reference).
    * "deocding horizon" -> "decoding horizon"
    * "alighment" -> "alignment"
* The limitations of the proposed approach are not sufficiently discussed. For example, emphasis is placed on the proposed method not being trained in an end-to-end manner. However, there can be disadvantages to training without end-goal awareness [7,8]. In this case, the high-level policy training may benefit from end-goal (trajectory prediction) performance awareness to achieve targeted success in choosing the correct intention for difficult, highly interactive scenarios.
* In Sec. B.1, no context was provided for why GAT might do better than the proposed method.
* The conclusion in Sec. 4 regarding attention not being helpful to the behavior prediction task needs to be made more precise as Trajectron++ does quite well in the considered experiments and uses attention within the architecture.

[1] Itkina, Masha et al. "Dynamic environment prediction in urban scenes using recurrent representation learning." 2019 IEEE Intelligent Transportation Systems Conference (ITSC). IEEE, 2019.

[2] Toyungyernsub, Maneekwan et al. "Double-prong ConvLSTM for spatiotemporal occupancy prediction in dynamic environments." 2021 IEEE International Conference on Robotics and Automation (ICRA). IEEE, 2021.

[3] Lange, Bernard et al. "Attention Augmented ConvLSTM for Environment Prediction." 2021 IEEE/RSJ International Conference on Intelligent Robots and Systems (IROS). IEEE, 2021.

[4] Mohajerin, Nima and Mohsen Rohani. "Multi-step prediction of occupancy grid maps with recurrent neural networks." Proceedings of the IEEE/CVF Conference on Computer Vision and Pattern Recognition. 2019.

[5] Thomas, Hugues, et al. "Learning Spatiotemporal Occupancy Grid Maps for Lifelong Navigation in Dynamic Scenes." 2022 International Conference on Robotics and Automation (ICRA). IEEE, 2022.

[6] Mahjourian, Reza, et al. "Occupancy flow fields for motion forecasting in autonomous driving." IEEE Robotics and Automation Letters 7.2 (2022): 5639-5646.

[7] McAllister, Rowan, et al. "Robustness to out-of-distribution inputs via task-aware generative uncertainty." International Conference on Robotics and Automation (ICRA). IEEE, 2019.

[8] McAllister, Rowan, et al. "Control-Aware Prediction Objectives for Autonomous Driving." arXiv (2022).

---

> ### Author Response · Authors · 2022-10-11
> **Response to reviewer Qe3M (Part 1/10)**
>
> We thank the reviewer for taking the time to review the paper and giving detailed and helpful suggestions. We are glad to know that you like our idea, motivation, illustration, presentation, discussion, and experiment. We appreciate your recognition! The following are responses to the comments and suggestions you made. The index of some tables and figures may be different from the initially submitted paper due to the modifications, and we use the updated index in the response. We hope that they could solve your concerns. Please let us know if you want to know anything further or discuss any questions and concerns further.
>
> > Comment 1: There are quite a few sweeping statements and/or assumptions in the paper that need to be softened to be made more precise, in my opinion. For example, "humans can drive through and across these environments easily, even while talking to friends or shaking to the music". This is not necessarily true, as accidents often happen on the road due to distracted driving, which motivates the need for AVs. Another example is, "humans usually take in [low-dimensional] state [features] to make decisions at a low resolution". This is a large assumption about human cognition that should either be softened or supported with citations.
>
> **Response:** We thank the reviewer for raising the detailed suggestions. We agree that these statements should be made more precise. In the first statement, we actually wanted to emphasize that AVs can still learn a lot from humans. In the updated manuscript, we have added the conditions for the statement, which then becomes ``In many safe driving cases, humans can drive through and across these environments easily, even multi-tasking such as talking to friends or shaking to the music". In the second statement, we have added three citations, where [1] is explaining this from the hierarchical learning perspective, and the [2][3] reveals humans' hierarchical decision with evidence in neuroscience and cognition science.
>
> [1] Dayan, Peter, and Geoffrey E. Hinton. "Feudal reinforcement learning." Advances in neural information processing systems 5 (1992).
>
> [2] Botvinick, Matthew M., Yael Niv, and Andew G. Barto. "Hierarchically organized behavior and its neural foundations: A reinforcement learning perspective." Cognition 113.3 (2009): 262-280.
>
> [3] Niv, Y., 2019. Learning task-state representations. Nature neuroscience, 22(10), pp.1544-1553.
>
> > Comment 2: In Fig. 1, it is not clear why the ego vehicle is considering inserting into oncoming traffic. This becomes clear later in the paper (the insertion slots are dynamic), however requires more context at this stage of the paper.
>
> **Response:** We thank the reviewer for the detailed and kind comments. We have updated Fig 1, adding more notations on the figure, drawing the goal point corresponding to each dynamic slot, and drawing the future trajectory corresponding to each goal point. In the caption of Fig.1, we also added a reminder to guide readers to see case studies figures in latter part of the paper for illustrations of the dynamic insertion process: ``One can refer to Fig 5 6 7 for dynamic illustration of the insertion process."
>
> > Comment 3: Since the trajectories from the low-level policy are not constrained by the goal point from the high-level policy, they do not have the same end-point (see Fig. 1(b)). This seems to defeat the purpose of a hierarchical approach as the low-level policy is doing the majority of the work to predict the trajectory with optional guidance from an input goal position. This also hurts the discussed added interpretability of the approach.
>
> **Response:** Thanks for the detailed comments. Actually, the end point is indeed used to guide the trajectory prediction in the low-level policy, as indicated in Fig.2, Sec 5.2, Fig 8, and Table 2. So that the advantage of hierarchies and interpretability is still valid. We agree that we did not make this clear in the early stage of the paper, and not expressing this information clearly in Fig.1, which leads to confusion. We have updated Fig.1, where we added the goal point corresponding to each dynamic slot, and drew the future trajectory corresponding to each goal point. So that the relationships between dynamic slot, goal point, and future trajectory are strengthened (dynamic slot relates to goal points, and the goal point constrains/guides the future trajectory). We really thank the reviewer for pointing this out and making our paper clear.

---

> > ### Author Response · Authors · 2022-10-11
> > **Response to reviewer Qe3M (Part 3/10)**
> >
> > > Comment 5: From my understanding, the proposed approach combines three mostly existing approaches together into a hierarchical framework. Although this is a valid contribution, it was not immediately clear that this is the case. For example, I am not fully clear on what parts of the semantic graph network (SGN) have been published previously.
> >
> > **Response:** We thank the reviewer for pointing this out. The novelty of our paper, in our opinion, mainly lies in our hierarchical framework and our insight of 1) dividing the monolithic driving task into several simplified sub-tasks; 2) applying generic and unified representation for each sub-task so that they are reusable and transferable across different scenarios; 3) online adapt the model parameter to deal with the heterogeneity and stochasticity in the prediction task. To the best of our knowledge, this is the **very first method** to explicitly and simultaneously take the driver’s nature of hierarchy, transferability, and adaptability into account. As for the three components in our paper (SGN, EDN, OA), they are basically the specific algorithms we choose to achieve our goal of hierarchy, transferability, and adaptability, and there can be some other choices on the specific algorithms. Besides, these algorithms are not immediately suitable to achieve our goal and some important modifications need to be made to make the solution holistic. For example, the modifications for the high-level policy (SGN) include: 1) the original method in Hu et al, 2020 considers how the vehicles will insert into DIAs, such as which location in the DIA to insert into, and how much time is needed for the insertion. However, these features are DIA-centric and cannot be used to guide the vehicles' trajectory prediction in the low-level policy. Thus we modified the algorithm to predict vehicles' own goal point, which can then be used in the low-level policy. 2) our SGN takes the spatial-temporal 3D semantic graph from historic time step $t − T_h$ to the current time step $t$ as the input, rather than only the spatial 2D semantic graph of the current time step t in previous work. So that the temporal information of the DIAs is considered. 3) our SGN considers the prediction of all vehicles in the scene instead of only the ego vehicle, to further augment the data. The effect of the two changes is ablated in Table 2 of the updated manuscript (No-Temporal method, and Single-Agent method). It turns out that these modifications have been pretty effective and important. The goal point prediction error of the No-Temporal and Single-Agent methods is higher than our method by 69% and 7% respectively.
> >
> > > Comment 6: Multimodality of the goal intention is not considered in the proposed approach for simplicity. Multimodality at the high level is often important for behavior reasoning; however, in HATN it can be circumvented with multimodality at the low-level as it is not constrained by the end goal position.
> >
> > **Response:** We thank the reviewer for the comment. Multimodality is essentially in many aspects. In the high-level policy, there is multimodality in terms of the inserting areas, which is considered in the training of the high-level policy. In the low-level policy, however, we did not consider the inserting areas multimodality for simplicity, but only selected the most-likely area and used the mean of that area to generate the most-likely trajectory. We agree that the multimodality in the trajectory generation is important, and list the sampling in insertion areas and the end point as an important step for future work. Besides, the multimodality in the reference lines will also be considered in the future. These future steps have been added in Sec 8.

---

> > ### Author Response · Authors · 2022-10-11
> > **Response to reviewer Qe3M (Part 4/10)**
> >
> > > Comment 7: I could not find descriptions of what exactly are relative and absolute features.
> >
> > **Response:** We thank the reviewer for the detailed comments. Actually, we have defined the relative and absolute features in the first paragraph of Sec 4.1, but it seems these statements have not been obvious and rich enough. Thus we have enriched the statements and added additional notations for clearer definition in the updated manuscripts, which is also attached below:
> >
> > *"To capture each DIA's crucial information for humans' decision, we extract four high-level features under the Frenet coordinate for each DIA: $\textbf{X}=(d_{f/r}^{lon}, v_{f/r}, \phi_{f/r}, l)$, where $d_{f/r}^{lon}$ denotes longitudinal distance to the conflict point of front or rear boundary; $v_{f/r}$ denotes the velocity of front or rear boundary; $\phi_{f/r}$ denotes the angle of front or rear boundary; $l = d_r^{lon} - d_f^{lon}$ measures the length of the DIA. These features are defined as the absolute features $\textbf{X}$. To facilitate relationship inference among DIAs, we also define the relative feature $\textbf{X}'$ for each DIA by aligning it with the reference DIA. Specifically, we choose the front DIA as the reference DIA $\textbf{X}_{ref}$ , because the ego vehicle is implicitly represented by the rear boundary of the front DIA. Each DIA's relative feature $\textbf{X}'$ is then derived by subtracting the DIA's absolute feature $\textbf{X}$ by the reference DIA's absolute feature $\textbf{X}_{ref}$."*
> >
> > > Comment 8: The decoding step is not included in Eq. (21).
> >
> > **Response:** We thank the reviewer for the detailed reading and suggestion. We have added $t'$ as the notation for the decoding step and added it in Eq (21) in the updated manuscript, resulting in the equation below:
> > \begin{equation*}
> > 	\hat{\textbf{Y}}_t = f_{dec}(c_t, s_t, g_t, t'; \theta^{D}).
> > \end{equation*}
> >
> > > Comment 9: Figs. 5-7 are difficult to evaluate without ground-truth future trajectories illustrated. The darker DIA corresponding to higher likelihood was difficult to see in the figures. A potential solution would be to make non-interacting cars gray and transparent, and have transparency correspond to likelihood for the DIA?
> >
> > **Response:** We thank the reviewer for the valuable suggestions that can make our illustration more clear and illustrative. In Fig. 5-7 of our updated manuscript: 1) we have added the ground-truth trajectory for each vehicle; 2) we have also added a dashed-line box and one circle node to mark each DIA, and used transparency to present the likelihood; 3) yes, making non-interacting cars gray and transparent is definitely a good idea, and we have also made the change.
> >
> > > Comment 10: Fig. 6 caption is referring to a yellow car in Fig. 6(a) and an orange car in Figs. (c) and (d), but the car colors seem to be the same.
> >
> > **Response:** We thank the reviewer for the detailed observation and valuable suggestions that can make our figures more clear. We have updated the figures and checked the color reference carefully to make them consistent.
> >
> > > Comment 11: It is not clear that the slot-insertion mechanism is crucial for the performance of the proposed framework as only the goal is passed to the low-level module.
> >
> > **Response:** We thank the reviewer for raising the valuable question. The slot-insertion mechanism is indeed important for our framework. On the one hand, the slot insertion choice and the goal point are correlated. When vehicles choose different slots to insert into, they will generate different goal points to reach, which are then used in the low-level module. On the other hand, in the training of the high-level policy (Eq.17), the classification loss training (slot insertion) can serve as an auxiliary task, which can facilitate the training of the main task [5][6], namely the regress loss training (goal point). To support such claims, in the ablation study of the high-level policy (Table. 2), we also added one baseline No-Class-Loss, which only considers the regress loss for goal state prediction ($\mathcal{L}_{regress}$), and does not consider the classification loss for insertion area prediction ($\mathcal{L}_{class}$). It turns out the accuracy for goal point prediction decreased compared to our method, which demonstrated the effectiveness of incorporating insertion area prediction as an auxiliary task for goal point prediction.
> >
> > [5] Hasenclever, Leonard, et al. "Comic: Complementary task learning & mimicry for reusable skills." International Conference on Machine Learning. PMLR, 2020.
> >
> > [6] Mirowski, Piotr, et al. "Learning to navigate in complex environments." arXiv preprint arXiv:1611.03673 (2016).
> >
> > > Comment 12: The real-time capability of the method should be moved into the main paper as this is an important point from the contributions paragraph.
> >
> > **Response:** We thank the reviewer for the valuable suggestions. We have moved these contents to the Sec. 7.6 in the updated manuscript.

---

> > ### Comment · Reviewer_Qe3M · 2022-10-26
> > **Goal-Conditioning Question**
> >
> > I just wanted to clarify this point with the authors. The goal point is used to **guide** the trajectory prediction. But the trajectory prediction is not **constrained** to end in the goal point. Is that correct?
> >
> > Thanks!

---

> > > ### Author Response · Authors · 2022-10-27
> > > **Response to Goal-Conditioning Question**
> > >
> > > Dear reviewer Qe3M,
> > >
> > > Thanks for the question. Yes, we use the end point to **guide** the trajectory prediction instead of exactly **constraining** the trajectory prediction in the low-level trajectory prediction policy. The reason is that, as shown in the high-level intention policy evaluation in Table 2, the predicted goal point has itself errors compared to the ground-truth goal point. The predicted goal point can only be close to but not exactly match the ground-truth goal point (a perfect prediction of the vehicle’s future location in the long term is impossible, especially in complicated interactive driving environments.). Therefore, constraining or enforcing the prediction to exactly match the predicted goal point could bring the errors in the high-level policy into the low-level policy, leading to additional prediction errors. As a result, we instead regard the predicted goal point as a coarse target point to guide the trajectory prediction. Such a mechanism is similar to humans’ driving behavior, who does not pursue a goal point as a hard constraint, but instead use it as a coarse target point. In a such scheme, the goal point and the inserted area can provide more interpretability, serving as a coarse goal. Besides, as shown in Fig 8 and Table 3, using the goal point to guide trajectory prediction can effectively increase the prediction accuracy and suppress error growth in the long term prediction, which supports our method's effectiveness.
> > >
> > > Best regards,
> > > authors

---

> ### Author Response · Authors · 2022-10-11
> **Response to reviewer Qe3M (Part 2/10)**
>
>
> > Comment 4: The paper argues that BEV-based prediction approaches can handle only a fixed number of agents and are not order-invariant. It would be helpful to also discuss occupancy grid prediction approaches [1-6] here as they circumvent these issues.
>
> **Response:** We thank the reviewer for suggesting these papers that utilize the occupancy grid as one of the input representations. For these six papers [1-6], they all utilized dynamic occupancy grid maps to store the dynamic state of agents in each grid and the occupancy belief, which is then treated as images and fed into CNN-based networks. However, one of the major drawbacks of such representation is that it usually does not consider HD map-related information such as lane and lane relations. Such information is important for accurate motion prediction, especially in interactive scenarios. Besides, there is an uneasy trade-off between the resolution of the spatial grid and the field of view. Moreover, representations obtained from the occupancy grid by CNN-based models are with high abstraction levels, which may fail under scenarios that are not well covered by the training data.
>
> Moreover, in our submitted paper, we talked about the problem of agent number and order invariance. But it seems our statement was not accurate. It is the methods that simply use DNN for interaction modeling that suffer from the dynamic agent number and input order. And the methods using CNN process BEV images in the form of pixels or grids can deal with dynamic agent numbers and unfixed input order.
>
> Based on the information above, we have re-clarified these statements in the updated manuscript, as attached below:
>
> *"When modeling agents' interaction, one intuitive idea is simply using Deep Neuron Network (DNN), by flattening features of all agents and feeding them into deep neuron networks. However, such designs lack flexibility as they usually only consider a fixed number of agents. On the other hand, These methods are also not order-invariant: processing agents in a different order would produce different results, while we would expect the same results for the same scene. Thus such methods have been rarely used in related works.
> To bypass these problems and further impose agents' spatial relationship in the reasoning process, Convolution Neural Network (CNN) applies convolution operations on data (commonly in grid or pixel form) to model spatial and temporal relationships, such as 3D voxelization, rasterization in 2D bird’s-eye view (BEV) and occupancy grid (Radwan et al., 2020; Su et al., 2021; Itkina et al., 2019; Toyungyernsub et al., 2021; Lange et al., 2021; Mohajerin & Rohani, 2019; Thomas et al., 2022; Mahjourian et al., 2022; Lee et al., 2017; Zhao
> et al., 2019; Tang & Salakhutdinov, 2019). However, such representations have several drawbacks: 1) there is an uneasy trade-off between the resolution of the spatial grid/image and the field of view. 2) the occupancy grid does not take into account HD map-related information such as lane and lane relations, which are important for accurate future prediction, especially in interactive scenarios. 3) representations obtained from the BEV images or occupancy grids via CNNs are of high abstraction level, which may fail under scenarios that are not well covered by the training data. 4) explicit relationship reasoning among agents is still missing, and it is difficult to capture long-range interactions via convolutions with small receptive fields."*

---

> ### Author Response · Authors · 2022-10-11
> **Response to reviewer Qe3M (Part 5/10)**
>
> > Comment 13: The paper lacks polish and needs thorough proof-reading as the typos and grammatical errors detract from the presented message. I include a few examples below
>
> **Response:** Thanks for the detailed comments to polish our paper. All the mentioned issues have been resolved and we have proofread our paper again to improve it. We will continue to polish the paper after the rebuttal window.
>
> > Comment 14: The limitations of the proposed approach are not sufficiently discussed. For example, emphasis is placed on the proposed method not being trained in an end-to-end manner. However, there can be disadvantages to training without end-goal awareness [7,8]. In this case, the high-level policy training may benefit from end-goal (trajectory prediction) performance awareness to achieve targeted success in choosing the correct intention for difficult, highly interactive scenarios.
>
> **Response:** We thank the reviewer for the valuable comment. We quite agree with the reviewer that, though training the model in a hierarchical manner has multiple advantages, it can also have disadvantages that each module may fall in the local optima in the intermediate objectives and does not always align with the overall 'end goal' system performance. Thus it is pretty interesting and important to 1) evaluate each module's effect on the overall system performance, 2) explore more 'end-goal' objectives as the two papers mentioned by the reviewer, 3) add an additional end-to-end training stage to explore the interaction between high-level policy and low-level policy, where they can accommodate to each other and improve overall performance. We thank the reviewer for pointing this out and have added these discussions and the two papers on our limitation and future work in Sec 8, as attached below:
>
> *As for our paper, there are several limitations and important next steps include: 1) including vehicle dynamics into the model and training to provide dynamic-feasible trajectory predictions and more accurate short-term predictions; 2) simultaneously adapting the parameter in both the high-level policy and low-level policy; 3) evaluating transferability and generalizability in more scenarios and benchmarks; 4) comprehensively consider the multimodality in the prediction task, such as sampling in the high-level intention (insertion areas and the end point) and low-level trajectory, and formulating probabilistic matching with the lane reference; 5) explore the characteristics of hierarchical training and end-to-end training (McAllister et al., 2022; 2019), such as evaluating each module’s effect on the overall system influence, adding an additional end-to-end training stage to explore the interaction between high-level policy and low-level policy, where they can accommodate to each other and improve overall performance.*
>
> > Comment 15: In Sec. B.1, no context was provided for why GAT might do better than the proposed method.
>
> **Response:** We thank the reviewer for the detailed comment. We think this phenomenon is as interesting as that of the No-Temporal method. The No-Temporal method has the best transferability performance, and one possible explanation is that the absence of temporal information constrained the model's capability and thus avoided over-fit. The reason why the GAT performs best in the inserted area prediction is possibly similar: the inserted area prediction is a relatively simple classification task. While absolute features may be enough for the model to learn, additional relative features may actually provide redundant information and make learning more difficult. We thank the reviewer for the question and are also open to other explanations.

---

> ### Author Response · Authors · 2022-10-11
> **Response to reviewer Qe3M (Part 6/10)**
>
> > Comment 16: The conclusion in Sec. 4 regarding attention not being helpful to the behavior prediction task needs to be made more precise as Trajectron++ does quite well in the considered experiments and uses attention within the architecture.
>
> **Response:** We thank the reviewer for the suggestion. We agree that the conclusion should be made more precisely. In our paper, there are actually multiple aspects of attention. In high-level policy, we did apply the attention mechanism to help with the relationship/interaction reasoning among multiple agents, which is consistent with Trajectron++[7] and many other methods [8-10]. But in the low-level policy, the attention is on the temporal information, namely attending to historic steps differently when predicting each step of the future trajectory. Such temporal attention has been quite popular in language processing field [11][12]. However, our experiments in Table 6 indicate that such attention on one vehicle’s own historic trajectory did not help much. We have added such discussions in the Appendix B.4 of our updated manuscript.
>
> [7] T. Salzmann, B. Ivanovic, P. Chakravarty, and M. Pavone, “Trajectron++: Dynamically-feasible trajectory forecasting with heterogeneous data,” in European Conference on Computer Vision, pp. 683–700, Springer, 2020
>
> [8] C. Tang and R. R. Salakhutdinov, “Multiple futures prediction,” Advances in Neural Information Processing Systems, vol. 32, 2019.
>
> [9] Y. Hu, W. Zhan, and M. Tomizuka, “Scenario-transferable semantic graph reasoning for interaction-aware probabilistic prediction,” IEEE Transactions on Intelligent Transportation Systems, 2022.
>
> [10] Jiachen Li, Fan Yang, Masayoshi Tomizuka, and Chiho Choi. Evolvegraph: Multi-agent trajectory prediction with dynamic relational reasoning. Advances in neural information processing systems, 33:19783–19794, 2020.
>
> [11] . Yang, D. Yang, C. Dyer, X. He, A. Smola, and E. Hovy, “Hierarchical attention networks for document classification,” in Proceedings of the 2016 conference of the North American chapter of the association for computational linguistics: human language technologies, pp. 1480–1489, 2016
>
> [12] M.-T. Luong, H. Pham, and C. D. Manning, “Effective approaches to attention-based neural machine translation,” arXiv preprint arXiv:1508.04025, 2015
>
> > Comment 17: Baseline against a version of HATN that directly predicts the goal in the high-level stage without considering the insertion into slots mechanism. (Critical)
>
> **Response:** We thank the reviewer for the detailed comments. This comment is similar to comment 11. But we also attach the response here for easy checking. The slot-insertion mechanism is indeed important for our framework. On the one hand, the slot insertion choice and the goal point are correlated. When vehicles choose different slot to insert into, they will generate have different goal points to reach, which is then used in the low-level module. On the other hand, in the training of the high-level policy (Eq.17), the classification loss training (slot insertion) can serve as an auxiliary task, which can facilitate the training of the main task [7][8], namely the regress loss training (goal point). To support such claims, in the ablation study of the high-level policy (Table. 2), we also added one baseline No-Class-Loss, where only considers the regress loss for goal state prediction ($\mathcal{L}_{regress}$), and does not consider the classification loss for insertion area prediction ($\mathcal{L}_{class}$). It turns out the accuracy for goal point prediction decreased by 6% compared to our method, which demonstrated the effectiveness of incorporating insertion area prediction as an auxiliary task for goal point prediction.
>
> [7] Hasenclever, Leonard, et al. "Comic: Complementary task learning & mimicry for reusable skills." International Conference on Machine Learning. PMLR, 2020.
>
> [8] Mirowski, Piotr, et al. "Learning to navigate in complex environments." arXiv preprint arXiv:1611.03673 (2016).
>
> > Comment 18: Improve Fig. 1 intuition (see above) and incorporate notation from Sec. 3 into Fig. 2 for improved clarity. (Highly encouraged)
>
> **Response:** We thank the reviewer for the detailed comments. As mentioned in the earlier response to comment 2, we have updated Fig 1, adding more notations on the figure, drawing the goal point corresponding to each dynamic slot, and drawing the future trajectory corresponding to each goal point. In the caption of Fig.1, we also added a reminder to guide readers to see case studies figures for illustration of the dynamic insertion process: ``One can refer to Fig 5 6 7 for dynamic illustration of the insertion process." Notations in Sec 3 have also been added to Fig 2. We thank the reviewer for helping to make our paper more understandable.

---

> ### Author Response · Authors · 2022-10-11
> **Response to reviewer Qe3M (Part 7/10)**
>
> > Comment 19: Improved clarity surrounding whether DIAs are formed only with respect to other vehicles or also traffic signs/road geometries and whether $M$ vehicles correspond to the selected interacting vehicles or whether there are just $M$ vehicles in the scene. (Highly encouraged)
>
> **Response:** We thank the reviewer for the comments. DIAs can be formed by vehicles, traffic signs, and road geometries. Specifically, the rear boundary of one DIA is always a vehicle, but the front boundary of one DIA can be a vehicle, a traffic sign (e.g. stop sign and yield sign), the end point of one reference line, or a conflict point of two reference lines. Yes, the $M$ represents $M-1$ selected interacting vehicles plus the 1 ego vehicle, and also denotes that there are $M$ DIAs extracted from the scene. These contents and clarifications have been added to Sec. 3 and Sec. 4.1 in the updated manuscript.
>
> > Comment 20: Clarification for why some observations have two time coordinates, while others have one in Fig. 4 (e.g., $Y_{t-\tau,t}$ versus $Y_{t-\tau}$). (Highly encouraged)
>
> **Response:** We thank the reviewer for the detailed observation. We have checked Sec.3, Sec.5, Sec.6, Fig. 4, and made all the observation notations consistent to include two time coordinates, so that they can clearly indicate the time span.
>
> > Comment 21: Clarification in the first paragraph of Sec. 6.2 for what was the problem considered by Abuduweili & Liu (2021) as compared to driving behavior prediction. (Highly encouraged)
>
> **Response:** We thank the reviewer for the comments. The work by Abuduweili and Liu applied the online adaptation on the human-motion tracking problem (predicting the trajectory of the human wrist). Clarifications have been added to the updated manuscript.
>
> > Comment 22: There was no validation set used in the experiments. Do the authors believe that this could have resulted in over-fitting to the test set in the results? (Nice to have)
>
> **Response:** We thank the reviewer for the detailed comments. In fact, the shown performance on the trained scenario is indeed evaluated on the validation set. The showed performance on transferred scenarios is evaluated on unseen data. So we think such comparisons are fair and the over-fitting issues do not hold. We thank the reviewer for pointing this out and have added these clarifications at the beginning of Sec 7.4, Sec 7.5, and in each evaluation table.
>
> > Comment 23: The tables have distributional metrics indicated. Are these standard errors over the data? Were the experiments run over multiple seeds? (Highly encouraged)
>
> **Response:** We thank the reviewer for the detailed comments. We trained each of the methods with three different seeds and displayed the best performance. The distributional metrics denote the standard deviations of the prediction errors when we apply the best model on all the data. These clarifications have been added in Sec 7.1.
>
> > Comment 24: In Table 3, were HATN and all other considered learning-based methods trained only on the intersection scenario and then one-shot transferred to the roundabout scenario? How were methods like Trajectron++ able to generalize given that the HD map serves as input into that architecture? (Highly encouraged)
>
> **Response:** We thank the reviewer for raising the question. Yes, they are all trained on the intersection scenarios, and then directly transferred to unseen scenarios without training a new set of parameters. Yes, we do think it is hard for methods like Trajectron++ that directly takes HD map (e.g. BEV images) as scene information to generalize in scenes that are not well covered in training data. And this is exactly the advantage of our method, which takes the semantic graph, a generic representation of the scene. Our experiment results in Table 3 also support such thoughts that our method has better transferability.
>
> > Comment 25: Real-time performance discussion should be moved to the main paper. Why were the adaptation run times not included in Fig. 12? (Highly encouraged)
>
> **Response:** We thank the reviewer for the suggestions. We have moved these contents to the Sec. 7.6 in the updated manuscript. As for the online adaptation, the running time correlates with the adaptation steps. For instance, the adaptation took an average of 0.03 seconds per sample when set the adaptation step as 1, but the average time for the adaptation step of 3 was 0.1 seconds per sample. These introductions were provided by texts in the Sec 7.6.

---

> ### Author Response · Authors · 2022-10-11
> **Response to reviewer Qe3M (Part 8/10)**
>
>
> > Comment 26: A discussion should be included for how the hierarchical and interpretability goals of the approach are achieved while not constraining the low-level policy to end in the goal position. (Critical)
>
> **Response:** Thanks for the comments. This question is responded in comment 3, and we also attach the response here for easy checking. Actually, the end point is indeed used to guide the trajectory prediction in the low-level policy, as indicated in Fig.2, Sec 5.2, Fig 8, Table 2, and the advantage of hierarchies and interpretability is still valid. We apologize for not making this clear in the early stage of the paper, and not expressing this information clearly in Fig.1, which leads to confusion. We have updated Fig.1, strengthening the relationships between dynamic slot, goal point, and future trajectory. We really thank the reviewer for pointing this out and making our paper clear.
>
> > Comment 27: Some context around occupancy-based BEV prediction approaches would be helpful (see above [1-6]). (Nice to have)
>
> **Response:** We thank the reviewer for suggesting these papers that utilize the occupancy grid as one of the input representations. This request is responded previously in comment 4, but we also attach the response here for easy checking. For these six papers [1-6], they all utilized dynamic occupancy grid maps to store the dynamic state of agents in each grid and the occupancy belief, which is then treated as images and fed into CNN-based networks. However, one of the major drawbacks of such representation is that it usually does not consider HD map-related information such as lane and lane relations. Such information is important for accurate motion prediction, especially in interactive scenarios. Besides, there is an uneasy trade-off between the resolution of the spatial grid and the field of view. Moreover, representations obtained from the occupancy grid by CNN-based models are with high abstraction levels, which may fail under scenarios that are not well covered by the training data. These discussions have been added to Sec 2 in the updated manuscript.
>
> > Comment 28: Clarification surrounding the contribution of the paper as compared to prior work. (Critical)
>
> **Response:** We thank the reviewer for pointing this out. The comment is similar to the previous comment 5, but we also attach the response here. The novelty of our paper, in our opinion, mainly lies in our hierarchical framework and our insight of 1) dividing the monolithic driving task into several simplified sub-tasks; 2) applying generic and unified representation for each sub-task so that they are reusable and transferable across different scenarios; 3) online adapt the model parameter to deal with the heterogeneity and stochasticity in the prediction task. To the best of our knowledge, this is the very first method to explicitly and simultaneously take the driver’s nature of hierarchy, transferability, and adaptability into account. As for the three components in our paper (SGN, EDN, OA), they are basically the specific algorithms we choose to achieve our goal of hierarchy, transferability, and adaptability, and there can be some other choices on the specific algorithms. Besides, these algorithms are not immediately suitable to achieve our goal and some important modifications need to be made to make the solution holistic. For example, the modifications for the high-level policy (SGN) include: 1) the original method in Hu et al, 2020 considers how the vehicles will insert into DIAs, such as which location in the DIA to insert into, and how much time is needed for the insertion. These features can describe the relationship between the vehicles and inserted DIAs. However, these features can not describe the relationship between the vehicles and the lane that the vehicles run on, and cannot be used to guide the trajectory prediction in the low-level policy. Thus we modified the algorithm to predict the vehicles' own goal point, which can then be used in the low-level policy. 2) our SGN takes the spatial-temporal 3D semantic graph from historic time step $t − T_h$ to the current time step $t$ as the input, rather than only the spatial 2D semantic graph of the current time step t in previous work. So that the temporal information of the DIAs is considered. 3) our SGN considers the prediction of all vehicles in the scene instead of only the ego vehicle, to further augment the data. The effect of the two changes is ablated in Table 2 of the updated manuscript (No-Temporal method, and Single-Agent method). It turns out that these modifications have been pretty effective and important. The goal point prediction error of the No-Temporal and Single-Agent methods is higher than our method by 69% and 7% respectively.

---

> ### Author Response · Authors · 2022-10-11
> **Response to reviewer Qe3M (Part 9/10)**
>
> > Comment 29: More detailed definitions for relative and absolute features as well as position alignment and incremental prediction would be useful. (Highly encouraged)
>
> **Response:** We thank the reviewer for the detailed comments. This comment about absolute and relative features is partly similar to comment 7, but we also attach it here for easy checking. Actually, we have defined the relative and absolute features in the first paragraph of Sec 4.1, but it seems these statements have not been obvious and rich enough. Thus we have enriched the statements and added additional notations for clearer definition in the updated manuscripts, which is also attached below:
>
> *"To capture each DIA's crucial information for humans' decision, we extract four high-level features under the Frenet coordinate for each DIA: $\textbf{X}=(d_{f/r}^{lon}, v_{f/r}, \phi_{f/r}, l)$, where $d_{f/r}^{lon}$ denotes longitudinal distance to the conflict point of front or rear boundary; $v_{f/r}$ denotes the velocity of front or rear boundary; $\phi_{f/r}$ denotes the angle of front or rear boundary; $l = d_r^{lon} - d_f^{lon}$ measures the length of the DIA. These features are defined as the absolute features $\textbf{X}$. To facilitate relationship inference among DIAs, we also define the relative feature $\textbf{X}'$ for each DIA by aligning it with the reference DIA. Specifically, we choose the front DIA as the reference DIA $\textbf{X}_{ref}$, because the ego vehicle is implicitly represented by the rear boundary of the front DIA. Each DIA's relative feature $\textbf{X}'$ is then derived by subtracting the DIA's absolute feature $\textbf{X}$ by the reference DIA's absolute feature $\textbf{X}_{ref}$."*
>
> For the position alignment and incremental prediction, we have added detailed descriptions in Sec 7.4, as attached below:
>
> *The position alignment trick aligns the positions of each step to the vehicle's current position (Park et al., 2018). Specifically, when considering the prediction of a trajectory, the coordinate of all waypoints on the trajectory will be subtracted by the coordinate of the waypoint in the current step. The incremental prediction trick predicts the relative position compared to the position of the last step (displacement of each step), rather than directly predicting the absolute position n (Li et al., 2019).*
>
> > Comment 30: Including ground-truth future trajectories in the illustrations. Potential update to the color scheme (see above for suggestions). (Highly encouraged)
>
> **Response:** We thank the reviewer for the valuable suggestions that can make our illustration more clear and illustrative. In Fig. 5-7 of our updated manuscript: 1) we have added the ground-truth trajectory for each vehicle; 2) we have also added dashed-line box and one circle node to mark each DIA, and used transparency to present the likelihood; 3) we have made non-interacting cars gray and transparent.
>
> > Comment 31: Fix the color referencing in the caption of Fig. 6. (Critical)
>
> **Response:** We thank the reviewer for the detailed suggestion. We have checked the figure and made the color reference consistent.
>
> > Comment 32: Significantly improved paper polish. (Critical)
>
> **Response:** Thanks for the detailed comments to polish our paper. All the mentioned issues have been resolved and we have proof-read our paper again to improve it. We will continue to polish the paper after the rebuttal window.
>
> > Comment 33: More detailed discussion of limitations. (Highly encouraged)
>
> **Response:** We thank the reviewer for the valuable comment. This comment is related to the previous comment, but we also attach the response here. We quite agree with the reviewer that, though training the model in a hierarchical manner has multiple advantages, it can also have disadvantages that each module may fall in the local optima in the intermediate objectives and does not always align with the overall 'end-goal' system performance. Thus it is pretty interesting and important to 1) evaluate each module's effect on the overall system performance, 2) explore more 'end-goal' objectives as the two papers mentioned by the reviewer, 3) add an additional end-to-end training stage to explore the interaction between high-level policy and low-level policy, where they can accommodate to each other and improve overall performance. We thank the reviewer for pointing this out and has added these discussions on our limitation and future work in Sec 8. Besides, other limitations are also discussed in Sec 8, such as including vehicle dynamics into the learning model for dynamic-feasible prediction, and multimodality in trajectory generation.

---

> ### Author Response · Authors · 2022-10-11
> **Response to reviewer Qe3M (Part 10/10)**
>
>
> > Comment 34: The references should be proofread (e.g., to ensure the year is not entered twice in a citation, conference proceedings are formatted consistently, the conference is listed instead of ArXiv whenever possible). (Nice to have)
>
> **Response:** we thank the reviewer for the detailed comments. During the limited time window of the rebuttal phase, we have quickly proofread the reference, and will continue to proofread the reference after the rebuttal window.
>
> > Comment 35: The inclusion of context in Sec. B.1 for why GAT might do better than the proposed method. (Nice to have)
>
> **Response:** We thank the reviewer for the detailed comment. This comment is similar to comment 15 but we also attach the response here for easy checking. We think this phenomenon is as interesting as that of the No-Temporal method. The No-Temporal method has the best transferability performance, and one possible explanation is that the absence of temporal information constrained the model's capability and thus avoided over-fit. The reason why the GAT performs best in the inserted area prediction is possibly similar: the inserted area prediction is a relatively simple classification task. While absolute features may be enough for the model to learn, additional relative features may actually provide redundant information and make learning more difficult. We thank the reviewer for the question and are also open to other explanations.
>
> > Comment 36: Improved precision on the conclusion in Sec. 4 regarding attention not being helpful to the behavior prediction task (see above). (Highly encouraged)
>
> We thank the reviewer for the suggestion. We agree that the conclusion should be made more precisely. There are actually multiple aspects of attention. In high-level policy, we did apply the attention mechanism to help with the relationship/interaction reasoning among multiple agents, which is consistent with Trajectron++[7] and many other methods [8-10]. But in the low-level policy, the attention is on the temporal information, namely attending to historic steps differently when predicting each step of the future trajectory. Such temporal attention has been quite popular in language processing field [11][12]. However, our experiments in Table 6 indicate that such attention on one vehicle’s own historic trajectory did not help much. We have added such discussions in the Appendix B.4 of our updated manuscript.
>
> [7] T. Salzmann, B. Ivanovic, P. Chakravarty, and M. Pavone, “Trajectron++: Dynamically-feasible trajectory forecasting with heterogeneous data,” in European Conference on Computer Vision, pp. 683–700, Springer, 2020
>
> [8] C. Tang and R. R. Salakhutdinov, “Multiple futures prediction,” Advances in Neural Information Processing Systems, vol. 32, 2019.
>
> [9] Y. Hu, W. Zhan, and M. Tomizuka, “Scenario-transferable semantic graph reasoning for interaction-aware probabilistic prediction,” IEEE Transactions on Intelligent Transportation Systems, 2022.
>
> [10] Jiachen Li, Fan Yang, Masayoshi Tomizuka, and Chiho Choi. Evolvegraph: Multi-agent trajectory prediction with dynamic relational reasoning. Advances in neural information processing systems, 33:19783–19794, 2020.
>
> [11] . Yang, D. Yang, C. Dyer, X. He, A. Smola, and E. Hovy, “Hierarchical attention networks for document classification,” in Proceedings of the 2016 conference of the North American chapter of the association for computational linguistics: human language technologies, pp. 1480–1489, 2016
>
> [12] M.-T. Luong, H. Pham, and C. D. Manning, “Effective approaches to attention-based neural machine translation,” arXiv preprint arXiv:1508.04025, 2015

---

### Review · Reviewer_Ah68 · 2022-09-27

**Summary Of Contributions:**

The paper proposes a hierarchical model for motion prediction for self-driving. The model predicts a distribution over plausible "insertion slots" for the ego vehicle. Then, given a chosen insertion slot, the model deterministically outputs a trajectory. Additionally, the authors propose a method for adapting the weights of the model in real time based on the error between the trajectories predicted by the model and the observed trajectories of the vehicles in the scene.

**Broader Impact Concerns:**

I think broader impact is discussed a sufficient amount in this paper. For instance, the paper emphasizes that accurate motion prediction is critical for safe self-driving.

**Requested Changes:**

Changes that are critical to securing my recommendation for acceptance are bolded.

- Figure 1 - in (b) would it make sense to only color the the DIA that was selected? My understanding is only a selected DIA is fed to the EDN.
- **Figure 9** - after a few adaptation steps, using the MEKF appears to be worse than simply keeping the weights of the network fixed. The authors need to improve this result, otherwise I don't consider the adaptation method a contribution of the paper.
- **page 3 contributions** - "extensive experiments" and "thorough ablation" seems a bit extreme. The authors need to test on Argoverse, Waymo Open, or nuScenes for me to accept these claims as a contribution.
- Related Works intro - I'm not sure it makes sense to mention that the paper was previously presented at a workshop. I found this mention surprising but I defer to other reviewers who might know the policy for whether it's important to mention the existence of prior versions of the paper in a submission.
- Related Work "these methods lack flexibility as they usually only consider a fixed number of agents... these methods are also not order-invariant" - CNN-based approaches can handle an arbitrary number of agents and are permutation-invariant. I'm confused which model the authors are referring to here.
- Related Work "However, though equipped with hierarchies, most of these methods are still trained in an end-to-end manner". End-to-end training is generally considered desirable, why do the authors believe the fact that these methods are end-to-end trainable is a downside?
- in Equation 3 and 4, I believe $g_t$ is overloaded? In equation 3, $g_t$ is a distribution for each vehicle. In equation 4, $g_t$ is a sample from that distribution for each vehicle.
- **Encoder-Decoder Network** - I think "controller" is a more intuitive name for this network and how it functions in the model. Encoder-Decoder is vague and sounds like a deep image segmentation CNN when this model is instead a shallow LSTM with an MLP on top.
- **Section 4.2 intro** - do the authors ablate the number of historical timesteps to show this adjustment is significant?
- **Section 4.2** - the mean of the goal state distribution is an unintuitive choice for me. The DIAs e.g. in Figure 3 are not a convex set so the mean might not even lie within a DIA. Additionally, if the authors use the mean, they should be training the model end-to-end and gradients will flow back through the SGN. The authors need to ablate this target-location choice.
- equation 15 - is there a reason to not use softmax? Softmax should be more numerically more stable.
- line above equation 16 - is it a distribution over the DIA's future goal state? I thought it would be a distribution over the vehicle's future goal state.
- **Eq 21** - why use recursion in the decoder? Iteratively decoding is too costly. The authors should compare against dense layers that output the future trajectory all at once as is commonly done in motion prediction (e.g. MultiPath++).
- **Figure 8** - also compare against constant velocity model?
- **Table 2** - adaptation does not improve performance. It's especially important that the authors compare against constant velocity predictions at the 0.3 second time scale.
- **Table 3** - authors should use a more standard benchmark to compare against learning-based approaches. It can be difficult to get an apples-to-apples comparison across methods otherwise.

**Strengths And Weaknesses:**

I think the main strength of this paper is that the contribution is conceptually simple. In my mind, the simplest summary of the model is an an extension of the model from Hu et al, 2020 to motion prediction. The extension involves a shallow network that predicts a trajectory conditioned on a goal-state with weights that can be updated in real time with a Kalman Filter.

The main weakness of the paper is that I'm not convinced the model is particularly good at motion prediction. My understanding of Figure 9 is that the adaptation algorithm proposed by the authors quickly becomes worse than not adapting I therefore don't think the authors can claim their adaptation algorithm improves "better customizability and transferability" as claimed on page 3.

The time horizon (0.3 seconds and 3 seconds) that the authors choose to evaluate on is also unusual. On standard motion prediction benchmarks (e.g. Argoverse, Waymo Open, nuScenes), the time horizon for evaluation is 8-10 seconds, which makes it unnecessarily difficult to compare the author's model against prior work. In my opinion, the authors need to include results on one of the 3 standard motion prediction benchmarks to support their page 3 claim of "extensive experiments" and "thorough ablation studies".

Additionally, the main result the authors demonstrate for the low-level controller is that conditioning on the goal-state and speed and yaw information decreases the ADE. This feature selection is standard (see e.g. MultiPath++). Altogether, the model feels to me more like a baseline extension of Hu et al, 2020, not a holistic approach to motion prediction.

---

> ### Author Response · Authors · 2022-10-11
> **Response to reviewer Ah68(Part 1/8)**
>
> We thank the reviewer for taking the time to review the paper and giving detailed and constructive suggestions. We are glad to know that you like the simplicity and the presentation of our idea. We also appreciate your valuable comments have can help us improve our paper! The followings are responses to the comments and suggestions you made. We hope that they could solve your concerns. The updated manuscript is also attached, where the blue color denotes modified or added contents. The index of some tables and figures may be different from the initially submitted paper due to the modifications, and we refer to the updated index in the response. Please let us know if you want to know anything further or discuss any questions and concerns further.
>
> > Comment 1: The main weakness of the paper is that I'm not convinced the model is particularly good at motion prediction. My understanding of Figure 9 is that the adaptation algorithm proposed by the authors quickly becomes worse than not adapting I therefore don't think the authors can claim their adaptation algorithm improves ``better customizability and transferability'' as claimed on page 3.
>
> **Response:** We thank the reviewer for the comments. First we need to clarify that the adaptation is used to further improve our performance, while the model itself without adaptation is actually working well for the motion prediction (See Table 4, our method without adaptation beats Trajectron++ by 41\%, 34\%, 38\%, 37\% in ADE 3s in four evaluated scenarios.)
>
> Second, we need to clarify that the performance of online adaptation depends on several hyperparameters (adaptation steps, adjusted layers). Such characteristics have been discussed at the end of Sec 6.1:
>
> *“When adaptation step $\tau$ is too small, we may not obtain enough information for the online adaptation to benefit future behavior prediction. Intuitively, the problem can be mitigated by using errors of more steps. However, there exists a $\tau$-step time lag in $\tau$-step adaptation strategy. Too many steps may also create a big gap between historic behavior and current behavior, so that the model adapted at an earlier time may be outdated and incapable of tracking the current behavior pattern. Thus there is indeed a trade-off between obtaining more information and maintaining behavior continuity when we increase observation steps $\tau$. It also remains theoretically unknown which layer is the best to adapt."*
>
> Thus in Figure 9, we are not trying to prove that the online adaptation is guaranteed to work under any circumstances. Instead, our contribution in Figure 9 is to empirically ablate these design choices, and to analyze and understand how the online adaptation works in different circumstances. It turns out that adapting the last layer of the network with 2 or 3 adaptation steps works the best. We believe such analysis can provide an understanding of the adaptation mechanism and prior knowledge of the design choices.
>
> Moreover, we need to clarify that the performance of online adaptation would also highly depend on the data distribution. When the data lies in a 'scattered' distribution, there will be more space for adaptation. Though the online adaptation only improved the near-future prediction by 20\% and 28\% on the two scenarios from the INTERACTION dataset in the submitted paper, we added an additional evaluation on an intersection scenario from the InD dataset. As shown in Table 3, on the InD dataset, the online adaptation can improve the prediction on ADE 3s and ADE 0.3s by 11\% and 27\% respectively, which demonstrates that the online adaptation is more effective on this InD dataset. Therefore we think the claim of "better customizability and transferability" is proper. We would also like to evaluate the adaptation on more scenarios and datasets in our future work, as discussed in Sec 8. We really thank the reviewer for the comment to help us make our evaluation more comprehensive.

---

> > ### Author Response · Authors · 2022-10-11
> > **Response to reviewer Ah68 (Part 2/8)**
> >
> > > Comment 2: The time horizon (0.3 seconds and 3 seconds) that the authors choose to evaluate on is also unusual. On standard motion prediction benchmarks (e.g. Argoverse, Waymo Open, nuScenes), the time horizon for evaluation is 8-10 seconds, which makes it unnecessarily difficult to compare the author's model against prior work. In my opinion, the authors need to include results on one of the 3 standard motion prediction benchmarks to support their page 3 claim of  "extensive experiments'' and "thorough ablation studies''.
> >
> > **Response:** We thank the reviewer for the comments. First we need to clarify that the choice on the prediction horizon would depend on the evaluation purpose. Therefore there are different time horizons in different benchmarks and methods, and the time horizon of 3 seconds is actually very common. For example, the Argoverse 1 benchmark indeed sets prediction time horizon as 3 seconds. In our paper, we added an additional horizon of 0.3s to further evaluate the near-future prediction performance. Such evaluations are valuable for real-time safe control of AVs which usually run at 10hz, and a small prediction shift can make a big difference in terms of safety especially in close-distance interaction. Besides, the problem of vehicle dynamics infeasibility is also quite common for learning-based prediction methods, and the horizon of 0.3s can help to evaluate this. For example, in Table 4, the Trajectron++ which integrates vehicle dynamics model during the learning works the best in the prediction horizon of 0.3s. Such results demonstrate the importance of including vehicle dynamics into the model and is listed in our future work, which was discussed at the end of Sec 7.7.
> >
> > Second, the choice on the benchmark also depends on the evaluation purpose. In our paper, except for the pure prediction accuracy, we also aim at evaluating: 1) scenario transferability: how methods works when it is directly transferred to unseen scenarios (roundabout, intersection); 2) individual adaptability: how exploiting vehicles' historic behavior can improve future prediction. The two evaluation goals require the data 1) to be centered around one scenario (roundabout, intersection) to explicitly evaluate scenario-wise performance; 2) to contain one vehicle's multiple trajectory segments as the adaptation source. For the three benchmarks mentioned by the reviewer, they collect data in a whole city instead of one specific scenario by a moving vehicle, so it is hard to explicitly evaluate the scenario transferability. Besides, these datasets provide only one single trajectory segment for one vehicle, which makes the online adaptation undoable. This is why we utilized the INTERACTION dataset, which collects data in one specific scenario by a drone fixed above the scenario (for transferability evaluation) and contains all the historic segments of vehicles in the scenario (for adaptability evaluation). Also note that the INTERACTION dataset has also been pretty popular due to its inclusion of highly interactive driving data, and is frequently used to evaluate interaction modeling and motion prediction. But we agree with the reviewer that evaluating on one of the three benchmarks could better support our claim of "extensive experiments" and "thorough ablation studies". Due to the fact that our methods require the dataset to provide centerlines of the road lanes, we choose to evaluate our method on the Argoverse 1 benchmark. As shown in Table 4, on the Argoverse 1 benchmark our method also beats other methods by a considerable margin (beat Trajectron++ by 40\% in ADE 3s). Besides, on the InD dataset which was added to evaluate adaptability in comment 1, our method also outperforms other methods (beat Trajectron++ by 44\% in ADE 3s). We thank the reviewer for the comment to help us make our evaluation more comprehensive and convincing.

---

> ### Author Response · Authors · 2022-10-11
> **Response to reviewer Ah68 (Part 3/8)**
>
> > Comment 3: Additionally, the main result the authors demonstrate for the low-level controller is that conditioning on the goal-state and speed and yaw information decreases the ADE. This feature selection is standard (see e.g. MultiPath++). Altogether, the model feels to me more like a baseline extension of Hu et al, 2020, not a holistic approach to motion prediction.
>
> **Response:** We thank the reviewer for raising the question. In the submitted paper, we only provide a brief summary of the ablation studies on the low-level policy in the main paper to constrain the page length. And many detailed results and evaluations on the high-level policy, low-level policy, and online adaptation are placed in the appendix. For example, in the low-level policy, except for the goal state, speed, and yaw mentioned by the reviewer, we also investigated the effect of coordinate, representation, and different training tricks. Though some of the techniques have been used in prior works as mentioned by the reviewer, we here wanted to comprehensively ablate the effect of these methods in detail, to provide a good reference for readers and researchers when they consider the problem. Moreover, more systematic ablation studies on the high-level policy and online adaptation are also provided in the appendix. According to other reviewers' comments and suggestions, these contents are of importance and we have moved part of them to the main paper. All of these contents can better help readers understand each component of our methods.
>
> The novelty of our paper, in our opinion, mainly lies in our hierarchical framework and our insight of 1) dividing the monolithic driving task into several simplified sub-tasks; 2) applying generic and unified representation for each sub-task so that they are reusable and transferable across different scenarios; 3) online adapt the model parameter to deal with the heterogeneity and stochasticity in the prediction task. To the best of our knowledge, this is the **very first method** to explicitly and simultaneously consider drivers’ nature of hierarchy, transferability, and adaptability. As for the three components in our paper (SGN, EDN, OA), they are basically the specific algorithms we choose to achieve our goal of hierarchy, transferability, and adaptability, and there can be some other choices on the specific algorithms. Besides, these algorithms are not immediately suitable to achieve our goal and some important modifications need to be made to make the solution holistic. For example, the modifications for the high-level policy (SGN) include: 1) the original method in Hu et al, 2020 considers how the vehicles will insert into DIAs, such as which location in the DIA to insert into, and how much time is needed for the insertion. However, these features are DIA-centric, and cannot be used to guide the trajectory prediction in the low-level policy. Thus we modified the algorithm to predict vehicles' own goal point, which can then be used in the low-level policy. 2) our SGN takes the spatial-temporal 3D semantic graph from historic time step $t − T_h$ to the current time step $t$ as the input, rather than only the spatial 2D semantic graph of the current time step t in previous work. So that the temporal information of the DIAs is considered. 3) our SGN considers the prediction of all vehicles in the scene instead of only the ego vehicle, to further augment the data. The effect of the two changes is ablated in Table 2 of the updated manuscript (No-Temporal method, and Single-Agent method). It turns out that these modifications have been pretty effective and important. The goal point prediction error of the No-Temporal and Single-Agent methods is higher than our method by 69\% and 7\% respectively.
>
> > Comment 4: Figure 1 - in (b) would it make sense to only color the DIA that was selected? My understanding is only a selected DIA is fed to the EDN.
>
> **Response:** We thank the reviewer for raising the question. Yes, in our current deterministic formulation, we only fed the most-likely DIA into the EDN to predict the most likely trajectory. Extending our framework to consider the multi-modalities is one important future step. Fig 1 is an introductive figure, thus we mark all the extracted DIAs to show our insight. Only marking one DIA may not illustrate the DIA extraction process and the decision-making process. Besides, we have updated Fig 1(b), adding goal points corresponding to DIAs, and adding trajectories corresponding to the goal points. Hope these changes can make our idea clear.

---

> ### Author Response · Authors · 2022-10-11
> **Response to reviewer Ah68 (Part 4/8)**
>
> > Comment 5: Figure 9 - after a few adaptation steps, using the MEKF appears to be worse than simply keeping the weights of the network fixed. The authors need to improve this result, otherwise I don't consider the adaptation method a contribution of the paper.
>
> **Response:** We thank the reviewer for the comments. This comment is similar to earlier comment 1, but we also attach the response here for easy checking with some modifications.
>
> We first need to clarify that the performance of online adaptation depends on several hyperparameters (adaptation steps, adjusted layers). Such characteristics have been discussed at the end of Sec 6.1:
>
> *“ When adaptation step $\tau$ is too small, we may not obtain enough information for the online adaptation to benefit future behavior prediction. Intuitively, the problem can be mitigated by using errors of more steps. However, there exists a $\tau$-step time lag in $\tau$-step adaptation strategy. Too many steps may also create a big gap between historic behavior and current behavior, so that the model adapted at an earlier time may be outdated and incapable of tracking the current behavior pattern. Thus there is indeed a trade-off between obtaining more information and maintaining behavior continuity when we increase observation steps $\tau$. It also remains theoretically unknown which layer is the best to adapt."*
>
> Thus in Figure 9, we are not trying to prove that the online adaptation is guaranteed to work under any circumstances. Instead, our contribution in Figure 9 is to empirically ablate these design choices, to analyze and understand how the online adaptation works in different circumstances. It turns out that adapting the last layer of the network with 2 or 3 adaptation steps works the best. We believe such analysis can provide an understanding of the adaptation mechanism and prior knowledge of the design choices.
>
> Moreover, we need to clarify that the performance of online adaptation would also highly depend on the data distribution. When the data lies in a 'scattered' distribution, there will be more space for adaptation. Though the online adaptation only improved the near-future prediction by 20% and 28% on the two scenarios from the INTERACTION dataset in the submitted paper, we added an additional evaluation on an intersection scenario from the InD dataset. As shown in Table 3, on the InD dataset, the online adaptation can improve the prediction on ADE 3s and ADE 0.3s by 11% and 27% respectively, which demonstrates that the online adaptation is more effective on this InD dataset. Therefore we think the claim of "better customizability and transferability" is proper. We would also like to evaluate the adaptation on more scenarios and datasets in our future work, as discussed in Sec 8. We really thank the reviewer for the comment to help us make our evaluation more comprehensive.

---

> ### Author Response · Authors · 2022-10-11
> **Response to reviewer Ah68 (Part 5/8)**
>
> > Comment 6: page 3 contributions - "extensive experiments'' and "thorough ablation'' seems a bit extreme. The authors need to test on Argoverse, Waymo Open, or nuScenes for me to accept these claims as a contribution.
>
> **Response:** We thank the reviewer for the comments. This comment is similar to comment 2, but we also attach the response here for easy checking. First we need to clarify that, in the submitted paper, we provided systematic and detailed ablation studies on the high-level policy, low-level policy, and adaptation module in the appendix, which may be not obvious. According to other reviewers' comments and suggestions, these contents are of importance and we have moved them to the main paper. But we also agree with the reviewer that evaluating on one of the three benchmarks could better support our claim of "extensive experiments" and "thorough ablation studies". Due to the fact that our methods require the dataset to provide centerlines of the road lanes, we choose to evaluate our method on the Argoverse 1 benchmark. As shown in Table 4, our method also beats other methods by a considerable margin (beat Trajectron++ by 40\% in ADE 3s). Besides, on the InD dataset which was added to evaluate adaptability in comment 1, our method also beats other methods (beat Trajectron++ by 44\% in ADE 3s). We thank the reviewer for the comment to help us make our evaluation more comprehensive and convincing.
>
> > Comment 7: Related Works intro - I'm not sure it makes sense to mention that the paper was previously presented at a workshop. I found this mention surprising but I defer to other reviewers who might know the policy for whether it's important to mention the existence of prior versions of the paper in a submission.
>
> **Response:** We thank the reviewer for raising the concern. Actually, the policy of TMLR is that papers previously published in conferences will not be accepted. But papers that were previously reported at non-archival workshops are suitable for TMLR. So we include this information to clarify that our paper belongs to the second case and is suitable for submission.
>
> > Comment 8: Related Work "these methods lack flexibility as they usually only consider a fixed number of agents... these methods are also not order-invariant" - CNN-based approaches can handle an arbitrary number of agents and are permutation-invariant. I'm confused which model the authors are referring to here.
>
> **Response:** We thank the reviewer for pointing out this. We agree the statements here have not been clear. We have modified this paragraph and also integrate comments from other reviewers, as attached below:
>
> *"When modelling agents' interaction, one intuitive idea is simply using Deep Neuron Network (DNN), by flattening features of all agents and feeding them into deep neuron networks. However, such designs lack flexibility as they usually only consider a fixed number of agents. On the other hand, These methods are also not order-invariant: processing agents in a different order would produce different results, while we would expect the same results for the same scene. Thus such methods have been rarely used in related works.
> To bypass these problems and further impose agents' spatial relationship in the reasoning process, Convolution Neural Network (CNN) applies convolution operations on data (commonly in grid or pixel form) to model spatial and temporal relationship, such as 3D voxelization, rasterization in 2D bird’s-eye view (BEV) and occupancy grid d (Radwan et al., 2020; Su et al., 2021; Itkina et al., 2019; Toyungyernsub et al., 2021; Lange et al., 2021; Mohajerin & Rohani, 2019; Thomas et al., 2022; Mahjourian et al., 2022; Lee et al., 2017; Zhao et al., 2019; Tang & Salakhutdinov, 2019). However, such representations have several drawbacks: 1) there is an uneasy trade-off between the resolution of the spatial grid/image and the field of view. 2) the occupancy grid does not take into account HD map-related information such as lane and lane relations, which are important for accurate future prediction, especially in interactive scenarios. 3) representations obtained from the BEV images or occupancy grids via CNNs are of high abstraction level, which may fail under scenarios that are not well covered by the training data. 4) explicit relationship reasoning among agents is still missing, and it is difficult to capture long-range interactions via convolutions with small receptive fields."*

---

> ### Author Response · Authors · 2022-10-11
> **Response to reviewer Ah68 (Part 6/8)**
>
> > Comment 9: Related Work "However, though equipped with hierarchies, most of these methods are still trained in an end-to-end manner". End-to-end training is generally considered desirable, why do the authors believe the fact that these methods are end-to-end trainable is a downside?
>
> **Response:** We thank the reviewer for the question. As summarized in [1][2], the hierarchical training has several advantages by dividing the task into several sub-tasks: 1) information hiding: instead of considering a large pool of information, each sub-task only cares about the compact and key information that is relevant to the sub-task. So that both the input space and output space can be reduced and the learning can be more efficient. 2) reward hiding: each sub-policy only cares about optimizing for its own sub-task, while the other sub-tasks can be ignored. So that the learning can be simplified, and each sub-policy can be evaluated and tuned individually. Besides, compared to the end-to-end training methods, the hierarchical training approaches can leverage the prior knowledge (labels) in the middle layers, providing more interpretability.
>
> [1] Dayan, Peter, and Geoffrey E. Hinton. "Feudal reinforcement learning." Advances in neural information processing systems 5 (1992).
>
> [2] Flet-Berliac, Yannis. "The promise of hierarchical reinforcement learning." The Gradient 9 (2019).
>
> > Comment 10: in Equation 3 and 4, I believe $g_t$ is overloaded? In equation 3, $g_t$ is a distribution for each vehicle. In equation 4, $g_t$ is a sample from that distribution for each vehicle.
>
> **Response:** We thank the reviewer for the detailed suggestion. We have modified Eq.3 and use $p(g_t)$ to denote the distribution, so that it is distinguished from $g_t$ in Eq.4.
>
> > Comment 11: Encoder-Decoder Network - I think ``controller'' is a more intuitive name for this network and how it functions in the model. Encoder-Decoder is vague and sounds like a deep image segmentation CNN when this model is instead a shallow LSTM with an MLP on top.
>
> **Response:** We thank the reviewer for the detailed comments. "Controller" seems not to be the best choice, since it will confuse with an actual model-based vehicle controller. But we do agree that here we can make the name more precise, so we changed its name to "Trajectory Encoder Decoder Network (TEDN)", which can state the function of the module clearer. Thanks for the good suggestion.
>
> > Comment 12: Section 4.2 intro - do the authors ablate the number of historical timesteps to show this adjustment is significant?
>
> **Response:** We thank the reviewer for the question. In Table 5 of the appendix in the submitted paper, we did ablate the effect of whether utilizing historic time steps or not. In the updated manuscript, such contents have been moved to Sec 7.3 of the main paper. Specifically, we evaluated the method No-Temporal, which does not consider historic information and only considers the information of the current time step $t$. As shown in Table 2 of the updated manuscript, the No-Temporal method had the highest error in goal point prediction among all the ablated methods (1.59 ADE and 3.62 ADE in the intersection and roundabout scenario). In comparison, our method has the lowest error in goal point prediction (0.94 ADE and 1.70 ADE in the intersection and roundabout scenario, 40\% and 53\% lower than that of the No-Temporal method). Such results demonstrate the necessity to consider temporal information in the intention goal point prediction.
>
> > Comment 13: Section 4.2 - the mean of the goal state distribution is an unintuitive choice for me. The DIAs e.g. in Figure 3 are not a convex set so the mean might not even lie within a DIA. Additionally, if the authors use the mean, they should be training the model end-to-end and gradients will flow back through the SGN. The authors need to ablate this target-location choice.
>
> **Response:** We thank the reviewer for the question. Actually, the goal point for one vehicle in a future horizon essentially lies around the vehicle's lane reference line and does not necessarily lie within the DIA. So the convex set issue does not hold. Apologize for the misleading and we have modified Fig 1 to better illustrate this. In Table 3 of the updated manuscript, we added an ablation method Goal-Pt-Sample, which samples in the predicted goal point distribution to get the goal point. This Goal-Pt-Sample turned out to have a much higher goal point prediction error (1.31 ADE and 1.97 ADE in two evaluated scenarios) compared to our method (0.94 ADE and 1.10 ADE in two evaluated scenarios), which demonstrates the advantage of choosing the mean of the distribution as the goa point.

---

> ### Author Response · Authors · 2022-10-11
> **Response to reviewer Ah68 (Part 7/8)**
>
> > Comment 14: equation 15 - is there a reason to not use softmax? Softmax should be more numerically more stable.
>
> **Response:** We thank the reviewer for the detailed comments and we agree that softmax should be numerically more stable. However, we empirically found that classification using Eq.15 is more stable and performs better than softmax in our experimental setting and thus we did not use softmax in our framework.
>
> > Comment 15: line above equation 16 - is it a distribution over the DIA's future goal state? I thought it would be a distribution over the vehicle's future goal state.
>
> **Response:** We thank the reviewer for the detailed comment. Yes, it should be the distribution of vehicle's future goal state. We have modified it in the updated manuscript.
>
> > Comment 16: Eq 21 - why use recursion in the decoder? Iteratively decoding is too costly. The authors should compare against dense layers that output the future trajectory all at once as is commonly done in motion prediction (e.g. MultiPath++).
>
> **Response:** We thank the reviewer for raising this question. Using recursion in the decoder has been very common in related works to capture the temporal information and relationship between each time step [3-7]. As a matter of fact, though MultiPath++ mentioned by the reviewer independently models the distribution over time steps for efficient decoding, it also uses an additional recursion procedure (Algorithm 1 of MultiPath++), where actions are propagated in vehicle control model to get positions of each step. As indicated at the end of Sec 3.6 in the MultiPathh++: *such a recursion procedure can be viewed as a special type of recurrent network without learned parameters, where the decoding stage mirrors other works which use a learned RNN (LSTM or GRU cells) to iteratively decode an embedding vector into a trajectory.* But we also thank the reviewer for the comment on computation time, we would take it as our future steps to consider.
>
> [3] Hong, Joey, Benjamin Sapp, and James Philbin. "Rules of the road: Predicting driving behavior with a convolutional model of semantic interactions." Proceedings of the IEEE/CVF Conference on Computer Vision and Pattern Recognition. 2019.
>
> [4] Khandelwal, Siddhesh, et al. "What-if motion prediction for autonomous driving." arXiv preprint arXiv:2008.10587 (2020).
>
> [5] Mercat, Jean, et al. "Multi-head attention for multi-modal joint vehicle motion forecasting." 2020 IEEE International Conference on Robotics and Automation (ICRA). IEEE, 2020.
>
> [6] Salzmann, Tim, et al. "Trajectron++: Multi-agent generative trajectory forecasting with heterogeneous data for control." (2020).
>
> [7] Tang, Charlie, and Russ R. Salakhutdinov. "Multiple futures prediction." Advances in Neural Information Processing Systems 32 (2019).
>
> > Comment 17: Figure 8 - also compare against constant velocity model?
>
> **Response:** We thank the reviewer for the suggestion. We have evaluated the constant velocity model and plotted its error by step in Figure 8, which grows far more quickly than our methods, especially in the long-term prediction error.
>
> > Comment 18: Table 2 - adaptation does not improve performance. It's especially important that the authors compare against constant velocity predictions at the 0.3 second time scale.
>
> **Response:** We thank the reviewer for the comment. First, we need to indicate that the performance of online adaptation would highly depend on the data distribution. When the data lies in a 'scattered' distribution, there will be more space for adaptation. Though the online adaptation only improved the near-future prediction by 20\% and 28\% on the two scenarios from the INTERACTION dataset in the submitted paper, we added an additional evaluation on an intersection scenario from the InD dataset. As shown in Table 3, on the InD dataset, the online adaptation can improve the prediction on ADE 3s and ADE 0.3s by 11\% and 27\% respectively, which demonstrates that the online adaptation is more effective on this dataset. We would also like to evaluate the adaptation on more scenarios and datasets in our future work, as discussed in Sec 8.
>
> We have also evaluated the constant velocity model and added it to Table 3 of the updated manuscript, which performs better than ours in the 0.3 second time scale. However, as mentioned earlier, the problem of vehicle dynamics infeasibility is quite common for learning-based prediction methods, and the horizon of 0.3s is set to evaluate this. For this problem, the Trajectron++ (Table 4) which integrates vehicle dynamics model during the learning works better than our method and the constant velocity model (Table 3) in the prediction horizon of 0.3s. Such results demonstrate the importance of including vehicle dynamics into the model and motivate us to consider this in our future work, which was discussed at the end of Sec 7.7 and Sec 8.

---

> ### Author Response · Authors · 2022-10-11
> **Response to reviewer Ah68 (Part 8/8)**
>
> > Comment 19: Table 3 - authors should use a more standard benchmark to compare against learning-based approaches. It can be difficult to get an apples-to-apples comparison across methods otherwise.
>
> **Response:** We thank the reviewer for the comment. The comment is similar to comment 2, but we also attach the response here for easy checking. First, we need to clarify that the choice on the benchmark also depends on the evaluation purpose. In our paper, except for the pure prediction accuracy, we also aim at evaluating: 1) scenario transferability: how methods work when it is directly transferred to unseen scenarios (roundabout, intersection); 2) individual adaptability: how exploiting vehicles' historic behavior can improve future prediction. The two evaluation goals require the data 1) to be centered around one scenario (roundabout, intersection) to explicitly evaluate scenario-wise performance; 2) to contain one vehicle's multiple trajectory segments as the adaptation source. For the three benchmarks mentioned by the reviewer, they collect data in a whole city instead of one specific scenario by a moving vehicle, so it is hard to explicitly evaluate the scenario transferability. Besides, these datasets provide only one single trajectory segment for one vehicle, which makes the online adaptation undoable. This is why we utilized the INTERACTION dataset, which collects data in one specific scenario by a drone fixed above the specific scenario (for transferability evaluation), and contains all the historic segments of vehicles in the scenario (for adaptability evaluation). Also note that the INTERACTION dataset has also been pretty popular due to its inclusion of highly interactive driving data, and is frequently used to evaluate interaction modeling and motion prediction. But we agree with the reviewer that evaluating on one of the three benchmarks could better support our claim of "extensive experiments" and "thorough ablation studies". Due to the fact that our methods require the dataset to provide centerlines of the road lanes, we choose to evaluate our method on the Argoverse 1 benchmark. As shown in Table 4, our method also beats other methods by a considerable margin (beat Trajectron++ by 40\% in ADE 3s). Besides, on the InD dataset which was added to evaluate adaptability in comment 1, our method also beats other methods (beat Trajectron++ by 44\% in ADE 3s). We thank the reviewer for the comment to help us make our evaluation more comprehensive.

---

### Comment · Reviewer_exFQ · 2022-10-11
**Response editing**

Dear authors,

Please note that each time that you edit your answers, the reviewers receive an email notification. It would be great if you post final answers to avoid sending us 30+ emails.

Best regards,
exFQ

---

> ### Author Response · Authors · 2022-10-11
> **Response to note from reviewer exFQ**
>
> Dear reviewer exFQ,
>
> Thanks a lot for the kind reminding and sorry for the email interruptions. Yes, we are exactly uploading our final answers. It is just the case that each response has a character limit, so we have to split our response into multiple threads. Sorry for the interruption again and now I have basically finished the posting. Thanks a lot for your precious time reviewing our paper!
>
> Sincerely,
> Authors

---

### Decision · Action_Editors · 2022-11-26

**Recommendation:** Reject

**Comment:**

The primary contribution of the paper is the hierarchical approach to driving behavior prediction that reasons separately over the dynamic merge points (the "insertion slots") and the corresponding trajectories together with the ability to adapt these predictions online. The reviewers appreciate the difficulty of this problem and find that the proposed framework is well motivated. The initial submission provides a reasonably extensive evaluation of the architecture with a thorough set of ablations and results that demonstrate promising performance on the INTERACTION dataset. The reviewers raised several concerns with the original submission, most notably questions about the paper's primary contributions in light of existing work; the need to evaluate on more than one benchmark dataset and compare against additional baselines; and the need for more clarity regarding the nature of the proposed framework. It is clear from their responses that the authors have made considerable effort to address these concerns with the inclusion of additional experiments, including evaluations on a new benchmark that also demonstrate promising performance, and detailed replies to most if not all of the reviewers' questions and concerns. The reviewers appreciate the work that the authors have put into the paper and the review process, and find that the work has improved as a result. However, they agree that the paper does not adequately convey the novelty and significance of the hierarchical approach to behavior/trajectory prediction as currently presented. The work is promising and with further attention to improve the clarity of the presentation, the paper has the potential to provide a solid contribution.

**Audience:**

The problem is well motivated and the corresponding challenges are largely clear. With adequate clarity, the proposed approach will be of interest to the community.

**Claims And Evidence:**

The paper presents a hierarchical motion prediction framework for self-driving vehicles operating in the presence of other moving vehicles. Implemented using a semantic graph network, the high-level module predicts a distribution over possible dynamic "insertion slots" for the agent. An encoder-decoder architecture then predicts the vehicle's trajectory. An extended Kalman filter adapts the weight of the model online to improve performance based upon the error between the predicted and actual trajectories of the vehicles. The framework is evaluated on a real-world traffic dataset through an extensive set of ablations that analyze the contributions of the different model components. Experiments conducted during the review process evaluate performance on an additional benchmark dataset. The reviewers identified several key concerns with the initial submission regarding the lack of clarity regarding the significance of the contributions and the lack of experimental evidence to support several of the claims made in the paper. The authors have clearly put considerable effort into addressing these concerns, though as noted below, further improvements are necessary.